# What's in a Prior?
# Learned Proximal Networks for Inverse Problems

**Zhenghan Fang**[*]
Mathematical Institute for Data Science
Johns Hopkins University
`zfang23@jhu.edu`

**Sam Buchanan**[*]
Toyota Technological Institute at Chicago
`sam@ttic.edu`

**Jeremias Sulam**
Mathematical Institute for Data Science
Johns Hopkins University
`jsulam1@jhu.edu`

## Abstract

Proximal operators are ubiquitous in inverse problems, commonly appearing as part of algorithmic strategies to regularize problems that are otherwise ill-posed. Modern deep learning models have been brought to bear for these tasks too, as in the framework of plug-and-play or deep unrolling, where they loosely resemble proximal operators. Yet, something essential is lost in employing these purely data-driven approaches: there is no guarantee that a general deep network represents the proximal operator of any function, nor is there any characterization of the function for which the network might provide some approximate proximal. This not only makes guaranteeing convergence of iterative schemes challenging but, more fundamentally, complicates the analysis of what has been learned by these networks about their training data. Herein we provide a framework to develop *learned proximal networks* (LPN), prove that they provide exact proximal operators for a data-driven nonconvex regularizer, and show how a new training strategy, dubbed *proximal matching*, provably promotes the recovery of the log-prior of the true data distribution. Such LPN provide general, unsupervised, expressive proximal operators that can be used for general inverse problems with convergence guarantees. We illustrate our results in a series of cases of increasing complexity, demonstrating that these models not only result in state-of-the-art performance, but provide a window into the resulting priors learned from data.

## 1 Introduction

Inverse problems concern the task of estimating underlying variables that have undergone a degradation process, such as in denoising, deblurring, inpainting, or compressed sensing (Bertero et al., 2021; Ongie et al., 2020). Since these problems are naturally ill-posed, solutions to any of these problems involve, either implicitly or explicitly, the utilization of *priors*, or models, about what type of solutions are preferable (Engl et al., 1996; Benning & Burger, 2018; Arridge et al., 2019). Traditional methods model this prior directly, by constructing regularization functions that promote specific properties in the estimate, such as for it to be smooth (Tikhonov & Arsenin, 1977), piecewise smooth (Rudin et al., 1992; Bredies et al., 2010), or for it to have a sparse decomposition under a given basis or even a potentially overcomplete dictionary (Bruckstein et al., 2009; Sulam et al., 2014). On the other hand, from a machine learning perspective, the complete restoration mapping can also be modeled by a regression function and by providing a large collection of input-output (or clean-corrupted) pairs of samples (McCann et al., 2017; Ongie et al., 2020; Zhu et al., 2018).

---

[*]Equal contribution.

An interesting third alternative has combined these two approaches by making the insightful observation that many iterative solvers for inverse problems incorporate the application of the proximal operator for the regularizer. Such a proximal step can be loosely interpreted as a denoising step and, as a result, off-the-shelf strong-performing denoising algorithms (as those given by modern deep learning methods) can be employed as a subroutine. The Plug-and-Play (PnP) framework is a notable example where proximal operators are replaced with such denoisers (Venkatakrishnan et al., 2013; Zhang et al., 2017b; Meinhardt et al., 2017; Zhang et al., 2021; Kamilov et al., 2023b; Tachella et al., 2019), but these can be applied more broadly to solve inverse problems, as well (Romano et al., 2017; Romano & Elad, 2015). While this strategy works very well in practice, little is known about the approximation properties of these methods. For instance, *do these denoising networks actually (i.e., provably) provide a proximal operator for some regularization function?* Moreover, and from a variational perspective, *would this regularization function recover the correct regularizer, such as the (log) prior of the data distribution?* Partial answers to some of these questions exist, but how to address all of them in a single framework remains unclear (Hurault et al., 2022b; Lunz et al., 2018; Cohen et al., 2021a; Zou et al., 2023; Goujon et al., 2023) (see a thorough discussion of related works in Appendix A). More broadly, the ability to characterize a data-driven (potentially nonconvex) regularizer that enables good restoration is paramount in applications that demand notions of robustness and interpretability, and this remains an open challenge.

In this work, we address these questions by proposing a new class of deep neural networks, termed *learned proximal networks* (LPN), that *exactly implement the proximal operator* of a general learned function. Such a LPN implicitly learns a regularization function that can be characterized and evaluated, shedding light onto what has been learned from data. In turn, we present a new training problem, which we dub *proximal matching*, that provably promotes the recovery of the correct regularization term (i.e., the log of the data distribution), which need not be convex. Moreover, the ability of LPNs to implement exact proximal operators allows for guaranteed convergence to critical points of the variational problem, which we derive for PnP reconstruction algorithms under no additional assumptions on the trained LPN. We demonstrate through experiments on that our LPNs can recover the correct underlying data distribution, and further show that LPNs lead to state-of-the-art reconstruction performance on image deblurring, CT reconstruction and compressed sensing, while enabling precise characterization of the data-dependent prior learned by the model. Code for reproducing all experiments is made publicly available at https://github.com/Sulam-Group/learned-proximal-networks.

## 2 BACKGROUND

Consider an unknown signal in an Euclidean space[1], $\mathbf{x} \in \mathbb{R}^n$, and a known measurement operator, $A : \mathbb{R}^n \to \mathbb{R}^m$. The goal of inverse problems is to recover $\mathbf{x}$ from measurements $\mathbf{y} = A(\mathbf{x}) + \mathbf{v} \in \mathbb{R}^m$, where $\mathbf{v}$ is a noise or nuisance term. This problem is typically ill-posed: infinitely many solutions $\mathbf{x}$ may explain (i.e. approximate) the measurement $\mathbf{y}$ (Benning & Burger, 2018). Hence, a prior is needed to regularize the problem, which can generally take the form

$$\min_{\mathbf{x}} \frac{1}{2}\|\mathbf{y} - A(\mathbf{x})\|_2^2 + R(\mathbf{x}), \tag{2.1}$$

for a function $R(\mathbf{x}) : \mathbb{R}^n \to \mathbb{R}$ promoting a solution that is likely under the prior distribution of $\mathbf{x}$. We will make no assumptions on the convexity of $R(\mathbf{x})$ in this work.

**Proximal operators** Originally proposed by Moreau (1965) as a generalization of projection operators, proximal operators are central in optimizing the problem (2.1) by means of proximal gradient descent (PGD) (Beck, 2017), alternating direction method of multipliers (ADMM) (Boyd et al., 2011), or primal dual hybrid gradient (PDHG) (Chambolle & Pock, 2011). For a given functional $R$ as above, its proximal operator $\text{prox}_R$ is defined by

$$\text{prox}_R(\mathbf{y}) := \underset{\mathbf{x}}{\text{argmin}} \frac{1}{2}\|\mathbf{y} - \mathbf{x}\|^2 + R(\mathbf{x}). \tag{2.2}$$

When $R$ is non-convex, the solution to this problem may not be unique and the proximal mapping is set-valued. Following (Gribonval & Nikolova, 2020), we define the proximal operator of a function

---

[1]The analyses in this paper can be generalized directly to more general Hilbert spaces.

$R$ as a *selection* of the set-valued mapping: $f(\mathbf{y})$ is a proximal operator of $R$ if and only if $f(\mathbf{y}) \in \arg\min_{\mathbf{x}} \frac{1}{2}\|\mathbf{y} - \mathbf{x}\|^2 + R(\mathbf{x})$ for each $\mathbf{y} \in \mathbb{R}^n$. A key result in (Gribonval & Nikolova, 2020) is that the continuous proximal of a (potentially nonconvex) function can be fully characterized as the gradient of a convex function, as the following result formalizes.

**Proposition 2.1.** *[Characterization of continuous proximal operators, (Gribonval & Nikolova, 2020, Corollary 1)] Let $\mathcal{Y} \subset \mathbb{R}^n$ be non-empty and open and $f : \mathcal{Y} \to \mathbb{R}^n$ be a continuous function. Then, $f$ is a proximal operator of a function $R : \mathbb{R}^n \to \mathbb{R} \cup \{+\infty\}$ if and only if there exists a convex differentiable function $\psi$ such that $f(\mathbf{y}) = \nabla\psi(\mathbf{y})$ for each $\mathbf{y} \in \mathcal{Y}$.*

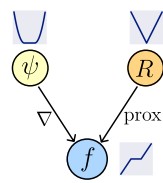

It is worth stressing the differences between $R$ and $\psi$. While $f$ is the proximal operator of $R$, i.e. $\text{prox}_R = f$, $f$ is also the gradient of a convex $\psi$, $\nabla\psi = f$ (see Figure 1). Furthermore, $R$ may be non-convex, while $\psi$ must be convex. As can be expected, there exists a precise relation between $R$ and $\psi$, and we will elaborate further on this connection shortly. The characterization of proximals of convex functions is similar but additionally requiring $f$ to be non-expansive (Moreau, 1965). Hence, by relaxing the nonexpansivity, we obtain a broader class of proximal operators. As we will show later, the ability to model proximal operators of non-convex functions will prove very useful in practice, as the log-priors[2] of most real-world data are indeed non-convex.

Figure 1: Sketch of Prop. 2.1 for $R(\cdot) = \|\cdot\|_1$.

**Plug-and-Play** This paper closely relates to the Plug-and-Play (PnP) framework (Venkatakrishnan et al., 2013). PnP employs off-the-shelf denoising algorithms to solve general inverse problems within an iterative optimization solver, such as PGD (Beck, 2017; Hurault et al., 2022b), ADMM (Boyd et al., 2011; Venkatakrishnan et al., 2013), half quadratic splitting (HQS) (Geman & Yang, 1995; Zhang et al., 2021), primal-dual hybrid gradient (PDHG) (Chambolle & Pock, 2011), and Douglas-Rachford splitting (DRS) (Douglas & Rachford, 1956; Lions & Mercier, 1979; Combettes & Pesquet, 2007; Hurault et al., 2022b). Inspired by the observation that $\text{prox}_R(\mathbf{y})$ resembles the *maximum a posteriori* (MAP) denoiser at $\mathbf{y}$ with a log-prior $R$, PnP replaces the explicit solution of this step with generic denoising algorithms, such as BM3D (Dabov et al., 2007; Venkatakrishnan et al., 2013) or CNN-based denoisers (Meinhardt et al., 2017; Zhang et al., 2017b; 2021; Kamilov et al., 2023b), bringing the benefits of advanced denoisers to general inverse problems. While useful in practice, such denoisers are *not* in general proximal operators. Indeed, modern denoisers need not be MAP estimators at all, but instead typically approximate a minimum mean squared error (MMSE) solution. Although deep learning denoisers have achieved impressive results when used with PnP, little is known about the implicit prior—if any—encoded in these denoisers, thus diminishing the interpretability of the reconstruction results. Some convergence guarantees have been derived for PnP with MMSE denoisers (Xu et al., 2020), chiefly relying on the assumption that the denoiser is non-expansive (which can be hard to verify or enforce in practice). Furthermore, when interpreted as proximal operators, the prior in MMSE denoisers can be drastically different from the original (true data) prior Gribonval (2011), raising concerns about correctness. There is a broad family of works that relate to the ideas in this work, and we expand on them in Appendix A.

## 3    LEARNED PROXIMAL NETWORKS

First, we seek a way to parameterize a neural network such that its mapping is the proximal operator of some (potentially nonconvex) scalar-valued functional. Motivated by Proposition 2.1, we will seek network architectures that parameterize *gradients of convex functions*. A simple way to achieve this is by differentiating a neural network that implements a convex function: given a scalar-valued neural network, $\psi_\theta : \mathbb{R}^n \to \mathbb{R}$, whose output is convex with respect to its input, we can parameterize a LPN as $f_\theta = \nabla\psi_\theta$, which can be efficiently evaluated via back propagation. This makes LPN a gradient field—and a conservative vector field—of an explicit convex function. Fortunately, this is not an entirely new problem. Amos et al. (2017) proposed input convex neural networks (ICNN) that guarantee to parameterize convex functions by constraining the network weights to be non-negative and the nonlinear activations convex and non-decreasing[3]. Consider a single-layer neural network characterized by the weights $\mathbf{W} \in \mathbb{R}^{m \times n}$, bias $\mathbf{b} \in \mathbb{R}^m$ and a scalar non-linearity $g : \mathbb{R} \to \mathbb{R}$. Such a network, at $\mathbf{y}$, is given by $\mathbf{z} = g(\mathbf{W}\mathbf{y} + \mathbf{b})$. With this notation, we now move to define our LPNs.

---

[2]In this paper, the "log-prior" of a data distribution $p_\mathbf{x}$ means its negative log-likelihood, $-\log p_\mathbf{x}$.

[3]Other ways to parameterize gradients of convex functions exist (Richter-Powell et al., 2021), but come with other constraints and limitations (see discussion in Appendix F.1).

**Proposition 3.1** (Learned Proximal Networks). *Consider a scalar-valued $(K+1)$-layered neural network $\psi_\theta : \mathbb{R}^n \to \mathbb{R}$ defined by $\psi_\theta(\mathbf{y}) = \mathbf{w}^T \mathbf{z}_K + b$ and the recursion*

$$\mathbf{z}_1 = g(\mathbf{H}_1\mathbf{y} + \mathbf{b}_1), \qquad \mathbf{z}_k = g(\mathbf{W}_k\mathbf{z}_{k-1} + \mathbf{H}_k\mathbf{y} + \mathbf{b}_k), \; k \in [2, K]$$

*where $\theta = \{\mathbf{w}, b, (\mathbf{W}_k)_{k=2}^K, (\mathbf{H}_k, \mathbf{b}_k)_{k=1}^K\}$ are learnable parameters, and $g$ is a convex, non-decreasing and $C^2$ scalar function, and $\mathbf{W}_k$ and $\mathbf{w}$ have non-negative entries. Let $f_\theta$ be the gradient map of $\psi_\theta$ w.r.t. its input, i.e. $f_\theta = \nabla_\mathbf{y}\psi_\theta$. Then, there exists a function $R_\theta : \mathbb{R}^n \to \mathbb{R} \cup \{+\infty\}$ such that $f_\theta(\mathbf{y}) \in \mathrm{prox}_{R_\theta}(\mathbf{y}), \; \forall \; \mathbf{y} \in \mathbb{R}^n$.*

The simple proof of this result follows by combining properties of ICNN from Amos et al. (2017) and the characterization of proximal operators from Gribonval & Nikolova (2020) (see Appendix C.1). The $C^2$ condition for the nonlinearity[4] $g$ is imposed to ensure differentiability of the ICNN $\psi_\theta$ and the LPN $f_\theta$, which will become useful in proving convergence for PnP algorithms with LPN in Section 4. Although this rules out popular choices like Rectifying Linear Units (ReLUs), there exist several alternatives satisfying these constraints. Following (Huang et al., 2021), we adopt the *softplus* function $g(x) = \frac{1}{\beta}\log(1 + \exp(\beta x))$, a $\beta$-smooth approximation of ReLU. Importantly, LPN can be highly expressive (representing any continuous proximal operator) under reasonable settings, given the universality of ICNN (Huang et al., 2021).

Networks defined by gradients of ICNN have been explored for inverse problems: Cohen et al. (2021a) used such networks to learn gradients of data-driven regularizers, thereby enforcing the learned regularizer to be convex. While this is useful for the analysis of the optimization problem, this cannot capture nonconvex log-priors that exist in most cases of interest. On the other hand, Hurault et al. (2022b) proposed parameterizing proximal operators as $f(\mathbf{y}) = \mathbf{y} - \nabla g(\mathbf{y})$, where $\nabla g$ is $L$-Lipschitz with $L < 1$. In practice, this is realized only approximately by regularizing its Lipschitz constant during training (see discussion in Appendix A). Separately, gradients of ICNNs are also important in data-driven optimal transport (Makkuva et al., 2020; Huang et al., 2021).

**Recovering the prior from its proximal**  Once an LPN $f_\theta$ is obtained, we would like to recover its prox-primitive[5], $R_\theta$. This is important, as this function is precisely the regularizer in the variational objective, $\min_\mathbf{x} \frac{1}{2}\|\mathbf{y} - A(\mathbf{x})\|_2^2 + R_\theta(\mathbf{x})$. Thus, being able to evaluate $R_\theta$ at arbitrary points provides explicit information about the prior, enhancing interpretability of the learned regularizer. We start with the relation between $f$, $R_\theta$ and $\psi_\theta$ from Gribonval & Nikolova (2020) given by

$$R_\theta(f_\theta(\mathbf{y})) = \langle \mathbf{y}, f_\theta(\mathbf{y}) \rangle - \frac{1}{2}\|f_\theta(\mathbf{y})\|_2^2 - \psi_\theta(\mathbf{y}). \tag{3.1}$$

Given our parameterization for $f_\theta$, all quantities are easily computable (via a forward pass of the LPN in Proposition 3.1). However, the above equation only allows to evaluate the regularizer $R_\theta$ at points in the image of $f_\theta$, $f_\theta(\mathbf{y})$, and not at an arbitrary point $\mathbf{x}$. Thus, we must invert $f_\theta$, i.e. find $\mathbf{y}$ such that $f_\theta(\mathbf{y}) = \mathbf{x}$. This inverse is nontrivial, since in general an LPN may not be invertible or even surjective. Thus, as in Huang et al. (2021), we add a quadratic term to $\psi_\theta$, $\psi_\theta(\mathbf{y}; \alpha) = \psi_\theta(\mathbf{y}) + \frac{\alpha}{2}\|\mathbf{y}\|_2^2$, with $\alpha > 0$, turning $\psi_\theta$ strongly convex, and its gradient map, $f_\theta = \nabla\psi_\theta$, invertible and bijective. To compute this inverse, it suffices to minimize the strongly convex objective

$$\min_\mathbf{y} \psi_\theta(\mathbf{y}; \alpha) - \langle \mathbf{x}, \mathbf{y} \rangle, \tag{3.2}$$

which has a unique global minimizer $\hat{\mathbf{y}}$ satisfying the first-order optimality condition $f_\theta(\hat{\mathbf{y}}) = \nabla\psi_\theta(\hat{\mathbf{y}}; \alpha) = \mathbf{x}$: the inverse we seek. Hence, computing the inverse amounts to solving a convex optimization problem—efficiently addressed by a variety of solvers, e.g. conjugate gradients.

Another feasible approach to invert $f_\theta$ is to simply optimize $\min_\mathbf{y} \|f_\theta(\mathbf{y}) - \mathbf{x}\|_2^2$, using, e.g., first-order methods. This problem is nonconvex in general, however, and thus does not allow for global convergence guarantees. Yet, we empirically find this approach work well on multiple datasets, yielding a solution $\hat{\mathbf{y}}$ with small mean squared error $\|f_\theta(\hat{\mathbf{y}}) - \mathbf{x}\|_2^2$. We summarize the procedures for estimating the regularizer from an LPN in Algorithm 2 and Appendix D.1.

---

[4]Proposition 3.1 also holds if the nonlinearities are different, which we omit for simplicity of presentation.
[5]Note our use of *prox-primitive* to refer to the function $R$ with respect to the operator $\mathrm{prox}_R$.

## 3.1 TRAINING LEARNED PROXIMAL NETWORKS VIA PROXIMAL MATCHING

To solve inverse problems correctly, it is crucial that LPNs capture the true proximal operator of the underlying data distribution. Given an unknown distribution $p_\mathbf{x}$, the goal of training an LPN is to learn the proximal operator of its log, $\text{prox}_{-\log p_\mathbf{x}} := f^*$. Unfortunately, paired ground-truth samples $\{\mathbf{x}_i, f^*(\mathbf{x}_i)\}$ do not exist in common settings—the prior distributions of many types of real-world data are unknown, making supervised training infeasible. Instead, we seek to train an LPN using *only i.i.d. samples from the unknown data distribution* in an unsupervised way.

To this end, we introduce a novel loss function that we call *proximal matching*. Based on the observation that the proximal operator is the *maximum a posteriori* (MAP) denoiser for additive Gaussian noise, i.e. for samples $\mathbf{y} = \mathbf{x} + \sigma\mathbf{v}$ with $\mathbf{x} \sim p_\mathbf{x}, \mathbf{v} \sim \mathcal{N}(0, \mathbf{I})$, we train LPN to perform *denoising* by minimizing a loss of the form

$$\mathbb{E}_{\mathbf{x},\mathbf{y}} \left[ d(f_\theta(\mathbf{y}), \mathbf{x}) \right], \tag{3.3}$$

where $d$ is a suitable metric. Popular choices for $d$ include the squared $\ell_2$ distance $\|f_\theta(\mathbf{y}) - \mathbf{x}\|_2^2$, the $\ell_1$ distance $\|f_\theta(\mathbf{y}) - \mathbf{x}\|_1$, or the Learned Perceptual Image Patch Similarity (LPIPS, (Zhang et al., 2018)), all of which have been used to train deep learning based denoisers (Zhang et al., 2017a; Yu et al., 2019; Tian et al., 2020). However, denoisers trained with these losses do not approximate the MAP denoiser, nor the proximal operator of the log-prior, $\text{prox}_{-\log p_\mathbf{x}}$. The squared $\ell_2$ distance, for instance, leads to the minimum mean square error (MMSE) estimategiven by the mean of the posterior, $\mathbb{E}[\mathbf{x} \mid \mathbf{y}]$. Similarly, the $\ell_1$ distance leads to the conditional marginal median of the posterior – and not its maximum. As a concrete example, Figure 2 illustrates the limitations of these metrics for learning the proximal operator of the log-prior of a Laplacian distribution.

We thus propose a new loss function that promotes the recovery of the correct proximal, dubbed **proximal matching loss**:

$$\mathcal{L}_{PM}(\theta; \gamma) = \mathbb{E}_{\mathbf{x},\mathbf{y}} \left[ m_\gamma(\|f_\theta(\mathbf{y}) - \mathbf{x}\|_2) \right], \quad m_\gamma(x) = 1 - \frac{1}{(\pi\gamma^2)^{n/2}} \exp\left(-\frac{x^2}{\gamma^2}\right), \gamma > 0. \tag{3.4}$$

Crucially, $\mathcal{L}_{PM}$ only depends on $p_\mathbf{x}$ (and Gaussian noise), allowing (approximate) proximal learning given only finite i.i.d. samples. Intuitively, $m_\gamma$ can be interpreted as an approximation to the Dirac function controlled by $\gamma$. Hence, minimizing the proximal matching loss $\mathcal{L}_{PM}$ amounts to maximizing the posterior probability $p_{\mathbf{x}|\mathbf{y}}(f_\theta(\mathbf{y}))$, and therefore results in the MAP denoiser (and equivalently, the proximal of log-prior). We now make this precise and show that minimizing $\mathcal{L}_{PM}$ yields the proximal operator of the log-prior almost surely as $\gamma \searrow 0$.

**Theorem 3.2** (Learning via Proximal Matching). *Consider a signal $\mathbf{x} \sim p_\mathbf{x}$, where $\mathbf{x}$ is bounded and $p_\mathbf{x}$ is a continuous density,[6] and a noisy observation $\mathbf{y} = \mathbf{x} + \sigma\mathbf{v}$, where $\mathbf{v} \sim \mathcal{N}(0, \mathbf{I})$ and $\sigma > 0$. Let $m_\gamma(x) : \mathbb{R} \to \mathbb{R}$ be defined as in (3.4). Consider the optimization problem*

$$f^* = \underset{f \text{ measurable}}{\operatorname{argmin}} \lim_{\gamma \searrow 0} \mathbb{E}_{\mathbf{x},\mathbf{y}} \left[ m_\gamma \left( \|f(\mathbf{y}) - \mathbf{x}\|_2 \right) \right]. \tag{3.5}$$

*Then, almost surely (i.e., for almost all $\mathbf{y}$), $f^*(\mathbf{y}) = \operatorname{argmax}_\mathbf{c} p_{\mathbf{x}|\mathbf{y}}(\mathbf{c}) \triangleq \text{prox}_{-\sigma^2 \log p_\mathbf{x}}(\mathbf{y})$.*

We defer the proof to Appendix C.2 and instead make a few remarks. First, while the result above was presented for the loss defined in (3.4) for simplicity, this holds in greater generality for loss functions satisfying specific technical conditions (see Appendix C.2). Second, an analogous result for discrete distributions can also be derived, and we include this companion result in Theorem B.1, Appendix B.1. Third, the Gaussian noise level $\sigma$ acts as a scaling factor on the learned regularizer, as indicated by $f^*(\mathbf{y}) = \text{prox}_{-\sigma^2 \log p_x}(\mathbf{y})$. Thus varying the noise level effectively varies the strength of the regularizer. Lastly, to bring this theoretical guarantee to practice, we progressively decrease $\gamma$ until a small positive amount during training according to a schedule function $\gamma(\cdot)$ for an empirical sample (instead of the expectation), and pretrain LPN with $\ell_1$ loss before proximal matching. We include an algorithmic description of training via proximal matching in Appendix D.2, Algorithm 3. Connections between the proximal matching loss (3.4) and prior work on impulse denoising and modal regression are discussed in Appendix A.

Before moving on, we summarize the results of this section: the parameterization in Proposition 3.1 guarantees that LPN implement a proximal operator for some regularizer function; the optimization

---

[6]That is, $\mathbf{x}$ admits a continuous probability density $p$ with respect to the Lebesgue measure on $\mathbb{R}^n$.

problem in (3.2) then provides a way to evaluate this regularizer function at arbitrary points; and lastly, Theorem 3.2 shows that if we want the LPN to recover the correct proximal (of the log-prior of data distribution), then *proximal matching* is the correct learning strategy for these networks.

## 4 SOLVING INVERSE PROBLEMS WITH LPN

Once an LPN is trained, it can be used to solve inverse problems within the PnP framework (Venkatakrishnan et al., 2013) by substituting any occurrence of the proximal step $\text{prox}_R$ with the learned proximal network $f_\theta$. As with any PnP method, our LPN can be flexibly plugged into a wide range of iterative algorithms, such as PGD, ADMM, or HQS. Chiefly, and in contrast to previous PnP approaches, our LPN-PnP approach provides the guarantee that the employed denoiser is indeed a proximal operator. As we will now show, this enables convergence guarantees absent any additional assumptions on the learned network. We provide an instance of solving inverse problems using LPN with PnP-ADMM in Algorithm 1, and another example with PnP-PGD in Algorithm 4.

**Convergence Guarantees in Plug-and-Play Frameworks** Because LPNs are by construction proximal operators, PnP schemes with plug-in LPNs correspond to iterative algorithms for minimizing the variational objective (2.1), with the implicitly-defined regularizer $R_\theta$ associated to the LPN. As a result, convergence guarantees for PnP schemes with LPNs follow readily from convergence analyses of the corresponding optimization procedure, under suitably general assumptions. We state and discuss such a guarantee for using an LPN with PnP-ADMM (Algorithm 1) in Theorem 4.1—our proof appeals to the nonconvex ADMM analysis of Themelis & Patrinos (2020).

---

**Algorithm 1** Solving inverse problem with LPN and PnP-ADMM

---

**Input:** Trained LPN $f_\theta$, operator $A$, measurement $\mathbf{y}$, initial $\mathbf{x}_0$, number of iterations $K$, penalty parameter $\rho$

1: $\mathbf{u}_0 \leftarrow 0, \mathbf{z}_0 \leftarrow \mathbf{x}_0$
2: **for** $k = 0$ to $K - 1$ **do**
3:     $\mathbf{x}_{k+1} \leftarrow \text{argmin}_\mathbf{x}\{\frac{1}{2}\|\mathbf{y} - A(\mathbf{x})\|_2^2 + \frac{\rho}{2}\|\mathbf{z}_k - \mathbf{u}_k - \mathbf{x}\|_2^2\}$
4:     $\mathbf{u}_{k+1} \leftarrow \mathbf{u}_k + \mathbf{x}_{k+1} - \mathbf{z}_k$
5:     $\mathbf{z}_{k+1} \leftarrow f_\theta(\mathbf{u}_{k+1} + \mathbf{x}_{k+1})$
6: **end for**
**Output:** $\mathbf{x}_K$

---

**Theorem 4.1** (Convergence guarantee for running PnP-ADMM with LPNs). *Consider the sequence of iterates $(\mathbf{x}_k, \mathbf{u}_k, \mathbf{z}_k)$, $k \in \{0, 1, \dots\}$, defined by Algorithm 1 run with a linear measurement operator $\mathbf{A}$ and an LPN $f_\theta$ with softplus activations, trained with $0 < \alpha < 1$. Assume further that the penalty parameter $\rho$ satisfies $\rho > \|\mathbf{A}^T\mathbf{A}\|$. Then the sequence of iterates $(\mathbf{x}_k, \mathbf{u}_k, \mathbf{z}_k)$ converges to a limit point $(\mathbf{x}^*, \mathbf{u}^*, \mathbf{z}^*)$ which is a fixed point of the PnP-ADMM iteration (Algorithm 1).*

We defer the proof of Theorem 4.1 to Appendix C.4.2. There, we moreover show that $\mathbf{x}_k$ converges to a critical point of the regularized reconstruction cost (2.1) with regularization function $R = \lambda R_\theta$, where $R_\theta$ is the implicitly-defined regularizer associated to $f_\theta$ (i.e. $f_\theta = \text{prox}_{R_\theta}$) and the regularization strength $\lambda$ depends on parameters of the PnP algorithm ($\lambda = \rho$ for PnP-ADMM). In addition, we emphasize that Theorem 4.1 requires the bare minimum of assumptions on the trained LPN: it holds for any LPNs by construction, under assumptions that are all actionable and achievable in practice (on network weights, activation, and strongly convex parameter). This should be contrasted to PnP schemes that utilize a black-box denoiser – convergence guarantees in this setting require restrictive assumptions on the denoiser, such as contractivity (Ryu et al., 2019), (firm) nonexpansivity (Sun et al., 2019; 2021; Cohen et al., 2021a;b; Tan et al., 2023), or other Lipschitz constraints (Hurault et al., 2022a;b), which are difficult to verify or enforce in practice without sacrificing denoising performance. Alternatively, other PnP schemes sacrifice expressivity for a principled approach by enforcing that the denoiser takes a restrictive form, such as being a (Gaussian) MMSE denoiser (Xu et al., 2020), a linear denoiser (Hauptmann et al., 2023), or the proximal operator of an implicit convex function (Sreehari et al., 2016; Teodoro et al., 2018).

The analysis of LPNs we use to prove Theorem 4.1 is general enough to be extended straightforwardly to other PnP optimization schemes. Under a similarly-minimal level of assumptions to Theorem 4.1, we give in Theorem B.2 (Appendix B.2) a convergence analysis for PnP-PGD (Algorithm 4), which tends to perform slightly worse than PnP-ADMM in practice.

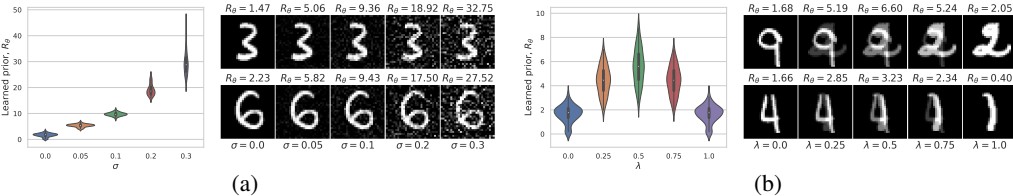

(a)                              (b)

Figure 3: Left: log-prior $R_\theta$ learned by LPN on MNIST (computed over 100 test images), evaluated at images corrupted by (a) additive Gaussian noise, and (b) convex combination of two images $(1 - \lambda)\mathbf{x} + \lambda\mathbf{x}'$. Right: the prior evaluated at individual examples.

## 5 EXPERIMENTS

We evaluate LPN on datasets of increasing complexity, from an analytical one-dimensional example of a Laplacian distribution to image datasets of increasing dimensions: MNIST ($28 \times 28$) (LeCun, 1998), CelebA ($128 \times 128$) (Liu et al., 2018), and Mayo-CT ($512 \times 512$) (McCollough, 2016). We

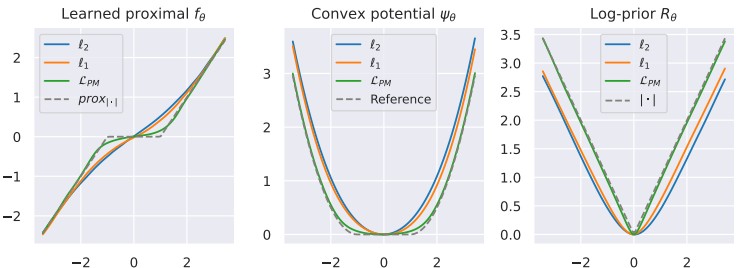

Figure 2: The proximal $f_\theta$, convex potential $\psi_\theta$, and log-prior $R_\theta$ learned by LPN via the squared $\ell_2$ loss, $\ell_1$ loss, and proximal matching loss $\mathcal{L}_{PM}$ for a Laplacian distribution (ground truth in gray).

demonstrate how the ability of LPN to learn an exact proximal for the correct prior reflects on natural values for the obtained log-likelihoods. Importantly, we showcase the performance of LPN for real-world inverse problems on CelebA and Mayo-CT, for deblurring, sparse-view tomographic reconstruction, and compressed sensing, comparing it with other state-of-the-art unsupervised approaches for (unsupervised) image restoration. See full experimental details in Appendix E.

### 5.1 WHAT IS YOUR PRIOR?

**Learning soft-thresholding from Laplacian distribution** We first experiment with a distribution whose log-prior has a known proximal operator, the 1-D Laplacian distribution $p(x \mid \mu, b) = \frac{1}{2b}\exp\left(-\frac{|x-\mu|}{b}\right)$. Letting $\mu = 0$, $b = 1$ for simplicity, the negative log likelihood (NLL) is the $\ell_1$ norm, $-\log p(x) = |x| - \log(\frac{1}{2})$, and its proximal can be written is the soft-thresholding function $\text{prox}_{-\log p}(x) = \text{sign}(x)\max(|x| - 1, 0)$. We train a LPN on i.i.d. samples from the Laplacian and Gaussian noise, as in (3.3), and compare different loss functions, including the proximal matching loss $\mathcal{L}_{\mathcal{PM}}$, for which we consider different $\gamma \in \{0.5, 0.3, 0.1\}$ in $\mathcal{L}_{\mathcal{PM}}$ (see (3.4)).

As seen in Figure 2, when using either the $\ell_2$ or $\ell_1$ loss, the learned prox differs from the correct soft-thresholding function. Indeed, verifying our analysis in Section 3.1, these yield the posterior mean and median, respectively, rather than the posterior mode. With the matching loss $\mathcal{L}_{PM}$ ($\gamma = 0.1$ in (3.4)), the learned proximal matches much more closely the ground-truth prox. The third panel in Figure 2 further depicts the learned log-prior $R_\theta$ associated with each LPN $f_\theta$, computed using the algorithm in Section 3. Note that $R_\theta$ does not match the ground-truth log-prior $|\cdot|$ for $\ell_2$ and $\ell_1$ losses, but converges to the correct prior with $\mathcal{L}_{PM}$ (see more results for different $\gamma$ in Appendix G.1). Note that we normalize the offset of learned priors by setting the minimum value to 0 for visualization: the learned log-prior $R_\theta$ has an arbitrary offset (since we only estimate the log-prior). In other words, LPN is only able to learn the relative density of the distribution due to the intrinsic scaling symmetry of the proximal operator.

**Learning a prior for MNIST** Next, we train an LPN on MNIST, attempting to learn a general restoration method for hand-written digits—and through it, a prior of the data. For images, we implement the LPN with convolution layers; see Appendix E.2 for more details. Once the model is learned, we evaluate the obtained prior on a series of inputs with different types and degrees of

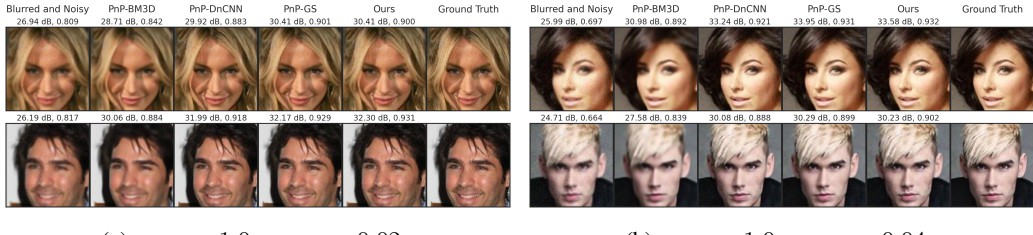

(a) $\sigma_{blur} = 1.0, \sigma_{noise} = 0.02$.        (b) $\sigma_{blur} = 1.0, \sigma_{noise} = 0.04$.

Figure 5: Visual results for deblurring on CelebA using Plug-and-Play with different denoisers (BM3D, DnCNN, the gradient step (GS) Prox-DRUNet, and our LPN), for different Gaussian blur kernel standard deviation $\sigma_{blur}$ and noise standard deviation $\sigma_{noise}$. PSNR and SSIM are presented above each prediction.

Table 1: Deblurring on CelebA, over 20 samples. **Bold** (underline) for the best (second best) score.

| METHOD | $\sigma_{blur}=1, \sigma_{noise}=.02$ | | $\sigma_{blur}=1, \sigma_{noise}=.04$ | | $\sigma_{blur}=2, \sigma_{noise}=.02$ | | $\sigma_{blur}=2, \sigma_{noise}=.04$ | |
|---|---|---|---|---|---|---|---|---|
| | PSNR($\uparrow$) | SSIM($\uparrow$) | PSNR($\uparrow$) | SSIM($\uparrow$) | PSNR($\uparrow$) | SSIM($\uparrow$) | PSNR($\uparrow$) | SSIM($\uparrow$) |
| Blurred and Noisy | $27.0 \pm 1.6$ | $.80 \pm .03$ | $24.9 \pm 1.0$ | $.63 \pm .05$ | $24.0 \pm 1.7$ | $.69 \pm .04$ | $22.8 \pm 1.3$ | $.54 \pm .04$ |
| PnP-BM3D (Venkatakrishnan et al., 2013) | $31.0 \pm 2.7$ | $.88 \pm .04$ | $29.5 \pm 2.2$ | $.84 \pm .05$ | $28.5 \pm 2.2$ | $.82 \pm .05$ | $27.6 \pm 2.0$ | $.79 \pm .05$ |
| PnP-DnCNN (Zhang et al., 2017a) | $32.3 \pm 2.6$ | $.90 \pm .03$ | $30.9 \pm 2.1$ | $.87 \pm .04$ | $29.5 \pm 2.0$ | $.84 \pm .04$ | $28.3 \pm 1.8$ | $.79 \pm .05$ |
| PnP-GS (Hurault et al., 2022b) | $\mathbf{33.0 \pm 3.0}$ | $\mathbf{.92 \pm .03}$ | $\mathbf{31.4 \pm 2.4}$ | $\mathbf{.89 \pm .03}$ | $\mathbf{30.1 \pm 2.5}$ | $\mathbf{.87 \pm .04}$ | $\mathbf{29.3 \pm 2.3}$ | $\mathbf{.84 \pm .05}$ |
| Ours | $\mathbf{33.0 \pm 2.9}$ | $\mathbf{.92 \pm .03}$ | $\underline{31.3 \pm 2.3}$ | $\mathbf{.89 \pm .03}$ | $\mathbf{30.1 \pm 2.4}$ | $\mathbf{.87 \pm .04}$ | $\underline{29.1 \pm 2.2}$ | $\mathbf{.84 \pm .04}$ |

perturbations in order to gauge how such modifications to the data are reflected by the learned prior. Figure 3a visualizes the change of prior $R_\theta$ after adding increasing levels of Gaussian noise. As expected, as the noise level increases, the values reported by the log-prior also increases, reflecting that noisier images are less likely according to the data distribution of real images.

The lower likelihood upon perturbations of the samples is general. We depict examples with image blur in Appendix G.2, and also present a study that depicts the non-convexity of the log-prior in Figure 3b: we eval-uate the learned prior at the convex combination of two samples, $\lambda \mathbf{x} + (1 - \lambda)\mathbf{x}'$ of two testing images $\mathbf{x}$ and $\mathbf{x}'$, with $\lambda \in [0, 1]$. As de-picted in Figure 3b, as $\lambda$ goes from 0 to 1, the learned prior first in-creases and then decreases, exhibit-ing a nonconvex shape. This is nat-ural, since the convex combination of two images no longer resembles a natural image, demonstrating that the true prior should indeed be noncon-vex. As we see, LPN can correctly learn this qualitative property in the prior, while existing approaches us-ing convex priors, either hand-crafted (Tikhonov & Arsenin, 1977; Rudin et al., 1992; Mallat, 1999; Beck &

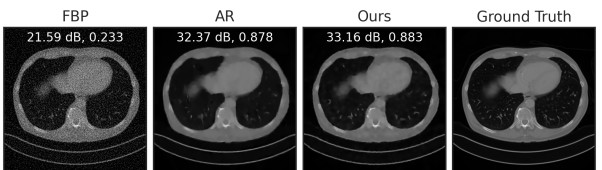

(a) Sparse-view tomographic reconstruction.

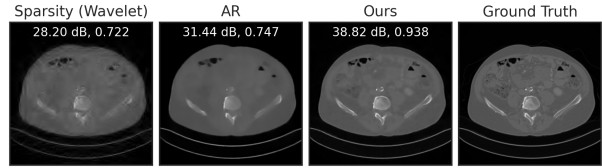

(b) Compressed sensing (compression rate = 1/16).

Figure 4: Results on the Mayo-CT dataset (details in text).

Teboulle, 2009; Elad & Aharon, 2006; Chambolle & Pock, 2011) or data-driven (Mukherjee et al., 2021; Cohen et al., 2021a), are suboptimal by not faithfully capturing such nonconvexity. All these results collectively show that LPN can learn a good approximation of the prior of images from data samples, and the learned prior either recovers the correct log-prior when it is known (in the Lapla-cian example), or provides a prior that coincides with human preference of natural, realistic images. With this at hand, we now move to address more challenging inverse problems.

## 5.2 SOLVING INVERSE PROBLEMS WITH LPN

**CelebA** We now showcase the capability of LPN for solving realistic inverse problems. We begin by training an LPN on the CelebA dataset, and employ the PnP-ADMM methodology for deblur-ring. We compare with state-of-the-art PnP approaches: PnP-BM3D (Venkatakrishnan et al., 2013),

which uses the BM3D denoiser (Dabov et al., 2007), PnP-DnCNN, which uses DnCNN as the denoiser (Zhang et al., 2017a) , and PnP-GS using the gradient step proximal denoiser called Prox-DRUNet (Hurault et al., 2022b). Both DnCNN and Prox-DRUNet have been trained on CelebA. As shown in Table 1, LPN achieves state-of-the-art result across multiple blur degrees, noise levels and metrics considered. As visualized in Figure 5, LPN significantly improves the quality of the blurred image, demonstrating the effectiveness of the learned prior for solving inverse problems.

Compared to the state-of-the-art methods, LPN can produce sharp images with comparable visual quality, while allowing for the evaluation of the obtained prior—which is impossible with any of the other methods.

**Mayo-CT** We train LPN on the public Mayo-CT dataset (McCollough, 2016) of Computed Tomography (CT) images, and evaluate it for two inverse tasks: sparse-view CT reconstruction and compressed sensing. For sparse-view CT reconstruction, we compare with filtered back-projection (FBP) (Willemink & Noël, 2019), the adversarial regularizer (AR) method of (Lunz et al., 2018) with an explicit regularizer, and its improved and subsequent version using unrolling (UAR) (Mukherjee et al., 2021). UAR is trained to solve the inverse problem for a specific measurement operator (i.e., task-specific), while both AR and LPN are generic regularizers that are applicable to any measurement model (i.e., task-agnostic). In other words, the comparison with UAR is not completely fair, but we still include it here for a broader comparison.

Table 2: Numerical results for inverse problems on Mayo-CT, computed over 128 test images.

| METHOD | PSNR (↑) | SSIM (↑) |
|---|---|---|
| **Tomographic reconstruction** | | |
| FBP | 21.29 | .203 |
| *Operator-agnostic* | | |
| AR (Lunz et al., 2018) | 33.48 | .890 |
| Ours | **34.14** | **.891** |
| *Operator-specific* | | |
| UAR (Mukherjee et al., 2021) | **34.76** | **.897** |
| **Compressed sensing (compression rate $= 1/16$)** | | |
| Sparsity (Wavelet) | 26.54 | .666 |
| AR (Lunz et al., 2018) | 29.71 | .712 |
| Ours | **38.03** | **.919** |
| **Compressed sensing (compression rate $= 1/4$)** | | |
| Sparsity (Wavelet) | 36.80 | .921 |
| AR (Lunz et al., 2018) | 37.94 | .920 |
| Ours | **44.05** | **.973** |

Following Lunz et al. (2018), we simulate CT sinograms using a parallel-beam geometry with 200 angles and 400 detectors, with an undersampling rate of $\frac{200 \times 400}{512^2} \approx 30\%$. See Appendix E.4 for experimental details. As visualized in Figure 4a, compared to the baseline FBP, LPN can significantly reduce noise in the reconstruction. Compared to AR, LPN result is slightly sharper, with higher PNSR. The numerical results in Table 2 show that our method significantly improves over the baseline FBP, outperforms the unsupervised counterpart AR, and performs just slightly worse than the supervised approach UAR—*without even having had access to the used forward operator*. Figure 4b and Table 2 show compressed sensing results with compression rates of $\frac{1}{4}$ and $\frac{1}{16}$. LPN significantly outperforms the baseline and AR, demonstrating better generalizability to different forward operators and inverse problems.

## 6   CONCLUSION

The learned proximal networks presented in this paper are guaranteed to parameterize proximal operators. We showed how the prox-primitive, regularizer function of the resulting proximal (parameterized by an LPN) can be recovered, allowing explicit characterization of the prior learned from data. Furthermore, via proximal matching, LPN can approximately learn the correct prox (i.e. that of the log-prior) of an unknown distribution from only i.i.d. samples. When used to solve general inverse problems, LPN achieves state-of-the-art results while providing more interpretability by explicit characterization of the (nonconvex) prior, with convergence guarantees. The ability to not only provide unsupervised models for general inverse problems but, chiefly, to characterize the priors learned from data open exciting new research questions of uncertainty quantification (Angelopoulos et al., 2022; Teneggi et al., 2023; Sun & Bouman, 2021), sampling (Kadkhodaie & Simoncelli, 2021; Kawar et al., 2021; Chung et al., 2022; Kawar et al., 2022; Feng et al., 2023), equivariant learning (Chen et al., 2023a; 2021; 2022a), learning without ground-truth (Tachella et al., 2023; 2022; Gao et al., 2023), and robustness (Jalal et al., 2021a; Darestani et al., 2021), all of which constitute matter of ongoing work.

ACKNOWLEDGMENTS

This research has been supported by NIH Grant P41EB031771, as well as by the Toffler Charitable Trust and by the Distinguished Graduate Student Fellows program of the KAVLI Neuroscience Discovery Institute.

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

## A   RELATED WORKS

**Deep Unrolling**   In addition to Plug-and-Play, deep unrolling is another approach using deep neural networks to replace proximal operators for solving inverse problems. Similar to PnP, the deep unrolling model is parameterized by an unrolled iterative algorithm, with certain (proximal) steps replaced by deep neural nets. In contrast to PnP, the unrolling model is trained in an end-to-end fashion by paired data of ground truth and corresponding measurements from specific forward operators. Truncated deep unrolling methods unfold the algorithm for a fixed number of steps (Gregor & LeCun, 2010; Adler et al., 2010; Liu et al., 2019; Aggarwal et al., 2018; Adler & Öktem, 2018; Zhang et al., 2020; Monga et al., 2021; Gilton et al., 2019; Tolooshams et al., 2023; Kobler et al., 2017; Chen et al., 2022b; Mardani et al., 2018; Sulam et al., 2019), while infinite-step models have been recently developed based on deep equilibrium learning (Gilton et al., 2021; Liu et al., 2022; Zou et al., 2023). In future work, LPN can improve the performance and interpretability of deep unrolling methods in e.g., medical applications (Lai et al., 2020; Fang et al., 2023; Shenoy et al., 2023) or in cases that demand the analysis of robustness (Sulam et al., 2020). The end-to-end supervision in unrolling can also help increase the performance of LPN-based methods for inverse problems in general.

**Explicit Regularizer**   A series of works have been dedicated to designing explicit data-driven regularizer for inverse problems, such as RED (Romano et al., 2017), AR (Lunz et al., 2018), ACR (Mukherjee et al., 2020), UAR (Mukherjee et al., 2021) and others (Li et al., 2020; Kobler et al., 2020; Cohen et al., 2021a; Zou et al., 2023; Goujon et al., 2023). Our work contributes a new angle to this field, by learning a proximal operator for the log-prior and then recovering the regularizer from the learned proximal.

**Gradient Denoiser**   Gradient step (GS) denoisers (Cohen et al., 2021a; Hurault et al., 2022a;b) are a cluster of recent approaches that parameterize a denoiser as a gradient descent step using the gradient map of a neural network. Although these works share similarities to our LPN, there are a few key differences.

1. Parameterization. In GS denoisers, the denoiser is defined as a gradient descent step: $f = \mathrm{Id} - \nabla g$, where $\mathrm{Id}$ represents the identity operator, and $g$ is a scalar-valued function that is either directly parameterized by a neural network (Cohen et al., 2021a), or implicitly defined by a network $N : \mathbb{R}^n \to \mathbb{R}^n$ as $g(\mathbf{x}) = \frac{1}{2}\|\mathbf{x} - N(\mathbf{x})\|_2^2$ (Hurault et al., 2022a;b). Cohen et al. (2021a) also experiment with a denoiser architecture analogous to our LPN architecture, but find its denoising performance to be inferior to the GS denoiser (we will discuss this further in the final bullet below). In order to have accompanying convergence guarantees when used in PnP schemes, these GS parameterizations demand special structures on the learned denoiser—in particular, Lipschitz constraints on $\nabla g$—which can be challenging to enforce in practice.

2. Proximal operator guarantee. The GS denoisers in Cohen et al. (2021a); Hurault et al. (2022a) are not a priori guaranteed to be proximal operators. Hurault et al. (2022b) proposed to constrain the GS denoiser to be a proximal operator by limiting the Lipschitz constant of $\nabla g$, also exploiting the characterization of Gribonval & Nikolova (2020). However, as a result, their denoiser necessarily has a bounded Lipschitz constant. Furthermore, in practice, such a constraint is not strictly enforced, but instead realized by adding a regularization term on the spectral norm of the network during training. Such a regularization only penalizes large Lipschitz constants, but does not guarantee that the Lipschitz constant will be lower than the required threshold. Additionally, the regularization is only computed at training data points, thus either not regularizing the network's behavior globally or resulting in loose upper-bounds for it. In other words, such proximal GS denoiser is only "encouraged" to resemble a proximal, but it is not guaranteed. On the other hand, our LPN provides the guarantee that the learned network will always parameterize a proximal operator.

3. Training. All GS denoiser methods used the conventional $\ell_2$ loss for training. We propose the proximal matching loss and show that it is essential for the network to learn the correct proximal operator of the log-prior of data distribution. Indeed, we attribute the inferior performance of the ICNN-based architecture that Cohen et al. (2021a) experiment with,

which is analogous to our LPN, to the fact that their experiments train this architecture on MMSE-based denoising, where "regression to the mean" on multimodal and nonlinear natural image data hinders performance (see, e.g., Delbracio & Milanfar (2023) in this connection). The key insight that powers our successful application of LPNs in experiments is the proximal matching training framework, which allows us to make full use of the constrained capacity of the LPN in representing highly expressive proximal operators (corresponding to (nearly) maximum a-posteriori estimators for data distributions).

**Comparisons to Diffusion Models**    Recently, score-based diffusion models have proven very efficient for unconditional and conditional image generation. There are several key differences between our work and diffusion models. First, conditional diffusion models do not minimize a variational problem as we do in this paper (as in (2.1)), but instead provide samples from the posterior distribution. Moreover, the diffusion models rely on inverting a diffusion process which requires an MMSE denoiser, and—just as in the case of regular denoisers—they do not approximate any MAP estimate, whereas we are concerned with networks that compute a MAP estimate for a learned prior. In terms of strict advantages, one should again note that our approach solves (provides a MAP estimate) for a denoising problem with a single forward-pass, whereas sampling with diffusion models requires a large sequence of forward passes of a denoising network. Lastly, but also importantly, our method provides an exact proximal operator for a learned prior distribution. Diffusion models have no such guarantee: all these results provide samples from an approximate posterior distribution, which relies on the approximation qualities of the MMSE denoiser that do not exist for general cases (Chen et al., 2023b).

**Proximal Matching Loss and Mode-Seeking Regression Objectives**    In the literature on both deep learning-based denoising and statistical methodology, prior works have explored training schemes that promote learning the mode of a distribution (or, in our denoising setting, the conditional mode/MAP estimate of the prior). On the methodological side, it is noted that training with respect to a *single* objective function cannot lead to the optimal denoiser being the mode uniformly over sufficiently-expressive classes of denoisers and priors, a concept formalized as *inelicitability of the mode functional* (Gneiting, 2011; Heinrich, 2014). In contrast, our nonparametric result on the proximal matching loss, Theorem 3.2, characterizes the minimizer of *a limit of a sequence of losses*. This is both outside the framework of the preceding references, and distinct from what occurs in practice, where we attempt to minimize the proximal matching loss with a sufficiently small parameter $\gamma > 0$. We expect this latter setting to coincide with correct learning of the mode/MAP estimate of the prior in practical settings of interest, when $\gamma$ is much smaller than the 'characteristic scale' of the prior. Prior work has also considered modal regression in an abstract statistical learning setting (Feng et al., 2020), where in contrast to our proximal matching-based objective, an approach based on kernel density estimation was advanced.

In the literature on learning deep denoisers, we note that a previous work (Lehtinen et al., 2018) used an annealed version of an "$\ell_0$ loss" for mode approximation, with motivation similar to that of proximal matching. Their loss takes a different form, $\sum_i (|f(\mathbf{y}) - \mathbf{x}|_i + \epsilon)^\gamma$, where $\epsilon$ is a small constant and $\gamma \in [0, 2]$ is the annealing parameter. Their loss is designed for learning from corrupted targets with random impulse noise, and does not recover the mode of the posterior (as in the case of proximal matching), but rather the zero-crossing of the Hilbert transform of the probability density function.

# B ADDITIONAL THEOREMS

## B.1 LEARNING VIA PROXIMAL MATCHING (DISCRETE CASE)

**Theorem B.1** (Learning via Proximal Matching (Discrete Case))**.** *Consider a signal* $\mathbf{x} \sim P(\mathbf{x})$, *with* $P(\mathbf{x})$ *a discrete distribution, and a noisy observation* $\mathbf{y} = \mathbf{x} + \sigma\varepsilon$, *where* $\varepsilon \sim \mathcal{N}(0, \mathbf{I})$ *and* $\sigma > 0$. *Let* $m_\gamma(x) : \mathbb{R} \to \mathbb{R}$ *be defined by* $m_\gamma(x) = 1 - \exp\left(-\frac{x^2}{\gamma^2}\right)$ [7]. *Consider the optimization*

---

[7]This definition of $m_\gamma$ differs slightly from the one in (3.4), but they are equivalent in terms of minimization objective as they only differ by a scaling constant.

*problem*

$$f^* = \underset{f \text{ measurable}}{\text{argmin}} \lim_{\gamma \searrow 0} \mathbb{E}_{\mathbf{x},\mathbf{y}} \left[ m_\gamma \left( \|f(\mathbf{y}) - \mathbf{x}\|_2 \right) \right].$$

*Then, almost surely (i.e., for almost all $\mathbf{y}$), $f^*(\mathbf{y}) = \text{argmax}_{\mathbf{c}} P(\mathbf{x} = \mathbf{c} \mid \mathbf{y})$.*

The proof is deferred to Appendix C.3.

### B.2 CONVERGENCE OF PNP-PGD USING LPN

**Theorem B.2** (Convergence guarantee for running PnP-PGD with LPNs). *Consider the sequence of iterates $\mathbf{x}_k$, $k \in \{0, 1, \dots\}$, defined by Algorithm 4 run with a linear measurement operator $\mathbf{A}$ and an LPN $f_\theta$ with softplus activations, trained with $0 < \alpha < 1$. Assume that the step size satisfies $0 < \eta < 1/\|\mathbf{A}^T\mathbf{A}\|$. Then, the iterates $\mathbf{x}_k$ converge to a fixed point $\mathbf{x}^*$ of Algorithm 4: that is, there exists $\mathbf{x}^* \in \mathbb{R}^n$ such that $\lim_{k \to \infty} \mathbf{x}_k = \mathbf{x}^*$, and*

$$f_\theta \left( \mathbf{x}^* - \eta \nabla h(\mathbf{x}^*) \right) = \mathbf{x}^*. \tag{B.1}$$

The proof is deferred to Appendix C.4.1.

## C PROOFS

In this section, we include the proofs for the results presented in this paper.

### C.1 PROOF OF PROPOSITION 3.1

*Proof.* By Amos et al. (2017, Proposition 1), $\psi_\theta$ is convex. Since the activation $g$ is differentiable, $\psi_\theta$ is also differentiable. Hence, $f_\theta = \nabla\psi_\theta$ is the gradient of a convex function. Thus, by Proposition 2.1, $f_\theta$ is a proximal operator of a function. □

### C.2 PROOF OF THEOREM 3.2

*Proof.* First, note by linearity of the expectation that for any measurable $f$, one has

$$\lim_{\gamma \searrow 0} \mathbb{E}_{\mathbf{x},\mathbf{y}} \left[ m_\gamma \left( \|f(\mathbf{y}) - \mathbf{x}\|_2 \right) \right] = 1 - \lim_{\gamma \searrow 0} \mathbb{E}_{\mathbf{x},\mathbf{y}} \left[ \varphi_{\gamma^2/2}(f(\mathbf{y}) - \mathbf{x}) \right], \tag{C.1}$$

where $\varphi_{\gamma^2/2}$ denotes the density of an isotropic Gaussian random variable with mean zero and variance $\gamma^2/2$. Because $p(\mathbf{x})$ is a continuous density with respect to the Lebesgue measure $d\mathbf{x}$, by Gaussian conditioning, we have that the conditional distribution of $\mathbf{x}$ given $\mathbf{y}$ admits a density $p_{\mathbf{x}|\mathbf{y}}$ with respect to $d\mathbf{x}$ as well. Taking conditional expectations, we have

$$\lim_{\gamma \searrow 0} \mathbb{E}_{\mathbf{x},\mathbf{y}} \left[ \varphi_{\gamma^2/2}(f(\mathbf{y}) - \mathbf{x}) \right] = \lim_{\gamma \searrow 0} \mathbb{E}_{\mathbf{y}} \mathbb{E}_{\mathbf{x}|\mathbf{y}} \left[ \varphi_{\gamma^2/2}(f(\mathbf{y}) - \mathbf{x}) \right]. \tag{C.2}$$

From here, we can state the intuition for the remaining portion of the proof. Intuitively, because the Gaussian density $\varphi_{\sigma^2/2}$ concentrates more and more at zero as $\gamma \searrow 0$, and meanwhile is nevertheless a probability density for every $\gamma > 0$,[8] the inner expectation over $\mathbf{x} \mid \mathbf{y}$ leads to simply replacing the integrand with its value at $\mathbf{x} = f(\mathbf{y})$; the integrand is of course the conditional density of $\mathbf{x}$ given $\mathbf{y}$, and from here it is straightforward to argue that this leads the optimal $f$ to be (almost surely) the conditional maximum a posteriori (MAP) estimate, under our regularity assumptions on $p(\mathbf{x})$.

To make this intuitive argument rigorous, we need to translate our regularity assumptions on $p(\mathbf{x})$ into regularity of $p_{\mathbf{x}|\mathbf{y}}$, interchange the $\gamma$ limit in (C.2) with the expectation over $\mathbf{y}$, and instantiate a rigorous analogue of the heuristic "concentration" argument. First, we have by Bayes' rule and Gaussian conditioning

$$p_{\mathbf{x}|\mathbf{y}}(\mathbf{x}) = \frac{\varphi_{\sigma^2}(\mathbf{y} - \mathbf{x})p(\mathbf{x})}{(\varphi_{\sigma^2} * p)(\mathbf{y})},$$

---

[8] For readers familiar with signal processing or Schwartz's theory of distributions, this could be alternately stated as "the small-variance limit of the Gaussian density behaves like a Dirac delta distribution".

where $*$ denotes convolution of densities; the denominator is the density of $\mathbf{y}$, and it satisfies $\varphi_{\sigma^2} * p > 0$ since $\varphi_{\sigma^2} > 0$. In particular, this implies that $p_{\mathbf{x}|\mathbf{y}}$ is a continuous function of $(\mathbf{x}, \mathbf{y})$, because $p(\mathbf{x})$ is continuous by assumption. We can then write, by the definition of convolution,

$$\mathbb{E}_{\mathbf{x}|\mathbf{y}} \left[ \varphi_{\gamma^2/2}(f(\mathbf{y}) - \mathbf{x}) \right] = \varphi_{\gamma^2/2} * p_{\mathbf{x}|\mathbf{y}}(f(\mathbf{y})),$$

so following (C.2), we have

$$\lim_{\gamma \searrow 0} \mathbb{E}_{\mathbf{x},\mathbf{y}} \left[ \varphi_{\gamma^2/2}(f(\mathbf{y}) - \mathbf{x}) \right] = \lim_{\gamma \searrow 0} \mathbb{E}_{\mathbf{y}} \left[ \varphi_{\gamma^2/2} * p_{\mathbf{x}|\mathbf{y}}(f(\mathbf{y})) \right]. \tag{C.3}$$

We are going to argue that the limit can be moved inside the expectation in (C.3) momentarily; for the moment, we consider the quantity that results after moving the limit inside the expectation. To treat this term, we apply a standard approximation to the identity argument to evaluate the limit of the preceding expression. (Stein & Shakarchi, 2005, Ch. 3, Example 3) implies that the densities $\varphi_{\gamma^2/2}$ constitute an approximation to the identity as $\gamma \to 0$, and because $p_{\mathbf{x}|\mathbf{y}}$ is continuous, we can then apply (Stein & Shakarchi, 2005, Ch. 3, Theorem 2.1) to obtain that

$$\lim_{\gamma \searrow 0} \varphi_{\gamma^2/2} * p_{\mathbf{x}|\mathbf{y}}(f(\mathbf{y})) = p_{\mathbf{x}|\mathbf{y}}(f(\mathbf{y})).$$

In particular, after justifying the interchange of limit and expectation in (C.3), we will have shown, by following our manipulations from (C.1), that

$$\lim_{\gamma \searrow 0} \mathbb{E}_{\mathbf{x},\mathbf{y}} \left[ m_\gamma \left( \|f(\mathbf{y}) - \mathbf{x}\|_2 \right) \right] = 1 - \mathbb{E}_{\mathbf{y}} \left[ p_{\mathbf{x}|\mathbf{y}}(f(\mathbf{y})) \right]. \tag{C.4}$$

We will proceed to conclude the proof from this expression, and justify the limit-expectation interchange at the end of the proof. The problem at hand is equivalent to the problem

$$\underset{f \text{ measurable}}{\mathrm{argmax}} \ \mathbb{E}_{\mathbf{y}} \left[ p_{\mathbf{x}|\mathbf{y}}(f(\mathbf{y})) \right].$$

Writing the expectation as an integral, we have by Bayes' rule as above

$$\mathbb{E}_{\mathbf{y}} \left[ p_{\mathbf{x}|\mathbf{y}}(f(\mathbf{y})) \right] = \int_{\mathbb{R}^n} \varphi_{\sigma^2}(\mathbf{y} - f(\mathbf{y})) p(f(\mathbf{y})) d\mathbf{y}.$$

Let us define an auxiliary function $g : \mathbb{R}^n \times \mathbb{R}^n \to \mathbb{R}$ by $g(\mathbf{x}, \mathbf{y}) = \varphi_{\sigma^2}(\mathbf{y} - \mathbf{x}) p(\mathbf{x})$. Then

$$\mathbb{E}_{\mathbf{y}} \left[ p_{\mathbf{x}|\mathbf{y}}(f(\mathbf{y})) \right] = \int_{\mathbb{R}^n} g(f(\mathbf{y}), \mathbf{y}) d\mathbf{y},$$

and moreover, for every $\mathbf{y}$, $g(\cdot, \mathbf{y})$ is continuous and compactly supported, by continuity and boundedness of the Gaussian density and the assumption that $p(\mathbf{x})$ is continuous and the random variable $\mathbf{x} \sim p(\mathbf{x})$ is bounded. We have for any measurable $f$

$$g(f(\mathbf{y}), \mathbf{y}) \leq \max_{\mathbf{x} \in \mathbb{R}^n} g(\mathbf{x}, \mathbf{y}). \tag{C.5}$$

Our aim is thus to argue that there is a choice of measurable $f$ such that the preceding bound can be made tight; this will imply that any measurable $f$ maximizing the objective $\mathbb{E}_{\mathbf{y}}[p_{\mathbf{x}|\mathbf{y}}(f(\mathbf{y}))]$ satisfies $g(f(\mathbf{y}), \mathbf{y}) = \max_{\mathbf{x} \in \mathbb{R}^n} g(\mathbf{x}, \mathbf{y})$ almost surely, or equivalently that $f(\mathbf{y}) \in \mathrm{argmax}_{\mathbf{x} \in \mathbb{R}^n} g(\mathbf{x}, \mathbf{y})$ almost surely. The claim will then follow, because $\mathrm{argmax}_{\mathbf{x} \in \mathbb{R}^n} g(\mathbf{x}, \mathbf{y}) = \mathrm{argmax}_{\mathbf{x} \in \mathbb{R}^n} p_{\mathbf{x}|\mathbf{y}}(\mathbf{x})$.

To this end, define $h(\mathbf{y}) = \max_{\mathbf{x} \in \mathbb{R}^n} g(\mathbf{x}, \mathbf{y})$. Then by the Weierstrass theorem, $h$ is finite-valued, and for every $\mathbf{y}$ there exists some $\mathbf{c} \in \mathbb{R}^n$ such that $h(\mathbf{y}) = g(\mathbf{c}, \mathbf{y})$. Because $g$ is continuous, it then follows from Rockafellar & Wets (1998, Theorem 1.17(c)) that $h$ is continuous. Moreover, because $g$ is continuous and for every $\mathbf{y}$, $g(\cdot, \mathbf{y})$ is compactly supported, $g$ is in particular level-bounded in $\mathbf{x}$ locally uniformly in $\mathbf{y}$ in the sense of Rockafellar & Wets (1998, Definition 1.16), and it follows that the set-valued mapping $\mathbf{y} \mapsto \mathrm{argmax}_{\mathbf{x}} g(\mathbf{x}, \mathbf{y}) : \mathbb{R}^n \rightrightarrows \mathbb{R}^n$ is compact-valued, by the Weierstrass theorem, and outer semicontinuous relative to $\mathbb{R}^n$, by Rockafellar & Wets (1998, Example 5.22). Applying Rockafellar & Wets (1998, Exercise 14.9, Corollary 14.6), we conclude that the set-valued mapping $\mathbf{y} \mapsto \mathrm{argmax}_{\mathbf{x}} g(\mathbf{x}, \mathbf{y})$ is measurable, and that in particular there exists a measurable function $f^* : \mathbb{R}^n \to \mathbb{R}^n$ such that $f^*(\mathbf{y}) \in \mathrm{argmax}_{\mathbf{x}} g(\mathbf{x}, \mathbf{y})$ for every $\mathbf{y} \in \mathbb{R}^n$. Thus, there is a measurable $f$ attaining the bound in (C.5), and the claim follows after we can justify the preceding interchange of limit and expectation.

To justify the interchange of limit and expectation, we will apply the dominated convergence theorem, which requires us to show an integrable (with respect to the density of $\mathbf{y}$) upper bound for the function $\mathbf{y} \mapsto \mathbb{E}_{\mathbf{x}|\mathbf{y}}[\varphi_{\gamma^2/2}(f(\mathbf{y}) - \mathbf{x})]$. For this, we calculate

$$
\begin{aligned}
\mathbb{E}_{\mathbf{x}|\mathbf{y}}\left[\varphi_{\gamma^2/2}(f(\mathbf{y}) - \mathbf{x})\right] &= \frac{1}{(\varphi_{\sigma^2} * p)(\mathbf{y})} \int_{\mathbb{R}^n} \varphi_{\sigma^2}(\mathbf{y} - \mathbf{x}) p(\mathbf{x}) \varphi_{\gamma^2/2}(f(\mathbf{y}) - \mathbf{x}) d\mathbf{x} \\
&\leq \frac{1}{(\varphi_{\sigma^2} * p)(\mathbf{y})} \left[\sup_{\mathbf{x}} \varphi_{\sigma^2}(\mathbf{y} - \mathbf{x}) p(\mathbf{x})\right] \int_{\mathbb{R}^n} \varphi_{\gamma^2/2}(f(\mathbf{y}) - \mathbf{x}) d\mathbf{x} \\
&= \frac{1}{(\varphi_{\sigma^2} * p)(\mathbf{y})} \left[\sup_{\mathbf{x}} \varphi_{\sigma^2}(\mathbf{y} - \mathbf{x}) p(\mathbf{x})\right],
\end{aligned}
$$

by Hölder's inequality and the fact that $\varphi_{\gamma^2/2}$ is a probability density. Because the random variable $\mathbf{x} \sim p(\mathbf{x})$ is assumed bounded, the density $p(\mathbf{x})$ has compact support, and the density $p(\mathbf{x})$ is assumed continuous, so there exists $R > 0$ such that if $\|\mathbf{x}\|_2 > R$ then $p(\mathbf{x}) = 0$, and $M > 0$ such that $p(\mathbf{x}) \leq M$. We then have

$$
\sup_{\mathbf{x}} \varphi_{\sigma^2}(\mathbf{y} - \mathbf{x}) p(\mathbf{x}) \leq M \sup_{\mathbf{x}} \varphi_{\sigma^2}(\mathbf{y} - \mathbf{x}) \mathbb{1}_{\|\mathbf{x}\|_2 \leq R}.
$$

This means that the supremum can attain a nonzero value only on points where $\|\mathbf{x}\|_2 \leq R$. On the other hand, for every $\mathbf{y}$ with $\|\mathbf{y}\|_2 \geq 2R$, whenever $\|\mathbf{x}\|_2 \leq R$ the triangle inequality implies $\|\mathbf{y} - \mathbf{x}\|_2 \geq \|\mathbf{y}\|_2 - \|\mathbf{x}\|_2 \geq \frac{1}{2}\|\mathbf{y}\|_2$. Because the Gaussian density $\varphi_{\sigma^2}$ is a radial function, we conclude that if $\|\mathbf{y}\|_2 \geq 2R$, one has

$$
\sup_{\mathbf{x}} \varphi_{\sigma^2}(\mathbf{y} - \mathbf{x}) p(\mathbf{x}) \leq M \varphi_{\sigma^2}(\mathbf{y}/2) = CM \varphi_{4\sigma^2}(\mathbf{y}),
$$

where $C > 0$ depends only on $n$. At the same time, we always have

$$
\sup_{\mathbf{x}} \varphi_{\sigma^2}(\mathbf{y} - \mathbf{x}) p(\mathbf{x}) \leq \frac{M}{(2\pi\sigma^2)^{n/2}}.
$$

Consequently, we have the composite upper bound

$$
\sup_{\mathbf{x}} \varphi_{\sigma^2}(\mathbf{y} - \mathbf{x}) p(\mathbf{x}) \leq \begin{cases} \frac{M}{(2\pi\sigma^2)^{n/2}} & \|\mathbf{y}\|_2 < 2R \\ 2M\varphi_{4\sigma^2}(\mathbf{y}) & \|\mathbf{y}\|_2 \geq 2R, \end{cases}
$$

and by our work above

$$
\mathbb{E}_{\mathbf{x}|\mathbf{y}}\left[\varphi_{\gamma^2/2}(f(\mathbf{y}) - \mathbf{x})\right] \leq \frac{1}{(\varphi_{\sigma^2} * p)(\mathbf{y})} \times \begin{cases} \frac{M}{(2\pi\sigma^2)^{n/2}} & \|\mathbf{y}\|_2 < 2R \\ 2M\varphi_{4\sigma^2}(\mathbf{y}) & \|\mathbf{y}\|_2 \geq 2R. \end{cases}
$$

Because $\varphi_{\sigma^2} * p$ is the density of $\mathbf{y}$, this upper bound is sufficient to apply the dominated convergence theorem to obtain

$$
\lim_{\gamma \searrow 0} \mathbb{E}_{\mathbf{x},\mathbf{y}}\left[\varphi_{\gamma^2/2}(f(\mathbf{y}) - \mathbf{x})\right] = \mathbb{E}_{\mathbf{y}} \lim_{\gamma \searrow 0} \mathbb{E}_{\mathbf{x}|\mathbf{y}}\left[\varphi_{\gamma^2/2}(f(\mathbf{y}) - \mathbf{x})\right].
$$

Combining this assertion with the argument surrounding (C.4), we conclude the proof.

$\square$

*Remark* (Other loss choices). Theorem 3.2 also holds for any $m_\gamma$ such that $m_\gamma$ is uniformly (in $\gamma$) bounded above, for each $\gamma > 0$ uniquely minimized at 0, and $\sup_{x \in \mathbb{R}} m_\gamma(x) - m_\gamma(\|\mathbf{x}\|_2)$ is an approximation to the identity as $\gamma \searrow 0$ (see (Stein & Shakarchi, 2005, Ch. 3, §2)).

## C.3 PROOF OF THEOREM B.1

*Proof.* For brevity, we denote $\operatorname{argmax}_{\mathbf{c}} P(\mathbf{x} = \mathbf{c} \mid \mathbf{y})$ by $\mathrm{MAP}[\mathbf{x} \mid \mathbf{y}]$, i.e., the maximum a posteriori estimate of $\mathbf{x}$ given $\mathbf{y}$.

First, we show that $\mathrm{MAP}[\mathbf{x} \mid \mathbf{y}]$ is unique for almost all $\mathbf{y}$.

Consider $\mathbf{y}$ such that $\mathrm{MAP}[\mathbf{x} \mid \mathbf{y}]$ is not unique. There exists $i \neq j$, such that

$$P(\mathbf{x}_i \mid \mathbf{y}) = P(\mathbf{x}_j \mid \mathbf{y})$$
$$\iff p(\mathbf{y} \mid \mathbf{x}_i)P(\mathbf{x}_i) = p(\mathbf{y} \mid \mathbf{x}_j)P(\mathbf{x}_j)$$
$$\iff -\frac{1}{2}\|\mathbf{y} - \mathbf{x}_i\|^2 + \sigma^2 \log P(\mathbf{x}_i) = -\frac{1}{2}\|\mathbf{y} - \mathbf{x}_j\|^2 + \sigma^2 \log P(\mathbf{x}_j)$$
$$\iff \langle \mathbf{y}, \frac{\mathbf{x}_i - \mathbf{x}_j}{2} \rangle = \frac{1}{2}\|\mathbf{x}_i\|^2 - \frac{1}{2}\|\mathbf{x}_j\|^2 - \sigma^2 \log P(\mathbf{x}_i) + \sigma^2 \log P(\mathbf{x}_j).$$

i.e., $\mathbf{y}$ lies in a hyperplane defined by $\mathbf{x}_i, \mathbf{x}_j$ (note that $\mathbf{x}_i \neq \mathbf{x}_j$). Denote the hyperplane by

$$\mathcal{H}_{i,j} := \left\{ \mathbf{y} \mid \langle \mathbf{y}, \frac{\mathbf{x}_i - \mathbf{x}_j}{2} \rangle = \frac{1}{2}\|\mathbf{x}_i\|^2 - \frac{1}{2}\|\mathbf{x}_j\|^2 - \sigma^2 \log P(\mathbf{x}_i) + \sigma^2 \log P(\mathbf{x}_j) \right\}.$$

Consider

$$\mathcal{U} := \cup_{i \neq j} \mathcal{H}_{i,j}.$$

We have that $\forall \mathbf{y}$ with non-unique $\mathrm{MAP}[\mathbf{x} \mid \mathbf{y}]$,

$$\exists i \neq j, \mathbf{y} \in \mathcal{H}_{i,j}$$
$$\iff \mathbf{y} \in \mathcal{U}.$$

Note that $\mathcal{U}$ has zero measure as a countable union of zero-measure sets, hence the measure of all $\mathbf{y}$ with non-unique $\mathrm{MAP}[\mathbf{x} \mid \mathbf{y}]$ is zero. Hence, for almost all $\mathbf{y}$, $\mathrm{MAP}[\mathbf{x} \mid \mathbf{y}]$ is unique.

Next, we show that for almost all $\mathbf{y}$,

$$f^*(\mathbf{y}) = \underset{\mathbf{c}}{\mathrm{argmin}}\, \mathbb{E}_{\mathbf{x} \mid \mathbf{y}}[\mathbb{1}_{\mathbf{c} \neq \mathbf{x}}].$$

Note that

$$\lim_{\gamma \searrow 0} \mathbb{E}_{\mathbf{x}, \mathbf{y}} \left[ m_\gamma \left( \|f(\mathbf{y}) - \mathbf{x}\|_2 \right) \right]$$
$$= \mathbb{E}_{\mathbf{x}, \mathbf{y}} \left[ \lim_{\gamma \searrow 0} m_\gamma \left( \|f(\mathbf{y}) - \mathbf{x}\|_2 \right) \right]$$
$$= \mathbb{E}_{\mathbf{x}, \mathbf{y}} \left[ \mathbb{1}_{\|f(\mathbf{y}) - \mathbf{x}\|_2 \neq 0} \right]$$
$$= \mathbb{E}_{\mathbf{x}, \mathbf{y}} \left[ \mathbb{1}_{f(\mathbf{y}) \neq \mathbf{x}} \right].$$

Above, the first equality uses the monotone convergence theorem. Use the law of iterated expectations,

$$\mathbb{E}_{\mathbf{x}, \mathbf{y}} \left[ \mathbb{1}_{f(\mathbf{y}) \neq \mathbf{x}} \right] = \mathbb{E}_{\mathbf{y}} \mathbb{E}_{\mathbf{x} \mid \mathbf{y}} \left[ \mathbb{1}_{f(\mathbf{y}) \neq \mathbf{x}} \right].$$

We will use this expression to study the global minimizers of the objective. By conditioning,

$$\mathbb{E}_{\mathbf{x} \mid \mathbf{y}} \left[ \mathbb{1}_{f(\mathbf{y}) \neq \mathbf{x}} \right] \geq \min_{\mathbf{c}} \mathbb{E}_{\mathbf{x} \mid \mathbf{y}}[\mathbb{1}_{\mathbf{c} \neq \mathbf{x}}],$$

and so

$$\mathbb{E}_{\mathbf{y}} \left[ \mathbb{E}_{\mathbf{x} \mid \mathbf{y}} \left[ \mathbb{1}_{f(\mathbf{y}) \neq \mathbf{x}} \right] - \min_{\mathbf{c}} \mathbb{E}_{\mathbf{x} \mid \mathbf{y}}[\mathbb{1}_{\mathbf{c} \neq \mathbf{x}}] \right] \geq 0.$$

Because $p(\mathbf{y}) > 0$, it follows that every global minimizer of the objective $f^*$ satisfies

$$\mathbb{E}_{\mathbf{x} \mid \mathbf{y}} \left[ \mathbb{1}_{f^*(\mathbf{y}) \neq \mathbf{x}} \right] = \min_{\mathbf{c}} \mathbb{E}_{\mathbf{x} \mid \mathbf{y}}[\mathbb{1}_{\mathbf{c} \neq \mathbf{x}}] \quad \text{a.s.}$$

Hence, for almost all $\mathbf{y}$,

$$f^*(\mathbf{y}) \in \underset{\mathbf{c}}{\mathrm{argmin}}\, \mathbb{E}_{\mathbf{x} \mid \mathbf{y}}[\mathbb{1}_{\mathbf{c} \neq \mathbf{x}}].$$

Finally, we show that $\mathrm{argmin}_{\mathbf{c}}\, \mathbb{E}_{\mathbf{x} \mid \mathbf{y}}[\mathbb{1}_{\mathbf{c} \neq \mathbf{x}}] = \mathrm{MAP}[\mathbf{x} \mid \mathbf{y}]$. The claim then follows from our preceding work showing that $\mathrm{MAP}[\mathbf{x} \mid \mathbf{y}]$ is almost surely unique. Consider

$$\mathbb{E}_{\mathbf{x} \mid \mathbf{y}}[\mathbb{1}_{\mathbf{c} \neq \mathbf{x}}] = \sum_i P(\mathbf{x}_i \mid \mathbf{y})\mathbb{1}_{\mathbf{c} \neq \mathbf{x}_i}$$
$$= \sum_i P(\mathbf{x}_i \mid \mathbf{y})(1 - \mathbb{1}_{\mathbf{c} = \mathbf{x}_i})$$
$$= \sum_i P(\mathbf{x}_i \mid \mathbf{y}) - \sum_{\mathbf{x}_i = \mathbf{c}} P(\mathbf{x}_i \mid \mathbf{y})$$
$$= 1 - P(\mathbf{x} = \mathbf{c} \mid \mathbf{y}).$$

Hence,

$$\operatorname*{argmin}_{\mathbf{c}} \mathbb{E}_{\mathbf{x}|\mathbf{y}}[\mathbb{1}_{\mathbf{c} \neq \mathbf{x}}] = \operatorname*{argmax}_{\mathbf{c}} P(\mathbf{x} = \mathbf{c} \mid \mathbf{y})$$
$$= \mathrm{MAP}[\mathbf{x} \mid \mathbf{y}].$$

$\square$

### C.4 PROOFS OF PnP OPTIMIZATION RESULTS

In this section, we restate and provide proofs of Theorem B.2 and Theorem 4.1. We prove Theorem B.2 under slightly more general assumptions, and state the conclusions of both Theorems 4.1 and B.2 with more precision. The restated results are given below, as Theorem C.1 and Theorem C.4.

Before proceeding to proofs, let us briefly describe the common high-level 'recipe' underlying each plug-and-play algorithm's proof. The recipe separates into two distinct steps:

1. **Leverage general, black-box convergence analyses from the optimization literature.** A plug-and-play algorithm is derived from a 'baseline' optimization algorithm; we therefore appeal to convergence analyses from the literature of the relevant baseline algorithm. Because the regularization function associated to a LPN is implicitly defined by the LPN architecture and need not be convex, it is necessary to appeal to general, 'black-box' convergence analyses which do not leverage special properties of the regularization function. We make use of convergence results on nonconvex proximal gradient descent of Boţ et al. (2016),[9] and on nonconvex ADMM of Themelis & Patrinos (2020). To make the presentation self-contained, we reproduce key results from these works in context. The principal technical activity is therefore to translate the iterate sequence generated by the relevant PnP algorithm into a form that allows these convergence analyses to be applied to it. Echoes of the same approach appear in prior work on convergent plug-and-play, for example work of Hurault et al. (2022b).

2. **Establish general regularity properties of the regularization function associated to LPNs.** To appeal to the aforementioned convergence analyses, it is necessary to ascertain a minimum level of regularity of the regularization function associated to an LPN, in order to establish that it possesses the Kurdyka-Łojasiewicz (KL) property (and, say, coercivity). We give a self-contained overview of the KL property and how we establish it in Appendix C.4.4 for clarity of presentation. We provide in Appendix C.4.3 technical lemmas that establish that LPNs of the architecture specified in Proposition 3.1 satisfy these properties, *regardless of the exact values of their parameters*. These results are essentially consequences of differentiability and surjectivity of the LPN when $0 < \alpha < 1$ is used as the strong convexity weight, and they enable us to assert convergence guarantees for LPNs *without any extra assumptions about the trained network*.

We anticipate that this recipe will be applicable to virtually any PnP scheme for which there exists a convergence analysis under the KL property of the corresponding baseline optimization algorithm. Because our technical work in Appendix C.4.3 establishes the KL property and coercivity for the regularization function associated to LPNs with the architecture of Proposition 3.1, obtaining a convergence analysis for such a PnP scheme with LPNs of this architecture only requires the first step of the above recipe. We expect our approach in Appendix C.4.3 to extend straightforwardly to LPNs with novel architectures—for instance, different computational graphs or weight-sharing schemes—as long as the nonlinear activation functions do not grow too rapidly (see the proofs for more precise statements).

### C.4.1 PROOF OF THEOREM B.2 (PnP-PGD)

**Theorem C.1** (Convergence guarantee for running PnP-PGD with LPNs)**.** *Consider the sequence of iterates $\mathbf{x}_k$, $k \in \{0, 1, \dots\}$, defined by Algorithm 4 run with a continuously differentiable measurement operator $A$ and an LPN $f_\theta$ with softplus activations, trained with $0 < \alpha < 1$. Assume*

---

[9] The form these results are stated in makes them most convenient for purposes of our presentation, although the result we need is originally due to (Attouch et al., 2013).

further that the data fidelity term $h(\mathbf{x}) = \frac{1}{2}\|\mathbf{y} - A(\mathbf{x})\|_2^2$ is definable in the o-minimal structure of Proposition C.11, Property 2[10] and has L-Lipschitz gradient[11], and that the step size satisfies $0 < \eta < 1/L$. Then, the iterates $\mathbf{x}_k$ converge to a fixed point $\mathbf{x}^*$ of Algorithm 4: that is, there exist $\mathbf{x}^* \in \mathbb{R}^n$ such that

$$f_\theta\left(\mathbf{x}^* - \eta\nabla h(\mathbf{x}^*)\right) = \mathbf{x}^*, \tag{C.6}$$

and $\lim_{k\to\infty} \mathbf{x}_k = \mathbf{x}^*$. Furthermore, $\mathbf{x}^*$ is a critical point[12] of $h + \frac{1}{\eta}R_\theta$, where $R_\theta$ is the regularization function associated to the LPN $f_\theta$ (i.e., $f_\theta = \mathrm{prox}_{R_\theta}$).

Before proceeding to the proof, we state a few settings and results from Boţ et al. (2016) that are useful for proving Theorem C.1, for better readability.

**Problem 1** ((Boţ et al., 2016, Problem 1)). *Let $f : \mathbb{R}^m \to (-\infty, +\infty]$ be a proper, lower semicontinuous function which is bounded below and let $h : \mathbb{R}^m \to \mathbb{R}$ be a Fréchet differentiable function with Lipschitz continuous gradient, i.e. there exists $L_{\nabla h} \geq 0$ such that $\|\nabla h(\mathbf{x}) - \nabla h(\mathbf{x}')\| \leq L_{\nabla h}\|\mathbf{x} - \mathbf{x}'\|$ for all $\mathbf{x}, \mathbf{x}' \in \mathbb{R}^m$. Consider the optimization problem*

$$(P) \quad \inf_{\mathbf{x}\in\mathbb{R}^m}[f(\mathbf{x}) + h(\mathbf{x})].$$

**Algorithm C.1** ((Boţ et al., 2016, Algorithm 1)). *Choose $\mathbf{x}_0, \mathbf{x}_1 \in \mathbb{R}^m, \underline{\alpha}, \overline{\alpha} > 0, \beta \geq 0$ and the sequences $(\alpha_n)_{n\geq 1}, (\beta_n)_{n\geq 1}$ fulfilling*

$$0 < \underline{\alpha} \leq \alpha_n \leq \overline{\alpha} \ \forall n \geq 1$$

*and*

$$0 \leq \beta_n \leq \beta \ \forall n \geq 1.$$

*Consider the iterative scheme*

$$(\forall n \geq 1) \ \mathbf{x}_{n+1} \in \underset{\mathbf{U}\in\mathbb{R}^m}{\mathrm{argmin}}\{D_F(\mathbf{U}, \mathbf{x}_n) + \alpha_n\langle\mathbf{U}, \nabla h(\mathbf{x}_n)\rangle + \beta_n\langle\mathbf{U}, \mathbf{x}_{n-1} - \mathbf{x}_n\rangle + \alpha_n f(\mathbf{U})\}. \tag{C.7}$$

*Here, $F : \mathbb{R}^m \to \mathbb{R}$ is $\sigma$-strongly convex, Fréchet differentiable and $\nabla F$ is $L_{\nabla F}$-Lipschitz continuous, with $\sigma, L_{\nabla F} > 0$; $D_F$ is the Bregman distance to $F$.*

**Theorem C.2** ((Boţ et al., 2016, Theorem 13)). *In the setting of Problem 1, choose $\underline{\alpha}, \overline{\alpha}, \beta$ satisfying*

$$\sigma > \overline{\alpha}L_{\nabla_h} + 2\beta\frac{\overline{\alpha}}{\underline{\alpha}}. \tag{C.8}$$

*Assume that $f + h$ is coercive and that*

$$H : \mathbb{R}^m \times \mathbb{R}^m \to (-\infty, +\infty], \ H(\mathbf{x}, \mathbf{x}') = (f + h)(\mathbf{x}) + \frac{\beta}{2\underline{\alpha}}\|\mathbf{x} - \mathbf{x}'\|^2, \ \forall(\mathbf{x}, \mathbf{x}') \in \mathbb{R}^m \times \mathbb{R}^m$$

*is a KL function[13]. Let $(\mathbf{x}_n)_{n\in\mathbb{N}}$ be a sequence generated by Algorithm C.1. Then the following statements are true:*

1. *$\sum_{n\in\mathbb{N}}\|\mathbf{x}_{n+1} - \mathbf{x}_n\| < +\infty$*

2. *there exists $\mathbf{x} \in \mathrm{crit}(f + h)$ such that $\lim_{n\to+\infty}\mathbf{x}_n = \mathbf{x}$.*

Now, we prove Theorem C.1.

---

[10]This mild technical assumption is satisfied by an extremely broad array of nonlinear operators $A$: for example, any $A$ which is a polynomial in the input $\mathbf{x}$ (in particular, linear $A$) is definable, and compositions and inverses of definable functions are definable, so that definability of $A$ implies definability of $h$. See an extensive overview of these ideas in Appendix C.4.4

[11]This is a very mild assumption. For example, when $A$ is linear, the gradient of the data fidelity term $\nabla h$ has a Lipschitz constant no larger than $\|A^*A\|$, where $\|\cdot\|$ denotes the operator norm of a linear operator and $A^*$ is the adjoint of $A$.

[12]In this work, the set of critical points of a function $f$ is defined by $\mathrm{crit}(f) := \{\mathbf{x} : 0 \in \partial f(\mathbf{x})\}$, where $\partial f$ is the limiting (Mordukhovich) Fréchet subdifferential mapping of $f$ (see definition in (Boţ et al., 2016, Section 2)).

[13]In this work, a function being KL means it satisfies the Kurdyka-Łojasiewicz property (Lojasiewicz, 1963), see Appendix C.4.4, Definition C.1.

*Proof of Theorem C.1.* By Lemma C.7, there is a coercive function $R_\theta : \mathbb{R}^n \to \mathbb{R} \cup \{+\infty\}$ such that $f_\theta = \mathrm{prox}_{R_\theta}$. The idea of the proof is to apply Theorem C.2 to our setting; this requires us to check that Algorithm 4 maps onto Algorithm C.1, and that our (implicitly-defined) objective function and parameter choices satisfy the requirements of this theorem. To this end, note that the application of $f_\theta$ in Algorithm 4 can be written as

$$
\begin{aligned}
\mathbf{x}_{k+1} &= f_\theta \left( \mathbf{x}_k - \eta \nabla h(\mathbf{x}_k) \right) \\
&= \underset{\mathbf{x}' \in \mathbb{R}^n}{\mathrm{argmin}} \; \frac{1}{2} \left\| \mathbf{x}' - (\mathbf{x}_k - \eta \nabla h(\mathbf{x}_k)) \right\|_2^2 + R_\theta(\mathbf{x}') \\
&= \underset{\mathbf{x}' \in \mathbb{R}^n}{\mathrm{argmin}} \; \frac{1}{2} \left\| \mathbf{x}' - \mathbf{x}_k \right\|_2^2 + \langle \mathbf{x}' - \mathbf{x}_k, \eta \nabla h(\mathbf{x}_k) \rangle + R_\theta(\mathbf{x}') \\
&= \underset{\mathbf{x}' \in \mathbb{R}^n}{\mathrm{argmin}} \; \frac{1}{2} \left\| \mathbf{x}' - \mathbf{x}_k \right\|_2^2 + \eta \langle \mathbf{x}', \nabla h(\mathbf{x}_k) \rangle + \eta \cdot \frac{1}{\eta} R_\theta(\mathbf{x}')
\end{aligned}
$$

showing that Algorithm 4 corresponds to Algorithm C.1 with the Bregman distance $D_F(\mathbf{x}, \mathbf{y}) = \frac{1}{2} \|\mathbf{x} - \mathbf{y}\|_2^2$ (and correspondingly $F(\mathbf{x}) = \frac{1}{2} \|\mathbf{x}\|_2^2$, which satisfies $\sigma = L_{\nabla F} = 1$), the momentum parameter $\beta = \beta_n = 0$, the step size $\alpha_n = \overline{\alpha} = \underline{\alpha} = \eta$, and $f = \frac{1}{\eta} R_\theta$. In the framework of Boţ et al. (2016), Algorithm 4 minimizes the implicitly-defined objective $h + \eta^{-1} R_\theta$. Moreover, one checks that our choice of constant step size $0 < \eta < 1/L$ verifies the necessary condition (C.8), and because $h \geq 0$, coercivity of $R_\theta$ implies that $h + \eta^{-1} R_\theta$ is coercive. Using Lemma C.13, we obtain that $R_\theta$ is definable, and by assumption, $h$ is also definable, so that by Proposition C.11, Properties 2 and 5, it follows that the objective $h + \eta^{-1} R_\theta$ is definable. Thus $h + \eta^{-1} R_\theta$ is definable, continuously differentiable (by Lemma C.7), and proper (as a sum of real-valued functions, again by Lemma C.7), and therefore has the KL property, by Proposition C.11, Property 1. We can therefore apply Theorem C.2 to conclude convergence to a critical point of $h + \eta^{-1} R_\theta$. Finally, by Lemma C.3 and the continuity of $f_\theta$ and $\nabla h$, we conclude convergence to a fixed point, $\mathbf{x} = f_\theta(\mathbf{x} - \eta \nabla h(\mathbf{x}))$, which is identical to (C.6). $\qquad\square$

**Lemma C.3** (Convergence Implies Fixed Point Convergence). *Suppose $\mathcal{F} : \mathbb{R}^n \to \mathbb{R}^n$ is a continuous map that defines an iterative process, $\mathbf{x}_{k+1} = \mathcal{F}(\mathbf{x}_k)$. Assume $\mathbf{x}_k$ converges, i.e., $\exists \, \mathbf{x}^*$ such that $\lim_{k \to \infty} \mathbf{x}_k = \mathbf{x}^*$. Then, $\mathbf{x}^*$ is a fixed point of $\mathcal{F}$, i.e., $\mathbf{x}^* = \mathcal{F}(\mathbf{x}^*)$.*

*Proof.*

$$
\mathbf{x}^* = \lim_{k \to \infty} \mathbf{x}_k = \lim_{k \to \infty} \mathbf{x}_{k+1} = \lim_{k \to \infty} \mathcal{F}(\mathbf{x}_k) = \mathcal{F}\left( \lim_{k \to \infty} \mathbf{x}_k \right) = \mathcal{F}(\mathbf{x}^*).
$$

The fourth equality follows from continuity of $\mathcal{F}$. $\qquad\square$

### C.4.2 PROOF OF THEOREM 4.1 (PNP-ADMM)

We present in this section a proof of convergence for PnP-ADMM schemes which incorporate an LPN for the regularizer (Algorithm 1), following the recipe we have described in Appendix C.4. These guarantees are analogous to those we have proved in Theorem C.1 for the PnP-PGD scheme Algorithm 4 with LPNs. For simplicity, we will assume in this section (in contrast to the more general setting of Theorem C.1, and in agreement with the result stated in Theorem 4.1) that the measurement operator $A$ in the underlying inverse problem (2.1) is linear and acts in the standard basis, and accordingly we identify it with its matrix representation $\mathbf{A}$. This means (among other things) that the data fidelity term $\mathbf{x} \mapsto \frac{1}{2} \|\mathbf{y} - \mathbf{A}\mathbf{x}\|_2^2$ is convex.

Before proceeding to the proof, we note that Algorithm 1 adopts an update ordering which is non-standard in the signal processing literature (c.f. (Venkatakrishnan et al., 2013; Kamilov et al., 2023a)) for technical reasons. We employ this update order due to its prevalence in the optimization literature, notably in the analysis of Themelis & Patrinos (2020), and we emphasize that all of our experiments are done following Algorithm 1. Although both the typical PnP-ADMM update order and the update order in Algorithm 1 correspond to an ADMM algorithm for the *same objective function*, the analysis of these two iterative optimization procedures is different, and seems to require different technical assumptions (c.f. (Themelis & Patrinos, 2020; Yan & Yin, 2016)). Prior work on convergent PnP seems to have also run into this barrier, suggesting it is not an artifact of

our analysis: for example, Hurault et al. (2022b) study a PnP variant of Douglas-Rachford splitting rather than ADMM, which is roughly analogous to the reversed-order of updates in Algorithm 1 by a reduction of Themelis & Patrinos (2020), and Sun et al. (2021) prove convergence of a sequence of associated residuals in the standard-order PnP-ADMM rather than of the sequence of iterates itself.

Our proof will be based on the work of Themelis & Patrinos (2020), which provides guarantees for ADMM in the nonconvex setting. We restate some of their results for convenience after stating our convergence result, then proceed to the proof.

**Theorem C.4** (Convergence guarantee for running PnP-ADMM with LPNs)**.** *Consider the sequence of iterates* $(\mathbf{x}_k, \mathbf{u}_k, \mathbf{z}_k)$, $k \in \{0, 1, \dots\}$, *defined by Algorithm 1 run with a linear measurement operator* $\mathbf{A}$ *and a LPN* $f_\theta$ *with softplus activations, trained with* $0 < \alpha < 1$. *Assume further that the penalty parameter* $\rho$ *satisfies* $\rho > \|\mathbf{A}^T \mathbf{A}\|$. *Then the sequence of iterates* $(\mathbf{x}_k, \mathbf{u}_k, \mathbf{z}_k)$ *converges to a limit point* $(\mathbf{x}^*, \mathbf{u}^*, \mathbf{z}^*)$ *which satisfies the KKT conditions (of the augmented problem):*

$$\mathbf{x}^* = \mathbf{z}^*,$$
$$\mathbf{u}^* = -\frac{1}{\rho}\mathbf{A}^T(\mathbf{A}\mathbf{x}^* - \mathbf{y}), \tag{C.9}$$
$$\mathbf{u}^* = \nabla R_\theta(\mathbf{z}^*),$$

*where* $R_\theta$ *is the regularization function associated to the LPN* $f_\theta$ *(i.e.,* $f_\theta = \operatorname{prox}_{R_\theta}$*), which is continuously differentiable. In particular, the primal limit* $\mathbf{x}^*$ *is a critical point of the regularized reconstruction cost* $\mathbf{x} \mapsto \frac{1}{2}\|\mathbf{y} - \mathbf{A}\mathbf{x}\|_2^2 + \rho R_\theta(\mathbf{x})$, *and the full limit iterate* $(\mathbf{x}^*, \mathbf{u}^*, \mathbf{z}^*)$ *is a fixed point of the PnP-ADMM iteration (Algorithm 1).*

We restate convergence results of Themelis & Patrinos (2020) in lesser generality, given the additional regularity properties present in our setting of interest.

**Problem 2.** *Let* $h_1 : \mathbb{R}^n \to \mathbb{R}$ *and* $h_2 : \mathbb{R}^n \to \mathbb{R}$ *be continuously differentiable. We consider the minimization problem*

$$\min_{\mathbf{x} \in \mathbb{R}^n} h_1(\mathbf{x}) + h_2(\mathbf{x}). \tag{C.10}$$

**Algorithm C.2** (Themelis & Patrinos (2020, Eqns. (1.2), (ADMM), (1.3)))**.** *Perform variable splitting in* (C.10) *to obtain an equivalent problem*

$$\min_{\mathbf{x} \in \mathbb{R}^n, \mathbf{z} \in \mathbb{R}^n} h_1(\mathbf{x}) + h_2(\mathbf{z}) \quad \text{s.t.} \quad \mathbf{x} - \mathbf{z} = \mathbf{0}. \tag{C.11}$$

*Fix* $\rho > 0$. *Form the augmented Lagrangian for* (C.11) *at level* $\rho$, *that is, the function*

$$\mathcal{L}_\rho(\mathbf{x}, \mathbf{z}, \mathbf{y}) = h_1(\mathbf{x}) + h_2(\mathbf{z}) + \langle \mathbf{y}, \mathbf{x} - \mathbf{z} \rangle + \frac{\rho}{2}\|\mathbf{x} - \mathbf{z}\|_2^2, \tag{C.12}$$

*and consider the following iteration,*[14]*:*

$$\mathbf{x}^+ \in \operatorname{argmin} \mathcal{L}_\rho(\,\cdot\,, \mathbf{z}, \mathbf{y}),$$
$$\mathbf{y}^+ = \mathbf{y} + \rho(\mathbf{x}^+ - \mathbf{z}), \tag{C.13}$$
$$\mathbf{z}^+ \in \operatorname{argmin} \mathcal{L}_\rho(\mathbf{x}^+, \,\cdot\,, \mathbf{y}^+).$$

*This iteration induces a set-valued map* $\mathcal{T}_\rho : \mathbb{R}^n \times \mathbb{R}^n \rightrightarrows \mathbb{R}^n \times \mathbb{R}^n \times \mathbb{R}^n$. *Given an initialization* $(\mathbf{y}_0, \mathbf{z}_0)$, *a sequence of ADMM iterates* $(\mathbf{x}_k, \mathbf{y}_k, \mathbf{z}_k)$ *is defined inductively by* $(\mathbf{x}_k, \mathbf{y}_k, \mathbf{z}_k) \in \mathcal{T}_\rho(\mathbf{y}_{k-1}, \mathbf{z}_{k-1})$, *for* $k \in \mathbb{N}$.[15]

Themelis & Patrinos (2020) provide the following convergence guarantee for Algorithm C.2 relative to the objective (C.10), under weak assumptions on $h_1$ and $h_2$:

---

[14]This corresponds to setting the "relaxation parameter" $\lambda$ in Themelis & Patrinos (2020, Eqn. (ADMM)) to 1.

[15]The variable $\mathbf{y}$ defined here follows the notation of Themelis & Patrinos (2020), and in particular should not be confused with the measurements in the inverse problems framework. We hope the reader will forgive this conflict of notation.

**Theorem C.5** (Themelis & Patrinos (2020, Theorem 5.6, Theorem 4.1, Theorem 5.8)[16]). *Suppose the objective $h_1 + h_2$ is coercive;[17] that $h_1$ is continuously differentiable, its gradient $\nabla h_1$ is L-Lipschitz, and there exists $\sigma \in \mathbb{R}$ such that $h_1 + \frac{\sigma}{2}\| \cdot \|_2^2$ is convex;[18] and $h_2$ is proper and lower semicontinuous. Moreover, suppose the penalty parameter $\rho$ is chosen so that $\rho > \max\{2 \max\{-\sigma, 0\}, L\}$, and that the augmented Lagrangian (C.12) is a KL function (see Appendix C.4.4, Definition C.1).[19] Then the sequence $(\mathbf{x}_k, \mathbf{y}_k, \mathbf{z}_k)_{k \in \{0,1,\dots\}}$ converges to a limit point $(\mathbf{x}^*, \mathbf{y}^*, \mathbf{z}^*)$ which satisfies the KKT conditions*

$$-\mathbf{y}^* = \nabla h_1(\mathbf{x}^*)$$
$$\mathbf{y}^* \in \partial h_2(\mathbf{z}^*)$$
$$\mathbf{x}^* - \mathbf{z}^* = \mathbf{0}.$$

*Here, $\partial h_2$ is the (limiting) subdifferential mapping of $h_2$.[20] More concisely, the limit satisfies $\mathbf{0} \in \nabla h_1(\mathbf{x}^*) + \partial h_2(\mathbf{x}^*)$, and in particular $\mathbf{x}^*$ is a critical point for the objective (C.10).*

It is standard to argue that Algorithm 1 corresponds to Algorithm C.2 up to a simple reparameterization (i.e., relabeling of variables), given that the LPN $f_\theta$ in Algorithm 1 is a proximal operator.

**Lemma C.6.** *An ADMM sequence $(\mathbf{x}_k, \mathbf{y}_k, \mathbf{z}_k)$ generated by the update (C.13) is linearly isomorphic to a sequence generated by the update rule*

$$\mathbf{x}^+ \in \text{prox}_{\frac{1}{\rho} h_1} (\mathbf{z} - \mathbf{u})$$
$$\mathbf{u}^+ = \mathbf{u} + (\mathbf{x}^+ - \mathbf{z}) \qquad\qquad (C.14)$$
$$\mathbf{z}^+ \in \text{prox}_{\frac{1}{\rho} h_2} (\mathbf{x}^+ + \mathbf{u}^+),$$

*in the sense that if $(\mathbf{x}'_k, \mathbf{u}'_k, \mathbf{z}'_k)$ is the corresponding sequence of iterates generated by this update rule with initialization $\mathbf{z}_0 = \mathbf{z}'_0$ and $\mathbf{u}_0 = \frac{1}{\rho}\mathbf{y}_0$, then we have $\mathbf{x}_k = \mathbf{x}'_k$, $\mathbf{z}_k = \mathbf{z}'_k$, and $\mathbf{u}_k = \frac{1}{\rho}\mathbf{y}_k$ for every $k \in \mathbb{N}_0$.*

*Proof.* Notice that we can write in (C.12) by completing the square

$$\mathcal{L}_\rho(\mathbf{x}, \mathbf{z}, \mathbf{y}) = h_1(\mathbf{x}) + h_2(\mathbf{z}) + \frac{\rho}{2} \left\| \frac{1}{\rho}\mathbf{y} + (\mathbf{x} - \mathbf{z}) \right\|_2^2 - \frac{\rho}{2} \left\| \frac{1}{\rho}\mathbf{y} \right\|_2^2, \qquad (C.15)$$

---

[16]In obtaining the result stated here from Themelis & Patrinos (2020, Theorem 5.6), we simplify the "image function" expressions (Themelis & Patrinos, 2020, Definition 5.1) using the simple constraint structure of the ADMM problem (C.11): in particular, in checking (Themelis & Patrinos, 2020, Assumption II), we have that $\varphi_1(\mathbf{s}) = \inf_{\mathbf{x} \in \mathbb{R}^n}\{h_1(\mathbf{x}) \mid \mathbf{x} = \mathbf{s}\} = h_1(\mathbf{s})$ and similarly $\varphi_2(\mathbf{s}) = \inf_{\mathbf{z} \in \mathbb{R}^n}\{h_2(\mathbf{z}) \mid -\mathbf{z} = \mathbf{s}\} = h_2(-\mathbf{s})$. In particular $\varphi_1 = h_1$ and $\varphi_2 = h_2 \circ - \text{Id}$, so that (Themelis & Patrinos, 2020, Assumption II.A4) is implied by (Themelis & Patrinos, 2020, Assumption II.A1). This also allows us to translate the convergence guarantees of Themelis & Patrinos (2020, Theorems 5.6, 5.8) from applying to the sequence of iterate images under the constraint maps to the sequence of iterates themselves.

[17]This implies that the objective function of the equivalent penalized version of (C.11) is level bounded and admits a solution (recall that the latter is a consequence of the Weierstrass theorem, e.g. (Bertsekas, 2016, Proposition A.8(2))).

[18]Such a $\sigma$ always exists, and satisfies $|\sigma| \le L$. Roughly speaking, the smaller a value of $\sigma$ can be chosen, the better—this is possible when $h_1$ is 'more convex'.

[19]Although Themelis & Patrinos (2020) state their global convergence result, Theorem 5.8, only in the semialgebraic setting, inspection of their arguments (notably (Themelis & Patrinos, 2020, p. 163 top), and the connecting discussion in (Li & Pong, 2016, Theorem 2, Remark 2(ii)), together with the equivalence between DRS and ADMM in (Themelis & Patrinos, 2020, Theorem 5.5)) reveals that it is only necessary that the augmented Lagrangian $\mathcal{L}_\rho$ is a KL function.

[20]This is the same notion of subdifferential introduced in Appendix C.4.1 in order to state the results of Boţ et al. (2016). We use the fact that the limiting subdifferential coincides (up to converting a single-valued set-valued map into a function) with the gradient for a $C^1$ function (Rockafellar & Wets, 1998, Theorem 9.18, Corollary 9.19).

in order to simplify the minimization operations in (C.13). Indeed, the iteration (C.13) then becomes equivalent to, with (C.15), the iteration

$$
\begin{aligned}
\mathbf{x}^+ &\in \operatorname{prox}_{\frac{1}{\rho}h_1}\left(\mathbf{z} - \tfrac{1}{\rho}\mathbf{y}\right) \\
\mathbf{y}^+ &= \mathbf{y} + \rho(\mathbf{x}^+ - \mathbf{z}) \\
\mathbf{z}^+ &\in \operatorname{prox}_{\frac{1}{\rho}h_2}\left(\mathbf{x}^+ + \tfrac{1}{\rho}\mathbf{y}^+\right).
\end{aligned}
\tag{C.16}
$$

Introducing now the "scaled dual variable" $\mathbf{u} = \tfrac{1}{\rho}\mathbf{y}$, we have the equivalent update

$$
\begin{aligned}
\mathbf{x}^+ &\in \operatorname{prox}_{\frac{1}{\rho}h_1}\left(\mathbf{z} - \mathbf{u}\right) \\
\mathbf{u}^+ &= \mathbf{u} + (\mathbf{x}^+ - \mathbf{z}) \\
\mathbf{z}^+ &\in \operatorname{prox}_{\frac{1}{\rho}h_2}\left(\mathbf{x}^+ + \mathbf{u}^+\right).
\end{aligned}
\tag{C.17}
$$

The equivalence claims in the statement of the lemma follow from this chain of reasoning. $\qquad\square$

With this preparation completed, we are now ready to give the proof of Theorem C.4.

*Proof of Theorem C.4.* Below, to avoid a notational conflict with the dual variables in Algorithm C.2, we will write $\mathbf{y}_{\mathrm{meas}}$ for the measurements that define the data fidelity term in the inverse problem cost.

We have by assumption and Proposition 3.1 and Lemma C.7 that there exists a coercive $C^1$ function $R_\theta : \mathbb{R}^n \to \mathbb{R}$ such that $f_\theta = \operatorname{prox}_{R_\theta}$. Given the initialization $\mathbf{u}_0 = \mathbf{0}$ in Algorithm 1, applying Lemma C.6 implies that Algorithm 1 corresponds to a sequence of ADMM iterates $(\mathbf{x}_k, \mathbf{y}_k, \mathbf{z}_k)$ generated via Algorithm C.2 with the initialization $\mathbf{y}_0 = \mathbf{0}$, the objectives $h_1(\mathbf{x}) = \frac{1}{2}\|\mathbf{y}_{\mathrm{meas}} - \mathbf{A}\mathbf{x}\|_2^2$ and $h_2(\mathbf{z}) = \rho R_\theta(\mathbf{z})$, and the correspondence $\mathbf{y}_k = \rho \mathbf{u}_k$. In addition, we observe that $h_1$ is smooth, nonnegative, convex, and satisfies $\|\nabla^2 h_1\| = \|\mathbf{A}^T\mathbf{A}\|$, where $\|\cdot\|$ denotes the operator norm. In turn, we know that $\mathbf{z} \mapsto h_2(\mathbf{z}) + \frac{\rho}{2}\|\mathbf{z}\|_2^2$ is (strongly) convex: Lemma C.10 implies that $f_\theta$ is Lipschitz, and (Gribonval & Nikolova, 2020, Proposition 2(1)) implies that an $L$-Lipschitz proximal operator's associated prox-primitive is $(1-1/L)$-weakly convex, from which it follows that the sum $\mathbf{z} \mapsto h_2(\mathbf{z}) + \frac{\rho}{2}\|\mathbf{z}\|_2^2$ is strongly convex. As a result of these facts, every minimization operation in Algorithm C.2 has a unique minimizer, and we can interchange set inclusion operations with equalities when describing the ADMM sequence corresponding to Algorithm 1 without any concern in the sequel.

Now, these instantiations of $h_1$ and $h_2$ verify the elementary hypotheses of Theorem C.5:

1. Nonnegativity of $h_1$ and coercivity of $h_2$ imply that $h_1 + h_2$ is coercive;

2. $h_1$ is smooth and we can take $L = \|\mathbf{A}^T\mathbf{A}\|$ and $\sigma = 0$, and therefore the hypothesis that $\rho > \|\mathbf{A}^T\mathbf{A}\|$ verifies the conditions on the penalty parameter;

3. By Lemma C.7, $h_2$ is real-valued and $C^1$, as above.

Finally, to check the KL property of the augmented Lagrangian $\mathcal{L}_\rho$ defined by (C.12), we will follow Appendix C.4.4 and verify that $\mathcal{L}_\rho$ is definable in an o-minimal structure, then apply Proposition C.11, Property 1, since $\mathcal{L}_\rho$ is $C^1$ as a sum of $C^1$ functions (both $h_1$ and $h_2$ are so). To this end, note by Corollary C.12 that $\mathcal{L}_\rho$ is definable in the o-minimal structure asserted by Property 2 if both $h_1$ and $h_2$ are definable in that o-minimal structure. Proposition C.11, Property 2 implies that $h_1$ is definable, since it is a degree two polynomial. Then Lemma C.13 implies that $h_2$ is definable in the same o-minimal structure, and it thus follows from the preceding reasoning that $\mathcal{L}_\rho$ has the KL property.

We can therefore apply Theorem C.5 to obtain that the sequence of iterates of Algorithm 1 converges, and its limit point $(\mathbf{x}^*, \mathbf{u}^*, \mathbf{z}^*)$ satisfies the KKT conditions (C.9)

$$\mathbf{u}^* = -\tfrac{1}{\rho}\mathbf{A}^T(\mathbf{A}\mathbf{x}^* - \mathbf{y}_{\mathrm{meas}}),$$
$$\mathbf{u}^* = \nabla R_\theta(\mathbf{z}^*),$$
$$\mathbf{x}^* - \mathbf{z}^* = \mathbf{0},$$

where we have used fact that the limiting subdifferential coincides (up to converting a single-valued set-valued map into a function) with the gradient for a $C^1$ function (Rockafellar & Wets, 1998, Theorem 9.18, Corollary 9.19). In particular, simplifying gives

$$\mathbf{A}^T(\mathbf{A}\mathbf{x}^* - \mathbf{y}_{\mathrm{meas}}) = -\rho\nabla R_\theta(\mathbf{x}^*), \tag{C.18}$$

which is equivalent to the claimed critical point property. To see that this is also equivalent to being a fixed point of Algorithm 1, it is convenient to use the expression (C.17) for the ADMM update expression, which appeared in the proof of Lemma C.6. The KKT conditions (C.9) imply that $\nabla h_1(\mathbf{x}^*) = -\nabla h_2(\mathbf{x}^*)$, and since both $\tfrac{1}{\rho}h_1$ and $\tfrac{1}{\rho}h_2$ are differentiable and yield a strongly convex function when summed with a quadratic $\tfrac{1}{2}\|\cdot\|_2^2$, we can express the action of their proximal operators as

$$\mathrm{prox}_{\frac{1}{\rho}h_i} = (\mathrm{Id} + \tfrac{1}{\rho}\nabla h_i)^{-1}, \quad i = 1, 2. \tag{C.19}$$

From (C.17), we get that $\mathbf{x}^+ = (\mathrm{Id} + \tfrac{1}{\rho}\nabla h_1)^{-1}(\mathbf{z}^* - \mathbf{u}^*)$. The KKT conditions (C.9) imply that $\mathbf{u}^* = -\tfrac{1}{\rho}\nabla h_1(\mathbf{x}^*)$ and $\mathbf{z}^* = \mathbf{x}^*$, so that $\mathbf{z}^* - \mathbf{u}^* = (\mathrm{Id} + \tfrac{1}{\rho}\nabla h_1)(\mathbf{x}^*)$, and therefore indeed $\mathbf{x}^+ = \mathbf{x}^*$. Proceeding, we then have that $\mathbf{u}^+ = \mathbf{u}^* - \mathbf{z}^* + \mathbf{x}^*$, which gives that $\mathbf{u}^+ = \mathbf{u}^*$, since $\mathbf{x}^* = \mathbf{z}^*$. Finally, we obtain from the previous two steps and the definition of $h_2$ that

$$\mathbf{z}^+ = (\mathrm{Id} + \nabla R_\theta)^{-1}(\mathbf{z}^* + \mathbf{u}^*), \tag{C.20}$$

and the final KKT condition (C.9) implies that $\mathbf{u}^* = \nabla R_\theta(\mathbf{z}^*)$. Hence

$$\mathbf{z}^+ = (\mathrm{Id} + \nabla R_\theta)^{-1}(\mathrm{Id} + \nabla R_\theta)(\mathbf{z}^*) = \mathbf{z}^*, \tag{C.21}$$

and we indeed conclude that $(\mathbf{x}^*, \mathbf{u}^*, \mathbf{z}^*)$ is a fixed point of Algorithm 1. $\qquad\square$

### C.4.3 Regularity of the Regularization Function of LPNs

As discussed in the "recipe" of Appendix C.4, in this section we prove basic regularity properties of the regularization function associated to any LPN with the architecture of Proposition 3.1.

**Lemma C.7** (Regularity Properties of LPNs). *Suppose $f_\theta$ is an LPN constructed following the recipe in Proposition 3.1, with softplus activations $\sigma(x) = (1/\beta)\log(1 + \exp(\beta x))$, where $\beta > 0$ is an arbitrary constant, and with strong convexity weight $0 < \alpha < 1$. Let $f_\theta(\mathbf{y}) = \nabla\psi_\theta(\mathbf{y}) + \alpha\mathbf{y}$ be the defining equation of the LPN. Then there is a real-valued function $R_\theta : \mathbb{R}^n \to \mathbb{R}$ such that $f_\theta = \mathrm{prox}_{R_\theta}$. Moreover, we have the following regularity properties:*

1. *$R_\theta$ is coercive, i.e., we have $R_\theta(\mathbf{x}) \to +\infty$ as $\|\mathbf{x}\|_2 \to +\infty$.*

2. *$f_\theta : \mathbb{R}^n \to \mathbb{R}^n$ is surjective and invertible, with an inverse mapping $f_\theta^{-1} : \mathbb{R}^n \to \mathbb{R}^n$ which is continuous.*

3. *$R_\theta$ is continuously differentiable. In particular, it holds*

$$R_\theta(\mathbf{x}) = (1-\alpha)\langle f_\theta^{-1}(\mathbf{x}), \nabla\psi_\theta(f_\theta^{-1}(\mathbf{x}))\rangle$$
$$+ \frac{\alpha(1-\alpha)}{2}\|f_\theta^{-1}(\mathbf{x})\|_2^2 - \tfrac{1}{2}\|\nabla\psi_\theta(f_\theta^{-1}(\mathbf{x}))\|_2^2 - \psi_\theta(f_\theta^{-1}(\mathbf{x})). \tag{C.22}$$

*Remark.* Lemma C.7 does not, strictly speaking, require the softplus activation: the proof shows that any Lipschitz activation function with enough differentiability and slow growth at infinity, such as another smoothed verison of the ReLU activation, the GeLU, or the Swish activation, would also work.

*Proof of Lemma C.7.* The main technical challenge will be to establish coercivity of $R_\theta$, which always exists as necessary, by Propositions 2.1 and 3.1. We will therefore pursue this estimate as the main line of the proof, establishing the remaining assertions in the result statement along the way.

By Proposition 3.1, there exists $R_\theta$ such that $f_\theta = \mathrm{prox}_{R_\theta}$. Now, using (Gribonval & Nikolova, 2020, Theorem 4(a)), for every $\mathbf{y} \in \mathbb{R}^n$,

$$R_\theta(f_\theta(\mathbf{y})) = \langle \mathbf{y}, f_\theta(\mathbf{y}) \rangle - \tfrac{1}{2}\|f_\theta(\mathbf{y})\|_2^2 - \left(\psi_\theta(\mathbf{y}) + \tfrac{\alpha}{2}\|\mathbf{y}\|_2^2\right).$$

Using the definition of $f_\theta$ and minor algebra, we rewrite this as

$$R_\theta(f_\theta(\mathbf{y})) = \langle \mathbf{y}, \nabla\psi_\theta(\mathbf{y}) + \alpha\mathbf{y} \rangle - \tfrac{1}{2}\|\nabla\psi_\theta(\mathbf{y}) + \alpha\mathbf{y}\|_2^2 - \left(\psi_\theta(\mathbf{y}) + \tfrac{\alpha}{2}\|\mathbf{y}\|_2^2\right)$$

$$= (1-\alpha)\langle \mathbf{y}, \nabla\psi_\theta(\mathbf{y}) \rangle + \frac{\alpha(1-\alpha)}{2}\|\mathbf{y}\|_2^2 - \tfrac{1}{2}\|\nabla\psi_\theta(\mathbf{y})\|_2^2 - \psi_\theta(\mathbf{y}). \qquad \text{(C.23)}$$

At this point, we observe that by Lemma C.8, the map $f_\theta : \mathbb{R}^n \to \mathbb{R}^n$ is invertible and surjective, with a continuous inverse mapping. This establishes the second assertion that we have claimed. In addition, taking inverses in (C.23) implies (C.22) and as a consequence the fact that $R_\theta$ is real-valued, and the fact that it is continuously differentiable on $\mathbb{R}^n$ is then an immediate consequence of (Gribonval & Nikolova, 2020, Corollary 6(b)). To conclude, it only remains to show that $R_\theta$ is coercive, which we will accomplish by lower bounding the RHS of (C.23). By Lemma C.9, $\psi_\theta$ is $L$-Lipschitz for a constant $L > 0$. Thus, we have for every $\mathbf{y}$ (by the triangle inequality)

$$|\psi_\theta(\mathbf{y})| \leq L\|\mathbf{y}\|_2 + K$$

for a (finite) constant $K \in \mathbb{R}$, depending only on $\theta$. Now, the Cauchy-Schwarz inequality implies from the previous two statements (and $\|\nabla\psi_\theta\|_2 \leq L$ by the Lipschitz property of $\psi_\theta$)

$$R_\theta(f_\theta(\mathbf{y})) \geq -(1-\alpha)\|\mathbf{y}\|_2\|\nabla\psi_\theta(\mathbf{y})\|_2 + \frac{\alpha(1-\alpha)}{2}\|\mathbf{y}\|_2^2 - \tfrac{1}{2}\|\nabla\psi_\theta(\mathbf{y})\|_2^2 - L\|\mathbf{y}\|_2 - K,$$

$$\geq -L(1-\alpha)\|\mathbf{y}\|_2 + \frac{\alpha(1-\alpha)}{2}\|\mathbf{y}\|_2^2 - \frac{L^2}{2} - L\|\mathbf{y}\|_2 - K.$$

We rewrite this estimate with some algebra as

$$R_\theta(f_\theta(\mathbf{y})) \geq \|\mathbf{y}\|_2\left(\frac{\alpha(1-\alpha)}{2}\|\mathbf{y}\|_2 - L(1-\alpha) - L\right) - \frac{L^2}{2} - K.$$

Next, we notice that when $0 < \alpha < 1$, the coefficient $\alpha(1-\alpha) > 0$; hence there is a constant $M > 0$ depending only on $\alpha$ and $L$ such that for every $\mathbf{y}$ with $\|\mathbf{y}\|_2 \geq M$, one has

$$\frac{\alpha(1-\alpha)}{2}\|\mathbf{y}\|_2 - L(1-\alpha) - L \geq \frac{\alpha(1-\alpha)}{4}\|\mathbf{y}\|_2.$$

In turn, iterating this exact argument implies that there is another constant $M' > 0$ (depending only on $\alpha$, $L$, and $K$) such that whenever $\|\mathbf{y}\|_2 \geq M'$, one has

$$R_\theta(f_\theta(\mathbf{y})) \geq \frac{\alpha(1-\alpha)}{8}\|\mathbf{y}\|_2^2.$$

We can therefore rewrite the previous inequality as

$$R_\theta(\mathbf{x}) \geq \frac{\alpha(1-\alpha)}{8}\|f_\theta^{-1}(\mathbf{x})\|_2^2, \qquad \text{(C.24)}$$

for every $\mathbf{x}$ such that $\|f^{-1}(\mathbf{x})\|_2 \geq M'$. To conclude, we will show that whenever $\|\mathbf{x}\|_2 \to +\infty$, we also have $\|f_\theta^{-1}(\mathbf{x})\|_2 \to +\infty$, which together with (C.24) will imply coercivity of $R_\theta$. To this end, write $\|\cdot\|_{\mathrm{Lip}}$ for the Lipschitz seminorm:

$$\|f\|_{\mathrm{Lip}} = \sup_{\mathbf{y} \neq \mathbf{y}'} \frac{\|f(\mathbf{y}) - f(\mathbf{y}')\|_2}{\|\mathbf{y} - \mathbf{y}'\|_2},$$

and note that $\|f_\theta\|_{\mathrm{Lip}} \leq \|\nabla\psi_\theta\|_{\mathrm{Lip}} + \alpha$. By Lemma C.10, $\nabla\psi_\theta$ is $L_{\nabla\psi_\theta}$-Lipschitz continuous, thus $f_\theta$ is $(L_{\nabla\psi_\theta} + \alpha)$-Lipschitz continuous,

$$\|f_\theta(\mathbf{y}) - f_\theta(\mathbf{y}')\|_2 \leq (L_{\nabla\psi_\theta} + \alpha)\|\mathbf{y} - \mathbf{y}'\|_2.$$

Thus, taking inverses, we have

$$\|f_\theta^{-1}(\mathbf{x}) - f_\theta^{-1}(\mathbf{0})\|_2 \geq \frac{1}{L_{\nabla\psi_\theta} + \alpha}\|\mathbf{x}\|_2,$$

and it then follows from the triangle inequality that whenever $\mathbf{x}$ is such that $\|\mathbf{x}\|_2 \geq 2(L_{\nabla\psi_\theta} + \alpha)\|f_\theta^{-1}(\mathbf{0})\|_2$, we have in fact

$$\|f_\theta^{-1}(\mathbf{x})\|_2 \geq \frac{1}{2(L_{\nabla\psi_\theta} + \alpha)}\|\mathbf{x}\|_2.$$

Combining this estimate with (C.24), we obtain that for every $\mathbf{x}$ such that $\|\mathbf{x}\|_2 \geq 2(L_{\nabla\psi_\theta} + \alpha)\|f_\theta^{-1}(\mathbf{0})\|_2$ and $\|\mathbf{x}\|_2 \geq 2M'(L_{\nabla\psi_\theta} + \alpha)$, it holds

$$R_\theta(\mathbf{x}) \geq \frac{\alpha(1-\alpha)}{32(L_{\nabla\psi_\theta} + \alpha)^2}\|\mathbf{x}\|_2^2.$$

Taking limits in this last bound yields coercivity of $R_\theta$, and hence the claim. $\qquad\square$

**Lemma C.8** (Invertibility of $f_\theta$ and Continuity of $f_\theta^{-1}$). *Suppose $f_\theta$ is an LPN constructed following the recipe in Proposition 3.1, with softplus activations $\sigma(x) = (1/\beta)\log(1 + \exp(\beta x))$, where $\beta > 0$ is an arbitrary constant, and with strong convexity weight $0 < \alpha < 1$. Then $f_\theta : \mathbb{R}^n \to \mathbb{R}^n$ is invertible and surjective, and $f_\theta^{-1} : \mathbb{R}^n \to \mathbb{R}^n$ is $C^0$.*

*Proof.* The proof uses the invertibility construction that we describe informally in Section 3. By construction, we have $f_\theta = \nabla\psi_\theta + \alpha\,\mathrm{Id}$, where $\mathrm{Id}$ denotes the identity operator on $\mathbb{R}^n$ (i.e., $\mathrm{Id}(\mathbf{x}) = \mathbf{x}$ for every $\mathbf{x} \in \mathbb{R}^n$).

For a fixed $\mathbf{x} \in \mathbb{R}^n$, consider the strongly convex minimization problem $\min_{\mathbf{y}} \psi_\theta(\mathbf{y}) + \frac{\alpha}{2}\|\mathbf{y}\|_2^2 - \langle\mathbf{x}, \mathbf{y}\rangle$. By first-order optimality condition, the minimizers are exactly $\{\mathbf{y} \mid \nabla\psi_\theta(\mathbf{y}) + \alpha\mathbf{y} = \mathbf{x}\}$. Furthermore, since the problem is strongly convex, it has a unique minimizer for each $\mathbf{x} \in \mathbb{R}^n$ (Boyd & Vandenberghe, 2004). Therefore, for each $\mathbf{x} \in \mathbb{R}^n$, there exists a unique $\mathbf{y}$ such that $\mathbf{x} = \nabla\psi_\theta(\mathbf{y}) + \alpha\mathbf{y} = f_\theta(\mathbf{y})$.

The argument above establishes that $f_\theta : \mathbb{R}^n \to \mathbb{R}^n$ is injective and surjective; hence there exists an inverse $f_\theta^{-1} : \mathbb{R}^n \to \mathbb{R}^n$. To conclude the proof, we will argue that $f_\theta^{-1}$ is continuous. To this end, we use the characterization of continuity which states that a function $g : \mathbb{R}^n \to \mathbb{R}^n$ is continuous if and only if for every open set $U \subset \mathbb{R}^n$, we have that $g^{-1}(U)$ is open, where $g^{-1}(U) = \{\mathbf{x} \in \mathbb{R}^n \mid g(\mathbf{x}) \in U\}$ (e.g., (Rudin, 1976, Theorem 4.8)). To show that $f_\theta^{-1}$ is continuous, it is therefore equivalent to show that for every open set $U \subset \mathbb{R}^n$, one has that $f_\theta(U)$ is open. But this follows from invariance of domain, a standard result in algebraic topology (e.g., (Dold, 2012, Proposition 7.4)), since $f_\theta$ is injective and continuous. We have thus shown that $f_\theta$ is invertible, and that its inverse is continuous, as claimed. $\qquad\square$

**Lemma C.9** (Lipschitzness of $\psi_\theta$). *Suppose $f_\theta$ is an LPN constructed following the recipe in Proposition 3.1, with softplus activations $\sigma(x) = (1/\beta)\log(1 + \exp(\beta x))$, where $\beta > 0$ is an arbitrary constant, and let $\psi_\theta$ denote the convex potential function for the LPN. Then $\psi_\theta$ is $L_{\psi_\theta}$-Lipschitz continuous for a constant $L_{\psi_\theta} > 0$, i.e., $|\psi_\theta(\mathbf{y}) - \psi_\theta(\mathbf{y}')| \leq L_{\psi_\theta}\|\mathbf{y} - \mathbf{y}'\|_2$, for all $\mathbf{y}, \mathbf{y}' \in \mathbb{R}^n$.*

*Proof.* Note that the derivative $\sigma'$ of the softplus activation satisfies $\sigma'(x) = 1/(1 + \exp(-\beta x))$, which is no larger than 1, since $\exp(x) > 0$ for $x \in \mathbb{R}$. If $F$ is a map between Euclidean spaces we will write $DF$ for its differential (a map from the domain of $F$ to the space of linear operators from the domain of $F$ to the range of $F$). Hence the activation function $g$ in Proposition 3.1 is 1-Lipschitz with respect to the $\ell_2$ norm, since the induced (by elementwise application) map $g : \mathbb{R}^n \to \mathbb{R}^n$ defined by $g(\mathbf{y}) = [\sigma(x_1), \ldots, \sigma(x_n)]^T$ satisfies

$$Dg(\mathbf{y}) = \begin{bmatrix} \sigma'(x_1) & & \\ & \ddots & \\ & & \sigma'(x_n) \end{bmatrix},$$

which is bounded in operator norm by $\sup_x |\sigma'(x)| \leq 1$. First, notice that

$$\|\psi_\theta(\mathbf{y}) - \psi_\theta(\mathbf{y}')\|_2 = \|\mathbf{w}^T(\mathbf{z}_K(\mathbf{y}) - \mathbf{z}_K(\mathbf{y}'))\|_2$$
$$\leq \|\mathbf{w}\|_2 \|\mathbf{z}_K(\mathbf{y}) - \mathbf{z}_K(\mathbf{y}')\|_2$$

by Cauchy-Schwarz. Meanwhile, we have similarly

$$\|\mathbf{z}_1(\mathbf{y}) - \mathbf{z}_1(\mathbf{y}')\|_2 \leq \|\mathbf{H}_1\| \|\mathbf{y} - \mathbf{y}'\|_2,$$

where $\| \cdot \|$ denotes the operator norm of a matrix, and for integer $0 < k < K + 1$

$$\|\mathbf{z}_k(\mathbf{y}) - \mathbf{z}_k(\mathbf{y}')\|_2 \leq \|\mathbf{W}_k\| \|\mathbf{z}_{k-1}(\mathbf{y}) - \mathbf{z}_{k-1}(\mathbf{y}')\|_2 + \|\mathbf{H}_k\| \|\mathbf{y} - \mathbf{y}'\|_2.$$

By a straightforward induction, it follows that $\psi_\theta$ is $L$-Lipschitz for a constant $L > 0$ (depending only on $\theta$). $\qquad\square$

**Lemma C.10** (Lipschitzness of $\nabla\psi_\theta$ and LPNs $f_\theta$). *Suppose $f_\theta$ is an LPN constructed following the recipe in Proposition 3.1, with softplus activations $\sigma(x) = (1/\beta)\log(1 + \exp(\beta x))$, where $\beta > 0$ is an arbitrary constant, and with strong convexity weight $0 < \alpha < 1$. Let $f_\theta(\mathbf{y}) = \nabla\psi_\theta(\mathbf{y}) + \alpha\mathbf{y}$ be the defining equation of the LPN. Then $\nabla\psi_\theta$ is $L_{\nabla\psi_\theta}$-Lipschitz continuous, for a constant $L_{\nabla\psi_\theta} > 0$. In particular, $f_\theta$ is $(\alpha + L_{\nabla\psi_\theta})$-Lipschitz continuous.*

*Proof.* The claimed expression for Lipschitzness of $f_\theta$ follows from the claimed expression for Lipschitzness of $\nabla\psi_\theta$, by differentiating and using the triangle inequality. We recall basic notions of Lipschitz continuity of differentiable mappings at the beginning of the proof of Lemma C.9, which we will make use of below. We will upper bound $\|\nabla\psi_\theta\|_{\mathrm{Lip}}$ by deriving an explicit expression for the gradient. By the defining formulas in Proposition 3.1, we have

$$\psi_\theta(\mathbf{y}) = \mathbf{w}^T \mathbf{z}_K(\mathbf{y}) + \mathbf{b}.$$

The chain rule gives

$$\nabla\psi_\theta(\mathbf{y}) = D\mathbf{z}_K(\mathbf{y})^T \mathbf{w},$$

where $^T$ denotes the adjoint of a linear operator, so for any $\mathbf{y}, \mathbf{y}'$ we have

$$\|\nabla\psi_\theta(\mathbf{y}) - \nabla\psi_\theta(\mathbf{y}')\|_2 = \left\| (D\mathbf{z}_K(\mathbf{y}) - D\mathbf{z}_K(\mathbf{y}'))^T \mathbf{w} \right\|_2$$
$$\leq \left\| (D\mathbf{z}_K(\mathbf{y}) - D\mathbf{z}_K(\mathbf{y}'))^T \right\| \|\mathbf{w}\|_2$$
$$= \|D\mathbf{z}_K(\mathbf{y}) - D\mathbf{z}_K(\mathbf{y}')\| \|\mathbf{w}\|_2$$
$$\leq \|D\mathbf{z}_K(\mathbf{y}) - D\mathbf{z}_K(\mathbf{y}')\|_{\mathrm{F}} \|\mathbf{w}\|_2,$$

where the first inequality uses Cauchy-Schwarz, the third line uses that the operator norm of a linear operator is equal to that of its adjoint, and the third line uses that the operator norm is upper-bounded by the Frobenius norm. This shows that we obtain a Lipschitz property in $\ell_2$ for $\nabla\psi_\theta$ by obtaining one for the differential $D\mathbf{z}_K$ of the LPN's last-layer features. To this end, we can use the chain rule to compute for any integer $1 < k < K + 1$ and any $\boldsymbol{\delta} \in \mathbb{R}^n$

$$D\mathbf{z}_k(\mathbf{y})(\boldsymbol{\delta}) = g'\left(\mathbf{W}_k\mathbf{z}_{k-1}(\mathbf{y}) + \mathbf{H}_k\mathbf{y} + \mathbf{b}_k\right) \odot \left[\mathbf{W}_k D\mathbf{z}_{k-1}(\mathbf{y})(\boldsymbol{\delta}) + \mathbf{H}_k\boldsymbol{\delta}\right],$$

where $g'$ is the derivative of the softplus activation function $g$, applied elementwise, and $\odot$ denotes elementwise multiplication, and similarly

$$D\mathbf{z}_1(\mathbf{y})(\boldsymbol{\delta}) = g'\left(\mathbf{H}_1\mathbf{y} + \mathbf{b}_1\right) \odot \left[\mathbf{H}_1\boldsymbol{\delta}\right].$$

Now notice that for any vectors $\mathbf{v}$ and $\mathbf{y}$ and any matrix $\mathbf{A}$ such that the sizes are compatible, we have $\mathbf{v} \odot (\mathbf{A}\mathbf{y}) = \mathrm{diag}(\mathbf{v})\mathbf{A}\mathbf{y}$. Hence we can rewrite the above recursion in matrix form as

$$D\mathbf{z}_k(\mathbf{y}) = \underbrace{\mathrm{diag}\left(g'\left(\mathbf{W}_k\mathbf{z}_{k-1}(\mathbf{y}) + \mathbf{H}_k\mathbf{y} + \mathbf{b}_k\right)\right)}_{\mathbf{D}_k(\mathbf{y})} \left[\mathbf{W}_k D\mathbf{z}_{k-1}(\mathbf{y}) + \mathbf{H}_k\right],$$

and similarly

$$D\mathbf{z}_1(\mathbf{y}) = \underbrace{\mathrm{diag}\left(g'\left(\mathbf{H}_1\mathbf{y} + \mathbf{b}_1\right)\right)}_{\mathbf{D}_1(\mathbf{y})} \mathbf{H}_1.$$

We will proceed with an inductive argument. First, by the submultiplicative property of the Frobenius norm and the triangle inequality for the Frobenius norm, note that we have if $1 < k < K+1$

$$
\begin{aligned}
\|D\mathbf{z}_k(\mathbf{y}) - D\mathbf{z}_k(\mathbf{y}')\|_\mathrm{F} &\leq \|\mathbf{D}_k(\mathbf{y}) - \mathbf{D}_k(\mathbf{y}')\|_\mathrm{F} \\
&\quad + \|\mathbf{D}_k(\mathbf{y})\mathbf{W}_k D\mathbf{z}_{k-1}(\mathbf{y}) - \mathbf{D}_k(\mathbf{y}')\mathbf{W}_k D\mathbf{z}_{k-1}(\mathbf{y}')\|_\mathrm{F} \\
&\leq \|\mathbf{D}_k(\mathbf{y}) - \mathbf{D}_k(\mathbf{y}')\|_\mathrm{F} \\
&\quad + \|\mathbf{D}_k(\mathbf{y})\mathbf{W}_k D\mathbf{z}_{k-1}(\mathbf{y}) - \mathbf{D}_k(\mathbf{y})\mathbf{W}_k D\mathbf{z}_{k-1}(\mathbf{y}')\|_\mathrm{F} \\
&\quad + \|\mathbf{D}_k(\mathbf{y})\mathbf{W}_k D\mathbf{z}_{k-1}(\mathbf{y}') - \mathbf{D}_k(\mathbf{y}')\mathbf{W}_k D\mathbf{z}_{k-1}(\mathbf{y}')\|_\mathrm{F} \\
&\leq \|\mathbf{D}_k(\mathbf{y}) - \mathbf{D}_k(\mathbf{y}')\|_\mathrm{F} \\
&\quad + \|\mathbf{D}_k(\mathbf{y})\mathbf{W}_k\|_\mathrm{F}\|D\mathbf{z}_{k-1}(\mathbf{y}) - D\mathbf{z}_{k-1}(\mathbf{y}')\|_\mathrm{F} \\
&\quad + \|D\mathbf{z}_{k-1}(\mathbf{y}')\|_\mathrm{F}\|\mathbf{D}_k(\mathbf{y})\mathbf{W}_k - \mathbf{D}_k(\mathbf{y}')\mathbf{W}_k\|_\mathrm{F} \\
&\leq (1 + \|\mathbf{W}_k\|_\mathrm{F})\, \|\mathbf{D}_k(\mathbf{y}) - \mathbf{D}_k(\mathbf{y}')\|_\mathrm{F} \\
&\quad + \|\mathbf{D}_k(\mathbf{y})\|_\mathrm{F}\|\mathbf{W}_k\|_\mathrm{F}\|D\mathbf{z}_{k-1}(\mathbf{y}) - D\mathbf{z}_{k-1}(\mathbf{y}')\|_\mathrm{F}.
\end{aligned}
$$

Now, as we have shown above, $g'(x) = (1 + \exp(-\beta x))^{-1} \leq 1$ for every $x \in \mathbb{R}$. This implies

$$
\|\mathbf{D}_k(\mathbf{y})\|_\mathrm{F} \leq \sqrt{n_k},
$$

where $n_k$ is the output dimension of $k$-th layer. Moreover, we calculate with the chain rule

$$
g''(x) = \frac{\beta e^{-\beta x}}{(1 + e^{-\beta x})^2},
$$

and by L'Hôpital's rule, we have that $\lim_{x \to +\infty} \frac{x}{(1+x)^2} = 0$, so that by continuity, $g''$ is bounded for $x \in \mathbb{R}$. It follows that $g'$ is Lipschitz. Notice now that

$$
\begin{aligned}
\|\mathbf{D}_k(\mathbf{y}) - \mathbf{D}_k(\mathbf{y}')\|_\mathrm{F} &= \|g'\left(\mathbf{W}_k\mathbf{z}_{k-1}(\mathbf{y}) + \mathbf{H}_k\mathbf{y} + \mathbf{b}_k\right) - g'\left(\mathbf{W}_k\mathbf{z}_{k-1}(\mathbf{y}') + \mathbf{H}_k\mathbf{y}' + \mathbf{b}_k\right)\|_2 \\
&\leq \|g'\|_\mathrm{Lip}\left(\|\mathbf{W}_k\|_\mathrm{F}\|\mathbf{z}_{k-1}(\mathbf{y}) - \mathbf{z}_{k-1}(\mathbf{y}')\|_2 + \|\mathbf{H}_k\|_\mathrm{F}\|\mathbf{y} - \mathbf{y}'\|_2,\right)
\end{aligned}
$$

where in the second line we used the fact that the derivative of an elementwise function is a diagonal matrix together with the triangle inequality and Cauchy-Schwarz. However, we have already argued previously by induction that $\psi_\theta$ is Lipschitz, and in particular each of its feature maps $\mathbf{z}_k$ is Lipschitz. We conclude that $\mathbf{D}_k$ is Lipschitz, and the Lipschitz constant depends only on $\theta$. This means that there are constants $L_k, L_k'$ depending only on $n$ and $\theta$ such that

$$
\|D\mathbf{z}_k(\mathbf{y}) - D\mathbf{z}_k(\mathbf{y}')\|_\mathrm{F} \leq L_k\|\mathbf{y} - \mathbf{y}'\|_2 + L_k'\|D\mathbf{z}_{k-1}(\mathbf{y}) - D\mathbf{z}_{k-1}(\mathbf{y}')\|_\mathrm{F}.
$$

Meanwhile, following the same arguments as above, but in a slightly simplified setting, we obtain

$$
\begin{aligned}
\|D\mathbf{z}_1(\mathbf{y}) - D\mathbf{z}_1(\mathbf{y}')\|_\mathrm{F} &= \|\mathbf{D}_1(\mathbf{y})\mathbf{H}_1 - \mathbf{D}_1(\mathbf{y}')\mathbf{H}_1\|_\mathrm{F} \\
&\leq \|\mathbf{H}_1\|_\mathrm{F}\|\mathbf{D}_1(\mathbf{y}) - \mathbf{D}_1(\mathbf{y}')\|_\mathrm{F} \\
&\leq \|g'\|_\mathrm{Lip}\|\mathbf{H}_1\|_\mathrm{F}^2\|\mathbf{y} - \mathbf{y}'\|_2,
\end{aligned}
$$

which demonstrates that $D\mathbf{z}_1$ is also Lipschitz, with the Lipschitz constant depending only on $\theta$. By induction, we therefore conclude that there is $L_{\nabla\psi_\theta} > 0$ such that

$$
\|\nabla\psi_\theta(\mathbf{y}) - \nabla\psi_\theta(\mathbf{y}')\|_2 \leq L_{\nabla\psi_\theta}\|\mathbf{y} - \mathbf{y}'\|_2,
$$

with $L_{\nabla\psi_\theta}$ depending only on $\theta$ and $n_k$. $\qquad\square$

### C.4.4 Nondegeneracy of the Prior

For our convergence results for PnP with LPNs, we make use of general convergence analyses for nonsmooth nonconvex optimization from the literature. To invoke these results, we need to show that the regularizer $R_\theta$ associated to the LPN $f_\theta$ satisfies a certain nondegeneracy property called the Kurdyka-Łojaciewicz (KL) inequality. We provide a self-contained proof of this fact in this section. We will instantiate various concepts from the literature in our specific setting (i.e., sacrificing generality for clarity), and include appropriate references to the literature for more technical aspects that are not essential to our setting.

**Definition C.1** (KL property; (Attouch et al., 2010, Definition 3.1), (Boţ et al., 2016, Definition 1)). A $C^1$ function $f : \mathbb{R}^n \to \mathbb{R}$ is said to have the KL property (or to be a KL function) at a point $\bar{x} \in \mathbb{R}^n$ if there exists $0 < \eta \leq +\infty$, a neighborhood $U \subset \mathbb{R}^n$ of $\bar{x}$, and a continuous concave function $\varphi : [0, \eta) \to \mathbb{R}$ such that

1. $\varphi(0) = 0$;

2. $\varphi \geq 0$;

3. $\varphi$ is $C^1$ on $(0, \eta)$;

4. for all $s \in (0, \eta)$, $\varphi'(s) > 0$;

5. for all $x \in U \cap \{x \mid f(\bar{x}) < f(x) < f(\bar{x}) + \eta\}$, the Kurdyka-Łojaciewicz inequality holds:

$$\|\nabla f(x)\|_2 \geq \frac{1}{\varphi'(f(x) - f(\bar{x}))}. \tag{C.25}$$

It can be shown that any $f \in C^1(\mathbb{R}^n)$ has the KL property at every point $\bar{x}$ that is not a critical point of $f$ (Attouch et al., 2010, Remark 3.2(b)). At critical points $\bar{x}$ of such a function $f$, the KL inequality is a kind of nondegeneracy condition on $f$. This can be intuited from the (common) example of functions $f$ witnessing the KL property via the function $\varphi(s) = cs^{1/2}$, where $c > 0$ is a constant: in this case, (C.25) gives a lower bound on the squared magnitude of the gradient near to a critical point in terms of the values of the function $f$. Every Morse function satisfies such an instantiation of the KL property: see (Attouch et al., 2010, beginning of §4). In general, the KL property need not require excessive regularity of $f$; we only state it as such because of the special structure in our setting (c.f. Lemma C.7).

The KL property is useful in spite of the unwieldy Definition C.1 because broad classes of functions can be shown to have the KL property, and these classes of functions satisfy a convenient calculus that allows one to construct new KL functions from simpler primitives (somewhat analogous to the situation with convex functions in convex analysis). The class of such functions we will use in this work are *functions definable in an o-minimal structure*. We refer to (Attouch et al., 2010, Definition 4.1) for the precise definition of this concept from model theory—for our purposes, it will suffice to understand that an o-minimal structure is a collection of subsets of $\mathbb{R}^n$, for each $n \in \mathbb{N}$, which satisfy certain algebraic properties and are called *definable sets*, or *definable* for short, and that a definable function is one whose graph is contained in this collection (i.e., its graph is a definable set)[21]—and instead state several useful properties of functions definable in an o-minimal structure that will suffice to complete our proof. We note that Attouch et al. (2010, §4) give an excellent readable optimization-motivated overview of these properties in greater generality and depth, as do Ji & Telgarsky (2020), the latter moreover in a deep learning context.

**Proposition C.11.** *The following properties of o-minimal structures and functions definable in an o-minimal structure hold.*

1. *(Attouch et al., 2010, Theorem 4.1) If $f : \mathbb{R}^n \to \mathbb{R}$ is continuously differentiable and definable in an o-minimal structure, then it has the KL property (Definition C.1) at every $x \in \mathbb{R}^n$.*

2. *(Wilkie) There is an o-minimal structure in which the following functions are definable: the exponential function $x \mapsto \exp(x)$ (for $x \in \mathbb{R}$), and polynomials of arbitrary degree in $n$ real variables $x_1, \ldots, x_n$ (van den Dries & Miller, 1994), (Van den Dries, 1998, Corollary 2.11).*

3. *(van den Dries & Miller, 1996, §B, B.7(3)) If $f : \mathbb{R}^n \to \mathbb{R}$ is $C^1$ and definable in an o-minimal structure, then its gradient $\mathbf{x} \mapsto \nabla f(\mathbf{x})$ is definable.*

---

[21]Here and below, the use of the article "an" is intentional—there may be multiple o-minimal structures in which a function is definable, and in some applications it is important to distinguish between them. For our purposes in this work, this distinction is immaterial: we will be content to simply work in a 'maximal' o-minimal structure that contains all functions we will need.

4. ((van den Dries & Miller, 1996, §B, B.4)) A function $f : \mathbb{R}^n \to \mathbb{R}^m$ with $f = (f_1, \dots, f_m)$ is definable in an o-minimal structure if and only if each $f_i$ is definable (in the same structure).

5. (Loi, 2010, Theorem 2.3(iii)) The composition of functions definable in an o-minimal structure is definable.

6. (Loi, 2010, Theorem 2.3(ii)) If $f : \mathbb{R}^n \to \mathbb{R}^n$ is definable in an o-minimal structure, then its image $\mathrm{im}(f)$ is definable; if moreover $f$ is invertible on its image, then its inverse $f^{-1}$ (defined on $\mathrm{im}(f)$) is definable.[22]

These properties represent a rather minimal set that are sufficient for our purposes. To illustrate how to use them to deduce slightly more accessible (and useful) corollaries, we provide the following result on definable functions that will be used repeatedly in our proofs.

**Corollary C.12.** *If functions $f_1 : \mathbb{R}^n \to \mathbb{R}$ and $f_2 : \mathbb{R}^n \to \mathbb{R}$ are definable in the o-minimal structure asserted by Proposition C.11, Property 2, then $f_1 + f_2$ is definable in the same structure.*

*Proof.* By Proposition C.11, Property 2, the function $g = x_1 + x_2$, for $x_1, x_2 \in \mathbb{R}$, is definable since it is a polynomial. By Proposition C.11, Property 4, $(f_1, f_2)$ is definable. So, by Proposition C.11, Property 5, $f_1 + f_2$ is definable since it is the composition of $(f_1, f_2)$ and $g$. □

Using these properties, we prove that the regularizer associated to an LPN with $0 < \alpha < 1$ is a KL function. For concision, we call a function $f$ "definable" if it is definable in the o-minimal structure whose existence is asserted by Proposition C.11, Property 2.

**Lemma C.13** (Definability and KL property of the prior $R_\theta$)**.** *Suppose $f_\theta$ is an LPN constructed following the recipe in Proposition 3.1, with softplus activations $\sigma(x) = (1/\beta) \log(1 + \exp(\beta x))$, where $\beta > 0$ is an arbitrary constant, and with strong convexity weight $0 < \alpha < 1$. Then there is a function $R_\theta : \mathbb{R}^n \to \mathbb{R}$ such that $f_\theta = \mathrm{prox}_{R_\theta}$, and $R_\theta$ is definable and has the KL property (Definition C.1) at every point of $\mathbb{R}^n$. Moreover, this $R_\theta$ can be chosen to simultaneously satisfy the conclusions of Lemma C.7.*

*Proof.* We appeal to Lemma C.7, given that $0 < \alpha < 1$, to obtain that a function $R_\theta : \mathbb{R}^n \to \mathbb{R}$ such that $f_\theta = \mathrm{prox}_{R_\theta}$ exists, and moreover that this function $R_\theta$ is $C^1$ and satisfies the relation (C.22). The idea of the proof is to use (C.22) to show that $R_\theta$ is definable in an o-minimal structure, then appeal to Proposition C.11, Property 1 to obtain that $R_\theta$ has the KL property at every point of $\mathbb{R}^n$. By Corollary C.12, to show that $R_\theta$ is definable, it suffices to show that four functions appearing in the representation (C.22) are definable: $\mathbf{x} \mapsto \langle f_\theta^{-1}(\mathbf{x}), \nabla \psi_\theta(f_\theta^{-1}(\mathbf{x})) \rangle$, $\mathbf{x} \mapsto \|f_\theta^{-1}(\mathbf{x})\|_2^2$, $\mathbf{x} \mapsto \|\nabla \psi_\theta(f_\theta^{-1}(\mathbf{x}))\|_2^2$, and $\mathbf{x} \mapsto \psi_\theta(f_\theta^{-1}(\mathbf{x}))$. We can reduce further. By Proposition C.11, Property 2, the squared norm $\mathbf{x} \mapsto \|\mathbf{x}\|_2^2$ is definable, and Proposition C.11, Property 5, namely that finite compositions of definable functions are definable, it suffices to show that the arguments of the norms appearing amongst these four functions are definable. Similarly, by Proposition C.11, Properties 2, 4, and 5, for the inner product term appearing amongst these four functions, it suffices to show that the two individual arguments of the inner product are definable. It then follows by one additional application of Proposition C.11, Property 5 that we need only argue that the following three functions are definable: $\mathbf{x} \mapsto \psi_\theta(\mathbf{x})$, $\mathbf{x} \mapsto f_\theta^{-1}(\mathbf{x})$, and $\mathbf{x} \mapsto \nabla \psi_\theta(\mathbf{x})$. Finally, by Proposition C.11, Property 6 and Lemma C.7, which asserts that $f_\theta$ is invertible on $\mathbb{R}^n$ with range equal to $\mathbb{R}^n$, it suffices merely to show that $\mathbf{x} \mapsto f_\theta(\mathbf{x})$ is definable to obtain that its inverse is definable.

To complete the proof, we argue in sequence below that each of these three functions are definable.

**ICNN $\psi_\theta$.** The ICNN $\psi_\theta$ is defined inductively in Proposition 3.1, and by our hypotheses, the activation function $g(\mathbf{x})$ is an elementwise application of the softplus activation with parameter $\beta > 0$: with a minor abuse of notation, this function is

$$g(x) = (1/\beta) \log(1 + \exp(\beta x)). \tag{C.26}$$

---

[22] The proof of this property follows readily from the cited result and the preceding properties by noticing that if one defines a map $F : \mathbb{R}^n \times \mathbb{R}^n \to \mathbb{R}^n \times \mathbb{R}^n$ by $F(x, y) = (f(y), f(x))$, then $\mathrm{gr}(f^{-1}) = F^{-1}(\mathrm{gr}(f))$, which expresses the graph of $f^{-1}$ as the inverse image of a definable set by a definable function.

This (scalar) activation function is definable. To see this, by Proposition C.11, Property 2, we have that $x \mapsto \exp x$ is definable, and by Proposition C.11, Property 6, we have that $x \mapsto \log x$ is definable. It then follows from Proposition C.11, Properties 2 & 5 that $g$ is definable, as a finite composition of definable functions. We then conclude from Proposition C.11, Property 4 that the elementwise activation $\mathbf{x} \mapsto g(\mathbf{x})$ is definable. Now, by the definition of $\psi_\theta$ in Proposition 3.1, we have that $\mathbf{z}_1$ is definable as a composition of definable functions (Proposition C.11, Properties 2 and 5), and arguing inductively, we have by the same reasoning that $\mathbf{z}_k$ is definable for each $k = 2, \ldots, K$. Because $\psi_\theta = \mathbf{w}^T \mathbf{z}_K + b$ is an affine function of $\mathbf{z}_K$, one additional application of Proposition C.11, Properties 2 and 5 establishes that $\psi_\theta$ is definable.

**ICNN gradient $\nabla \psi_\theta$.** We conclude this immediately from Proposition C.11, Property 3 and the definability of $\psi_\theta$, since $\psi_\theta$ is $C^2$ (because $\psi_\theta$ is composed of affine maps and $C^2$ activations).

**LPN $f_\theta$.** Recall that $f_\theta(\mathbf{x}) = \nabla \psi_\theta(\mathbf{x}) + \alpha \mathbf{x}$. Because we have shown above that $\nabla \psi_\theta$ is definable, using Proposition C.11, Properties 2 and 5 once more, we conclude that $f_\theta$ is definable.

**Concluding.** By our preceding reductions, we have shown that $R_\theta$ is definable. We conclude from Proposition C.11, Property 1 that $R_\theta$ has the KL property at every point of $\mathbb{R}^n$.

$\square$

*Remark.* Note that a byproduct of the proof of Lemma C.13 is that all constituent functions of the LPN are also definable in an o-minimal structure. This fact may be of interest for future work extending Lemma C.13 to priors associated with novel LPN architectures.

# D ALGORITHMS

## D.1 ALGORITHM FOR LOG-PRIOR EVALUATION

---

**Algorithm 2** Log-prior evaluation for LPN

---

**Input:** Learned proximal network $f_\theta(\cdot)$, $\psi_\theta(\cdot)$ that satisfies $f_\theta = \nabla \psi_\theta$, query point $\mathbf{x}$
1: Find $\mathbf{y}$ such that $f_\theta(\mathbf{y}) = \mathbf{x}$, by solving $\min_{\mathbf{y}} \psi_\theta(\mathbf{y}; \alpha) - \langle \mathbf{x}, \mathbf{y} \rangle$ or $\min_{\mathbf{y}} \|f_\theta(\mathbf{y}) - \mathbf{x}\|_2^2$
2: $R \leftarrow \langle \mathbf{y}, \mathbf{x} \rangle - \frac{1}{2} \|\mathbf{x}\|^2 - \psi_\theta(\mathbf{y})$
**Output:** $R$                  ▷ The learned log-prior (i.e., regularizer function) at $\mathbf{x}$

---

## D.2 ALGORITHM FOR LPN TRAINING

---

**Algorithm 3** Training the LPN with proximal matching loss

---

**Input:** Training dataset $\mathcal{D}$, initial LPN parameter $\theta$, loss schedule $\gamma(\cdot)$, noise standard deviation $\sigma$, number of iterations $K$, network optimizer $\text{Optm}(\cdot, \cdot)$
1: $k \leftarrow 0$
2: **repeat**
3:      Sample $\mathbf{x} \sim \mathcal{D}$, $\boldsymbol{\varepsilon} \sim \mathcal{N}(0, \mathbf{I})$
4:      $\mathbf{y} \leftarrow \mathbf{x} + \sigma \boldsymbol{\varepsilon}$
5:      $\mathcal{L}_{PM} \leftarrow m_{\gamma(k)}(\|f_\theta(\mathbf{y}) - \mathbf{x}\|_2)$
6:      $\theta \leftarrow \text{Optm}(\theta, \nabla_\theta \mathcal{L}_{PM})$                ▷ Update network parameters
7:      $k \leftarrow k + 1$
8: **until** $k = K$
**Output:** $\theta$                           ▷ Trained LPN

---

## D.3 ALGORITHM FOR USING LPN WITH PNP-PGD TO SOLVE INVERSE PROBLEMS

---

**Algorithm 4** Solving inverse problems with LPN and PnP-PGD

---

**Input:** Trained LPN $f_\theta$, measurement operator $A$, measurement $\mathbf{y}$, data fidelity function $h(\mathbf{x}) = \frac{1}{2}\|\mathbf{y} - A(\mathbf{x})\|_2^2$, initial estimation $\mathbf{x}_0$, step size $\eta$, number of iterations $K$
 1: **for** $k = 0$ to $K - 1$ **do**
 2:     $\mathbf{x}_{k+1} \leftarrow f_\theta\left(\mathbf{x}_k - \eta\nabla h(\mathbf{x}_k)\right)$
 3: **end for**
**Output:** $\mathbf{x}_K$

---

# E  EXPERIMENTAL DETAILS

## E.1  DETAILS OF LAPLACIAN EXPERIMENT

The LPN architecture contains four linear layers and 50 hidden neurons at each layer, with $\beta = 10$ in softplus activation. The LPN is trained by Gaussian noise with $\sigma = 1$, Adam optimizer (Kingma & Ba, 2014) and batch size of 2000. For either $\ell_2$ or $\ell_1$ loss, the model is trained for a total of $20k$ iterations, including $10k$ iterations with learning rate $lr = 1e - 3$, and another $10k$ with $lr = 1e - 4$. For the proximal matching loss, we initialize the model from the $\ell_1$ checkpoint and train according to the schedule in Table 3. To enforce nonnegative weights of LPN, weight clipping is applied during training, projecting the negative weights to zero at each iteration.

| Number of iterations | $\gamma$ in $\mathcal{L}_{\mathcal{PM}}$ | Learning rate |
|---|---|---|
| $2k$ | 0.5 | $1e - 3$ |
| $2k$ | 0.5 | $1e - 4$ |
| $4k$ | 0.4 | $1e - 4$ |
| $4k$ | 0.3 | $1e - 4$ |
| $4k$ | 0.2 | $1e - 5$ |
| $4k$ | 0.1 | $1e - 5$ |
| $4k$ | 0.1 | $1e - 6$ |

Table 3: The schedule for proximal matching training of LPN in the Laplacian experiment.

## E.2  DETAILS OF MNIST EXPERIMENT

The LPN architecture is implemented with four convolution layers and 64 hidden neurons at each layer, with $\alpha = 0.01$ and softplus $\beta = 10$. The model is trained on the MNIST training set containing $50k$ images, with Gaussian noise with standard deviation $\sigma = 0.1$ and batch size of 200. The LPN is first trained by $\ell_1$ loss for $20k$ iterations; and then by the proximal matching loss for $20k$ iterations, with $\gamma$ initialized at $0.64 * 28 = 17.92$ and halved every $5k$ iterations. The learned prior is evaluated on 100 MNIST test images. Conjugate gradient is used to solve the convex inversion problem: $\min_{\mathbf{y}} \psi_\theta(\mathbf{y}) - \langle \mathbf{x}, \mathbf{y} \rangle$ in prior evaluation.

## E.3  DETAILS OF CELEBA EXPERIMENT

We center-crop CelebA images from $178 \times 218$ to $128 \times 128$, and normalized the intensities to $[0, 1]$. Since CelebA images are larger and more complex than MNIST, we use a deeper and wider network. The LPN architecture includes 7 convolution layers and 256 hidden neurons per layer, with $\alpha = 1e - 6$ and $\beta = 100$. For LPN training, we train two separate models with two levels of training noise: $\sigma = 0.05$ and $0.1$. When applied for deblurring, the best model is selected for each blurring degree ($\sigma_{blur}$) and measurement noise level ($\sigma_{noise}$). We pretrain the network with $\ell_1$ loss for $20k$ iterations with $lr = 1e - 3$. Then, we train the LPN with proximal matching loss $\mathcal{L}_{\mathcal{PM}}$ for $20k$ iterations using $lr = 1e - 4$, with the schedule of $\gamma$ similar to MNIST: initialized at $0.64 \times \sqrt{128 \times 128 \times 3} \approx 142$, and multiplied by $0.5$ every $5k$ iterations. A batch size of 64

is used during training. We observed that initializing the respective weights to be nonnegative, by initializing them according to a Gaussian distribution and then taking the exponential, helped the training converge faster. Therefore, we applied such initialization in the experiments on CelebA and Mayo-CT. The same weight clipping as in Appendix E.1 is applied to ensure the weights stay nonnegative throughout training. The training time of LPN on the CelebA dataset is about 6 hours on a NVIDIA RTX A5000 GPU.

**PnP algorithm and comparison methods**    We use PnP-ADMM to perform deblurring on CelebA for BM3D, DnCNN, and our LPN (see Algorithm 1). We implement the PnP-ADMM algorithm using the SCICO package (Balke et al., 2022). We implement DnCNN (Zhang et al., 2017a) using their public code [23]. We implement the GS denoiser, Prox-DRUNet (Hurault et al., 2022b), using their public code[24] and follow their paper to use the Douglas–Rachford splitting (DRS) algorithm when solving inverse problem, which performs the best with Prox-DRUNet based on their paper. Both DnCNN and Prox-DRUNet are trained on CelebA.

### E.4    DETAILS OF MAYO-CT EXPERIMENT

We use the public dataset from Mayo-Clinic for the low-dose CT grand challenge (Mayo-CT) (Mc-Collough, 2016), which contains abdominal CT scans from 10 patients and a total of 2378 images of size $512 \times 512$. Following (Lunz et al., 2018), we use 128 images for testing and leave the rest for training. The LPN architecture contains 7 convolution layers with 256 hidden neurons per layer, with $\alpha = 1e - 6$ and $\beta = 100$. During training, we randomly crop the images to patches of size $128 \times 128$. At test time, LPN is applied to the whole image by sliding windows of the patch size with stride size of 64. The training procedure of LPN is the same as in CelebA, except that $\gamma$ in proximal matching loss is initialized to $0.64 \times \sqrt{128 \times 128} \approx 82$. As in the CelebA experiment, we use LPN with PnP-ADMM for solving inverse problems.

**Sparse-view CT**    Following Lunz et al. (2018), we simulate CT sinograms using a parallel-beam geometry with 200 angles and 400 detectors. The angles are uniformly spaced between $-90°$ and $90°$. White Gaussian noise with standard deviation $\sigma = 2.0$ is added to the sinogram data to simulate noise in measurement. We implement AR in PyTroch based on its public TensorFlow code[25]; for UAR, we use the publicly available code and model weights [26].

**Compressed sensing**    For compressed sensing, we implement the random Gaussian sampling matrix following Jalal et al. (2021b), and add noise of $\sigma = 0.001$ to the measurements. The Wavelet-based sparse recovery method for compressed sensing minimizes the object $\frac{1}{2}\|\mathbf{y} - A\mathbf{x}\|_2^2 + \lambda\|W\mathbf{x}\|_1$, where $A$ is the sensing matrix and $W$ is a suitable Wavelet transform. We select the "db4" Wavelet and $\lambda = 0.01$. To solve the minimization problem in Wavelet-based approach, we use proximal gradient descent with a step size of 0.5, stopping criterion $\|\mathbf{x}_{k+1} - \mathbf{x}_k\|_1 < 1e - 4$, and maximum number of iterations $= 1000$.

## F    DISCUSSIONS

### F.1    OTHER WAYS TO PARAMETERIZE GRADIENTS OF CONVEX FUNCTIONS VIA NEURAL NETWORKS

Input convex gradient networks (ICGN) (Richter-Powell et al., 2021) provide another way to parameterize gradients of convex functions. The model performs line integral over Positive Semi-Definite (PSD) Hessian matrices, where the Hessians are implicitly parameterized by the Gram product of Jacobians of neural networks, hence guaranteed to be PSD. However, this approach only permits single-layer networks in order to satisfy a crucial PDE condition in its formulation (Richter-Powell et al., 2021), significantly limiting the representation capacity. Furthermore, the evaluation of the

---

[23]https://github.com/cszn/KAIR
[24]https://github.com/samuro95/Prox-PnP
[25]https://github.com/lunz-s/DeepAdverserialRegulariser.
[26]https://github.com/Subhadip-1/unrolling_meets_data_driven_regularization.

convex function is less straightforward than ICNN, which is an essential step in prior evaluation for LPN (see Section 3). We therefore adopt the differentiation-based parameterization in this work and leave the exploration of other possibilities to future research.

# G    ADDITIONAL RESULTS

## G.1    LEARNING SOFT-THRESHOLDING FROM LAPLACIAN DISTRIBUTION

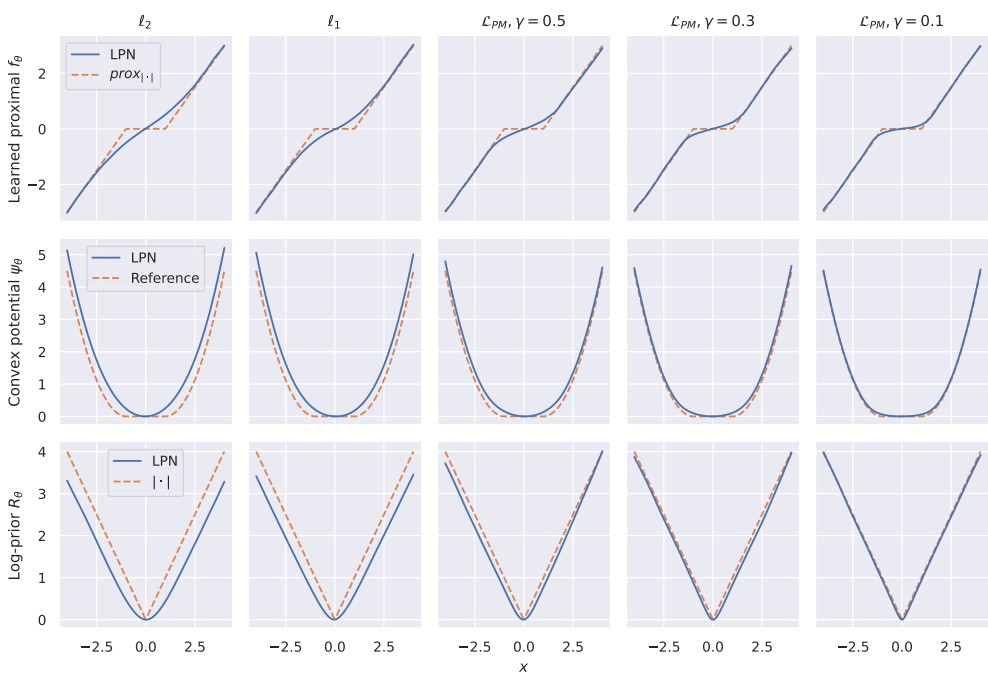

Figure 6: The proximal operator $f_\theta$, convex potential $\psi_\theta$, and log-prior $R_\theta$ learned by LPN via different losses: the square $\ell_2$ loss, $\ell_1$ loss, and the proposed proximal matching loss $\mathcal{L}_{PM}$ with different $\gamma \in \{0.5, 0.3, 0.1\}$. The ground-truth data distribution is the Laplacian $p(x) = \frac{1}{2}\exp(-|x|)$, with log-prior $-\log p(x) = |x| - \log(\frac{1}{2})$. With proximal matching loss, the learned proximal $f_\theta$ and log-prior $R_\theta$ progressively approach their ground-truth, $\mathrm{prox}_{|\cdot|}$ and $|\cdot|$ respectively, as $\gamma$ shrinks from 0.5 to 0.1.

## G.2    LEARNING A PRIOR FOR MNIST - IMAGE BLUR

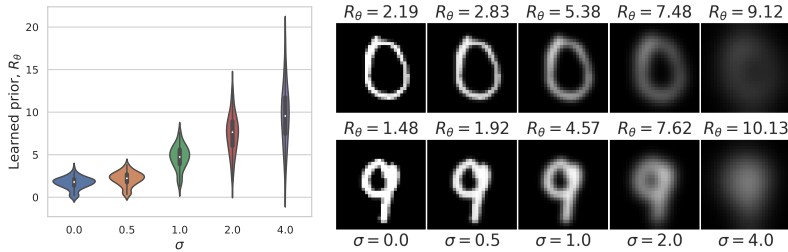

Figure 7: The log-prior $R_\theta$ learned by LPN on MNIST, evaluated at images blurred by Gaussian kernel with increasing standard deviation $\sigma$. Left: the prior over 100 test images. Right: the prior at individual examples.

Besides perturbing the images by Gaussian noise and convex combination in Section 5.1, we also evaluate the prior of LPN at blurry images, with results shown in Figure 7. Again, the prior increases as the image becomes blurrier, coinciding with the distribution of the hand-written digits in MNIST.

### G.3  SOLVING INVERSE PROBLEMS USING LPN WITH PNP-PGD

Besides PnP-ADMM, we also test LPN's performance for solving inverse problems using PnP-PGD (proximal gradient descent). Table 4 shows the numerical results for deblurring CelebA images: PGD is slightly less performant than ADMM in terms of PSNR.

Table 4: Numerical results for CelebA deblurring using LPN with PnP-PGD and PnP-ADMM, averaged over 20 test images.

| METHOD | $\sigma_{blur} = 1, \sigma_{noise} = .02$ | | $\sigma_{blur} = 1, \sigma_{noise} = .04$ | |
| --- | --- | --- | --- | --- |
| | PSNR($\uparrow$) | SSIM($\uparrow$) | PSNR | SSIM |
| LPN with PnP-PGD | $32.7 \pm 2.9$ | $.92 \pm .03$ | $31.2 \pm 2.5$ | $.89 \pm .04$ |
| LPN with PnP-ADMM | $33.0 \pm 2.9$ | $.92 \pm .03$ | $31.3 \pm 2.3$ | $.89 \pm .03$ |

