# OpenReview forum: "What's in a Prior? Learned Proximal Networks for Inverse Problems"
_ICLR.cc/2024/Conference — ICLR 2024 poster_

### Official Review · Reviewer_XDuj · 2023-10-29

**Soundness:** 2 fair
**Presentation:** 3 good
**Contribution:** 2 fair
**Rating:** 5
**Confidence:** 5

**Summary:**

In this work, the authors propose a way to learn a denoising map, called LPN, which is exactly the proximal operator of a scalar function which should approximate the true log image prior. They make use of the characterization of nonconvex proximity operators as gradient of convex functions from (Gribonva & Nikolova, 2020) and parametrize their denoiser as the gradient of (an ICNN.+ a quadratic term). They also suggest a specific loss, called "proximal matching loss" such that, when the denoiser is trained to denoise Gaussian noise with this loss, it should approximate the prox of the true log image prior. They also propose a convergence proof of the PnP-PGD algorithm with plugged LPN denoiser and experiment on different datasets and inverse problems.

**Strengths:**

- The paper is generally well presented and well written.
- The idea to parameterize a denoiser as the gradient of an ICNNs is new in the PnP literature.
- The most interesting contribution of the paper is for me the proximal matching loss.
- The verification of the nonconvexity of the learned prior is also interesting.

**Weaknesses:**

Here are several potential issues that I spotted while reviewing the paper.

Major weaknesses :
- The end the proof from Theorem 4.1 is I think not true, and the presented result not valid. Indeed, the definition of subdifferential subdiferential for convergence of nonconvex PGD is not the usual subdifferential but the limiting subdifferential. With this notion, the first equivalence from C.48 is not true when $\phi_\theta$ is nonconvex, and only the implication holds. Therefore, the algorithm can not converge to a fixed-point as presented, but only to a critical point of $h + \eta^{-1}\phi_\theta$.
- The targeted objective function $h + \eta^{-1}\phi_\theta$ contains the stepsize (which is then a regularization parameter). This is uncommon in optimization and not desirable, as a change in the stepsize affects the objective function and the obtained result. Moreover, the stepsize (and thus the regularization parameter) must be bounded by $1/L$, is this limiting in practice and does it impact the performance ?
- The algorithm analyzed for convergence is just the classical Proximal Gradient Descent (PGD) (or Forward-Backward).
Why do you use the convergence analysis from (Bot et. al, 2016) which is specific for an accelerated version of this scheme. You could use directly the (most commonly used) convergence result from (Attouch et. al, 2013) of PGD in the nonconvex setting.
- "Definable" is an important notion that should be defined. Moreover, the stability properties of Definable functions ( by sum, composition, inverse, derivative) are extensively used without referencing the proof of these results. Same comment for the fact that the exponential is definable.
- You experiment with PnP-ADMM when the presented theoretical result is with PGD. If this is correct, this is a major issue. Indeed, the theoretical convergence results of ADMM in the convex setting are not the same as the ones for PGD.

Minor weaknesses :
- Lemma C.2 :  I think that $f_\theta$ is only invertible on $Im(f_\theta)$ and $\phi_\theta$ is only differentiable on $Im(f_\theta)$. If this is really true on the whole space, can you explain why ?
- Section 1 : "for almost any inverse problem, a proximal step for the regularization function is always present" I do not understand this sentence and what you mean by "present".
- Section 1 : In (Romano et. al, 2017), there is no prox involved.
- Section 2 : Contrary to what you seem to explain, PnP is really not limited to ADMM !!
- Section 3 : Parameterizing a network as the gradient of an ICNN as already been proposed in the literature, in particular in the Optimal Transport community. These works should be cited as well.
- Section 3 : You explain that the chose parameterization is more general and universal than (Hurault et. al, 2022). I am very doubtful about this statement, given the fact that the ICNN parameterization is very (and I think way more) constraining. A way to support your affirmation would be to compare the performance with both denoisers for denoising and PnP restoration.
- In most PnP algorithm, the Gaussian noise parameter $\sigma$ on which the denoiser is trained acts as a regularization parameter. Here, you do not mention this parameter, and you do not explain how it is chosen.

These weaknesses explain my low score, and I am ready to raise my score if the authors answer these limits.

**Questions:**

See above.

---

> ### Author Response · Authors · 2023-11-21
> **Rebuttal**
>
> We sincerely thank the reviewer for their careful review of our work. We have addressed all of your questions and comments, and please don't hesitate to let us know if there are any pending matters.
>
> 1. > The end the proof from Theorem 4.1 is I think not true...
>
> Thank you for pointing this out. We have corrected the proof. The argument that the algorithm converges to a fixed-point still holds, due to the continuity of LPN. Specifically, since both $f_\theta$ and $\nabla h$ are continuous, the mapping from $x_k$ to $x_{k+1}$, $x_{k+1} = f_\theta(x_k - \eta \nabla h(x_k))$ is continuous. Then, given that the iterative algorithm converges, and that the mapping from $x_k$ to $x_{k+1}$ is continuous, the algorithm must converge to a fixed point. Please see the last paragraph of the proof of Theorem 4.1 and Lemma C.2 for the updated proof.
>
> 2. > The targeted objective function h + eta^-1 phi contains the step size ...
>
> Thank you for pointing this out. Indeed, this 1/eta weighing of the regularizer might seem strange, but we would like to argue that in our setting it is reasonable and may even be useful.
>
> First, we would like to comment on the cause of $1/\eta$ weighing. For an objective $f(x) + g(x)$, the proximal gradient descent (PGD) algorithm applies ${prox}_{\eta g}(x-\eta \nabla f(x))$
>
> at each iteration. Note that the proximal operator $prox_{\eta g}$ needs to be scaled appropriately by $\eta$.
> If we follow this strictly, the objective will not change with eta. However, in our case, the proximal operator is fixed at $prox_{\phi}$, and we did not scale it accordingly by $\eta$, which requires evaluation of $prox_{\eta \phi}$. Thus our calculation has a subtle difference from classic PGD, creating the $1/\eta$ weighing in the objective.
>
> Second, this weighing may be useful, as it provides a way to change the regularization weight 	without retraining LPN. Indeed, eta can be treated as a knob to control the regularization weight: smaller eta means higher weight on the regularization. However, as the reviewer points out, eta being too large would cause the algorithm to diverge (note that in practice $\eta > 1/L$ does not necessarily mean the algorithm will diverge, i.e. the actual upper bound may be larger than $1/L$), so this tuning is not full-spectrum. We do not see how this necessarily impacts the performance, as there exist other ways to tune the regularization weight, e.g., via Gaussian noise level at training.
>
> 3. > The algorithm analyzed for convergence is just the classical Proximal Gradient Descent (PGD) (or Forward-Backward) ...
>
> We used the result from Bot et al. (2016) because it includes coercivity of the objective as a hypothesis, whereas the main result in Attouch et al. (2013), i.e. Theorem 5.1, writes the result instead in the form “if the iterate sequence is bounded, then it converges”. Although it is straightforward to argue that coercivity of the objective implies iterate boundedness, we assumed that for a ML venue, it would be easier for readers to parse the proof if we appealed to a result with hypotheses that match our setting. In the main body of the submission, we made a note of the precedence of Attouch et al. (2013) via a citation in the “Convergence Guarantees in Plug-and-Play Frameworks” paragraph at the top of page 6 in the main body. If you want us to change the proof to appeal to Attouch et al. (2013) and argue boundedness of the iterate sequence using coercivity instead of appealing to Bot et al. (2016), we can do so.
>
> 4. > "Definable" is an important notion that should be defined. Moreover, the stability properties of Definable functions ...
>
> We have added a new section to the appendices, Section C.4.1, which contains what we believe is a far more complete and readable overview of the relevant aspects of the KL property that we need to prove our convergence results. The submission originally referenced Attouch et al. 2010 for these facts, which seems to us to be a standard and sufficient reference here (allowing one to avoid tedious verifications), but the revision has more precise references to proofs and spells out all details in the verifications. On this note, we would like to note that you are asking for what seems to us to be a very high standard of rigor for a machine learning conference submission, and we hope you will acknowledge our contribution in meeting it. Please let us know if you have further feedback in this connection.

---

> > ### Author Response · Authors · 2023-11-21
> > **Rebuttal continued**
> >
> > 5. > You experiment with PnP-ADMM when the presented theoretical result is with PGD. If this is correct, this is a major issue. Indeed, the theoretical convergence results of ADMM in the convex setting are not the same as the ones for PGD.
> >
> > Our intention of presenting a convergence analysis on LPN with PGD is to demonstrate an example of the improved theoretical guarantee that LPN could provide by parameterizing an exact proximal operator. We did not intend to imply in any way that the LPN with ADMM approach used in the Experiments section has been proved to converge. As we have emphasized in our paper, LPN can be plugged into many optimization algorithms-not limited to PGD or ADMM, and we have made it very clear that LPN with PGD and LPN with ADMM are two different approaches and presented them as two separate algorithms (Algorithm 1 and Algorithm 4). We do not think this would cause any confusion.
> >
> > In our initial experiments, we observed that LPN with PGD is slightly less performant than LPN with ADMM when solving inverse problems, and therefore adopted LPN with ADMM in the Experiment section. To address the reviewer’s concern, we have now added the experimental results for LPN with PGD for CelebA deblurring in Appendix G.2, Table 4 to illustrate this point (see the Table below), and to demonstrate the viability of plugging LPN into different optimization algorithms, PGD in particular, for solving inverse problems, as claimed in the paper.
> >
> > Meanwhile, we agree that having a demonstration of an additional PnP scheme where a LPN convergence guarantee can be readily obtained would further improve the submission. We are currently working on a proof of convergence for PnP-ADMM with LPNs, and hope to upload it here before the discussion deadline. Since our setting is similar to that of Hurault et al (2022), we are working towards adapting their analysis to our setting (alas, they treat a nonstandard variant of PnP-ADMM, which is demanding careful work). Applying the standard convergence analysis for nonconvex ADMM in the literature to the standard PnP-ADMM, namely Themelis and Patrinos (2020), runs into technical challenges because the order of updates in Themelis and Patrinos’s instantiation of ADMM is reversed from that of PnP-ADMM.
> >
> > 6. > Lemma C.2 : I think that f_theta is only invertible on Im(f_theta) and phi_theta is only differentiable on Im(f_theta). If this is really true on the whole space, can you explain why?
> >
> > We argue that $f_\theta$ is surjective, i.e. $Im(f_\theta) = R^n$, when $\alpha > 0$, using tools from monotone operator theory. Please see the proof of Lemma C.3 for details.
> >
> > There is a more direct way to see this, however, which we also transcribe here due to its conceptual simplicity. It relates to our methodology for inverting the LPN f in the “Recovering the prior from its proximal” paragraph in Section 3. Consider the minimization problem $\min_y \psi_\theta(y; \alpha) - <x, y>$. Note that when $\alpha > 0$ the problem is strongly convex, and hence has a unique minimizer. Furthermore, the minimizer $y^*$ satisfies $\nabla \psi_\theta(y^*; \alpha) = x$ (the first-order optimality condition). Thus, for any $x$, there is a unique $y^*$ such that $\nabla \psi_\theta(y^*; \alpha) = x$. Finally, since $f_\theta(y) = \nabla \psi_\theta(y; \alpha)$, the surjectivity (and bijectivity) is established. This argument is included as a footnote in the proof of Lemma C.3.
> >
> > 7.> "for almost any inverse problem, a proximal step for the regularization function is always present" I do not understand this sentence and what you mean by "present".
> >
> > Sorry about the confusion. We meant that “a proximal step of the regularizer is used in many solvers for inverse problems, for example, proximal gradient descent, ADMM, etc.” To make it more precise and less confusing, we have modified this sentence to  "Many iterative solvers for inverse problems incorporate the application of the proximal operator for the regularizer."
> >
> > 8. > Section 1 : In (Romano et. al, 2017), there is no prox involved.
> >
> > Thank you for mentioning this. In the original manuscript, we referred to (Romano et al. 2017) as an example of methods that utilize denoisers to solve general inverse problems, not using prox. Sorry for the confusion. In the revised version, we have updated our introduction to clarify the language around these points (see below). Please let us know if you have further objections.
> >
> > 9. >. Section 2 : Contrary to what you seem to explain, PnP is really not limited to ADMM !!
> >
> > Indeed, we highlight throughout  our work that PnP (and our methodology) can be combined with a variety of different optimization frameworks without issue. We have updated the text in Section 2 that unintentionally suggested this – please let us know if you have any further objections.

---

> > > ### Author Response · Authors · 2023-11-21
> > > **Rebuttal continued 2**
> > >
> > > 10. > Parameterizing a network as the gradient of an ICNN has already been proposed in the literature, in particular in the Optimal Transport community. These works should be cited as well.
> > >
> > > Thank you for bringing this up. We are aware of this connection – our methodology benefits from the proposal of Huang et al. (2021), working in an OT-type setting, to make the ICNN strongly convex, and we have cited this work throughout our submission (please see, for example, the top of page 4, in Section 3). After your comment, we dug up the work https://arxiv.org/abs/1908.10962, which was the earliest reference we could find that exploits the connection between ICNN and OT maps. We have cited this work in context. If you have other references in mind, please let us know.
> > >
> > > 11. > Section 3 : You explain that the chosen parameterization is more general and universal than (Hurault et. al, 2022). I am very doubtful about this statement, given the fact that the ICNN parameterization is very (and I think way more) constraining. A way to support your affirmation would be to compare the performance with both denoisers for denoising and PnP restoration.
> > >
> > > **On the generality and limitations of different parameterizations**
> > >
> > > We think there are two issues here: the issue of parameterizing a proximal operator (which the claims at the top of page 4 in the submission were intended to refer to) versus the parameterization of a neural network that implements the former parameterization. For parameterizing a proximal operator, note that in our framework (abstractly), we simply express $f = \nabla \psi$, where \psi is $C^1$ and convex; by Gribonval and Nikolova’s characterization, such a form can represent any continuous proximal operator. A gradient-step denoiser (see Hurault et al., 2022, Section 3, Proposition 3.1) represents f = Id - \nabla g, where \nabla g is $1$-Lipschitz. Such an f is always the proximal operator of some function, but by Gribonval and Nikolova’s characterization, it does not capture all proximal operators (because this $f$ might be (say) 2-Lipschitz).
> > >
> > > Please note as well that this restriction is a nontrivial modeling limitation. A simple example is furnished by the prior distribution (in a Bayesian estimation setting) $p(x = +1) = p(x = -1) = \tfrac{1}{2}$ (i.e., a Rademacher distribution), a simple model for the common setting where one wants to learn a prior for multimodal data. This prior has as the proximal operator associated to its negative logarithm (roughly speaking) the function $x \mapsto \sign(x)$. Any sequence of smooth proximal operators approximating this operator pointwise necessarily have their Lipschitz constants grow beyond $2$ (in fact, to $+\infty$).
> > >
> > > For the second issue, we understand your concerns about the neural network architecture (the ICNN) we employ in LPNs. First, let us note here that ICNNs are universal approximators of continuous convex functions on $[0, 1]^d$, by work of Huang et al. (2021), which suggests that in principle the parameterization need not be overly restrictive. On the other hand, we acknowledge that in practice, the parameterizations of ICNNs that we experiment with in the submission are constraining, and may not be rich enough in specific practical problems. Better and more expressive parameterization of ICNN will benefit the generality of LPN. Our experimental evidence support the observation that the chosen parametrization is not too restrictive (see below further comments on Experiments).
> > >
> > > We emphasize that we think this should be seen as a promising direction for future research that our work suggests, given that we are able to obtain competitive performance in our experiments with such minimal ICNN architectures. Indeed, we have barely scratched the surface on ways to import ideas from modern denoisers (e.g., bias-free denoisers, U-nets, etc., as Hurault et al. (2022) employ in their GS-denoiser) to the input-convex setting.
> > >
> > > We have adjusted the language/tone in the paragraph at the top of page 4 that makes this comparison in the revised manuscript. Please let us know if you have further concerns on this front.

---

> > > > ### Author Response · Authors · 2023-11-21
> > > > **Rebuttal continued 3**
> > > >
> > > > Continued from response to 11)
> > > >
> > > > **On the guarantee for learning a proximal operator**
> > > >
> > > > We would like to add another point on the comparison between our method and GS denoisers. Our method provides the guarantee that the learned denoiser is an exact proximal operator, whereas GS network does not. It is true that, theoretically, GS denoisers can be enforced to parameterize an exact proximal operator by constraining its Lipschitz constant, as presented in Hurault et al. 2022. However, in practice, such a constraint is not strictly enforced, but instead realized by adding a regularization term on the spectral norm of the network during training. Such a regularization only penalizes large Lipschitz constants, but does not guarantee that the Lipschitz constant will be lower than the required threshold. Furthermore, the regularization is only computed at training data points, thus does not regularize the network’s behavior globally. In other words, the GS denoiser is only “encouraged” to resemble a proximal, but it is not guaranteed. On the other hand, our LPN provides the guarantee that the learned network will always parameterize a proximal operator.
> > > >
> > > > **Experimental comparison**
> > > >
> > > > As suggested by the reviewer, we have now expanded the experimental section by comparing with the GS denoiser in Hurault et al. (2022). Please see Figure 5 and Table 1 for the new experiments. We observe that the results from LPN are comparable and on-par with those from GS networks, illustrating the competitive performance of our method compared to the state-of-the-art. Note that our method provides the added advantage of providing an exact proximal operator – which GS networks might not (as we explained above).
> > > >
> > > > To sum up, we agree with the reviewer that our original comparison between LPN vs. GS denoiser is incomprehensive, and thank the reviewer for pointing this out. We have provided a more thorough comparison, both conceptually and experimentally, between LPN and GS networks in the revised manuscript to better illustrate the pros and cons of both methods.
> > > >
> > > > * Hurault, Samuel, Arthur Leclaire, and Nicolas Papadakis. "Proximal denoiser for convergent plug-and-play optimization with nonconvex regularization." ICML 2022.
> > > >
> > > > 12. > In most PnP algorithms, the Gaussian noise parameter on which the denoiser is trained acts as a regularization parameter. Here, you do not mention this parameter, and you do not explain how it is chosen.
> > > >
> > > > As we note at the end of the conclusion of Theorem 3.1, the noise standard deviation $\sigma$ acts as a scaling factor on the learned prior. In experiments, we did not explore this parameter extensively and used sigma=1 for Laplacian, sigma=0.05 or 0.1 in all image experiments. These details are included in Section E. We have added an extra remark following Theorem 3.1 that explicitly mentions this role of $\sigma$ as a regularization scale.
> > > >
> > > > ----
> > > > ## Final comments:
> > > >
> > > > > These weaknesses explain my low score, and I am ready to raise my score if the authors answer these limits.
> > > >
> > > > We look forward to hearing if our comments have addressed the reviewer’s concerns, and we’d be happy to address any outstanding points.

---

> ### Comment · Reviewer_XDuj · 2023-12-04
>
> Thank you for the detailed answer. The majority of my questions have been clarified. In particular, the clarification on o-minimal structures improves significantly the proof.
>
>  I have however still some doubts on the following points.
> - First the authors experiment with the ADMM algorithms when they showed convergence of the PGD algorithm. This is for me a major limitation and I do not understand the point of proving convergence if not experimenting with the same algorithm afterwards.
> - Second the authors compare with the literature on the Celeba dataset. Their network is trained on Celeba when the compared methods seems not to be trained on Celeba. This looks like unfair comparison.
>
> I decided to raise my score to 5.

---

### Official Review · Reviewer_hkxu · 2023-10-30

**Soundness:** 3 good
**Presentation:** 2 fair
**Contribution:** 2 fair
**Rating:** 5
**Confidence:** 3

**Summary:**

The paper presents a framework for constructing learned proximal networks (LPN) that offer precise proximal operators for a data-driven regularizer. It also demonstrates a novel training strategy called proximal matching, which ensures that the resulting regularizer accurately captures the log-prior of the actual data distribution. Experiments test the effectiveness.

**Strengths:**

1.	The paper forms a class of neural networks to guarantee to parameterize proximal operators. The idea is novel.
2.	Some theoretical results are provided.
3.	Experiments show the effectiveness.

**Weaknesses:**

1.	The paper is not well organized, making it hard to follow.
2.	The experiments are a little weak. See the questions below.

**Questions:**

My main concerns are the experimental details.
1.	It is better to give some details on the training of the learned proximal networks. For example, how to learn the non-negative weights?
2.	How to set the \alpha in \phi_\theta in the training?
3.	How to set \eta in Theorem 4.1 in the training?
4.	The methods compared in this paper are relatively old. It is better to compare the recent methods. Then we could find out the superiority of the proposed method.
5.	The used datasets are very small. Deep learning can only be effective when there is a substantial amount of data.
6.	It is best to verify the effectiveness of the proposed method on common real tasks, e.g., real denoising or deblurring.
7.	Compared with other methods, how is the training efficiency? It is better to verify it by experiment.
8.	At the end of the Abstract, "demonstrating" should be "Demonstrating".

---

> ### Author Response · Authors · 2023-11-21
> **Rebuttal**
>
> We thank the reviewer for their questions and comments.
>
> 1. > It is better to give some details on the training of the learned proximal networks. For example, how to learn the non-negative weights?
>
> Thank you for the question. We have provided extensive training details, such as learning rate, Gaussian noise level, schedule of proximal matching $\gamma$, and batch size in the Appendix E. Sorry for missing the details about how to enforce non-negative weights. During training, we apply weight clipping, i.e., projecting negative weights to 0, at each iteration to ensure the weights are nonnegative. Additionally, in the experiments on CelebA and Mayo-CT, we enforce the weights to be nonnegative at initialization by initializing the weights according to a Gaussian distribution and then taking the exponential. We observed that this initialization helped the training converge faster. We have added these details to Appendix E. In addition, we will also release our code publicly for reproductivity.
>
> In Appendix E.1, Details of Laplacian Experiment:
> * To enforce nonnegative weights, weight clipping is applied, projecting the negative weights to zero at each training iteration.
>
> In Appendix E.3, Details of CelebA Experiment:
> * We observed that initializing the respective weights to be nonnegative, by initializing them according to a Gaussian distribution and then taking the exponential, helped the training converge faster. Therefore, we applied such initialization in the experiments on CelebA and Mayo-CT. The same weight clipping as in Appendix E.1 is applied to ensure the weights stay nonnegative throughout training.
>
> 2. > How to set the \alpha in \phi_\theta in the training?
>
> This is a user-defined parameter. We discussed its impact and trade-off in Section 3, the “Recovering the prior from its proximal” paragraph, and footnote 4. The values we used for each experiment are included in Appendix E.
>
> 3. > How to set \eta in Theorem 4.1 in the training?
>
> The $\eta$ is the step size of gradient descent while using a trained LPN for solving inverse problems in the PnP framework. It is not a parameter during training. Sorry about the confusion. As noted in Theorem 4.1, $\eta$ should be between 0 and $1 / \|A^TA\|$ to guarantee convergence of the PnP iterates with LPN. In practice, $\eta$ is a hyperparameter that can be tuned by the user.
>
> 4. > The methods compared in this paper are relatively old. It is better to compare the recent methods. Then we could find out the superiority of the proposed method.
>
> We thank the reviewer for this suggestion. Indeed, we have now added further comparison with a newer, highly relevant and state-of-the-art method (which was also suggested by another reviewer) :
>
> * Hurault, Samuel, Arthur Leclaire, and Nicolas Papadakis. "Proximal denoiser for convergent plug-and-play optimization with nonconvex regularization." ICML 2022.
>
> Note, however, that no other method exists that approximates the solution of a MAP problem based on learned networks that provide exact proximal operators. Our LPN is the first to provide such a guarantee.
> We refer the reviewers to Figure 5 and Table 1 in the revised manuscript for the new results. The results from LPN are comparable and on-par with those from GS networks, while our method provides the added advantage of providing an exact proximal operator – which GS networks might not (we explain in detail why this is the case in our response to Reviewer 4, comment 11).

---

> > ### Author Response · Authors · 2023-11-21
> > **Rebuttal continued**
> >
> > 5. > The used datasets are very small. Deep learning can only be effective when there is a substantial amount of data.
> >
> > Thank you for the comment. We would like to note that the datasets used in this study have been used in a variety of relevant works, all of them employing deep learning. For example:
> >
> > The Mayo-CT dataset has been used in the following works
> > * Sebastian Lunz, Ozan  ̈Oktem, and Carola-Bibiane Sch ̈onlieb. Adversarial regularizers in inverse problems. NeurIPS 2018.
> > * Subhadip Mukherjee, Marcello Carioni, Ozan  ̈Oktem, and Carola-Bibiane Sch ̈onlieb. End-to-end reconstruction meets data-driven regularization for inverse problems. NeurIPS 2021.
> >
> > The CelebA dataset has been used in
> > * Bora, Ashish, Ajil Jalal, Eric Price, and Alexandros G. Dimakis. "Compressed sensing using generative models." ICML 2017
> > * Gilton, Davis, Greg Ongie, and Rebecca Willett. "Neumann networks for linear inverse problems in imaging." IEEE TCI 2019
> > * Asim, Muhammad, Max Daniels, Oscar Leong, Ali Ahmed, and Paul Hand. "Invertible generative models for inverse problems: mitigating representation error and dataset bias." ICML 2020.
> > * Liu, Tianci, Tong Yang, Quan Zhang, and Qi Lei. "Optimization for Amortized Inverse Problems." ICML 2023
> > * Delbracio, Mauricio, and Peyman Milanfar. "Inversion by direct iteration: An alternative to denoising diffusion for image restoration." TMLR 2023.
> >
> > The fact that our numerical experiments demonstrate state-of-the-art restoration results indicate that the amount of data in these cases is appropriate. Naturally, better results can be expected with larger datasets, and experimenting with LPN in other settings constitutes ongoing work.
> >
> > 6. > It is best to verify the effectiveness of the proposed method on common real tasks, e.g., real denoising or deblurring.
> >
> > Indeed, we verified the effectiveness of the proposed method on common real tasks in our original manuscript, including deblurring (Fig. 5 and Table 1), tomography reconstruction and compressed sensing (Fig. 4 and Table 2).
> >
> > 7. > Compared with other methods, how is the training efficiency? It is better to verify it by experiment.
> >
> > Thank you for the comment. The training time varies with dataset, network size, training parameters and hardware specifications. As an example, training the LPN on the CelebA dataset as in our experiment takes about 6 hours on a NVIDIA RTX A5000 GPU. We consider it to be relatively efficient, partly due to our efficient network size. We will release our code publicly, for anyone interested to be able to train LPN in their own setting. We have added the information about training time to Appendix E.3 (see below).
> >
> > * The training time of LPN on the CelebA dataset is about 6 hours on a NVIDIA RTX A5000 GPU.
> >
> > 8. > At the end of the Abstract, "demonstrating" should be "Demonstrating".
> >
> > Thank you for spotting this and we have corrected it. Please see below.
> >
> > * "We illustrate our results in a series of cases of increasing complexity, demonstrating that these models not only result in state-of-the-art restoration results, but provide a window into the resulting priors learned from data."

---

### Official Review · Reviewer_8FVJ · 2023-10-30

**Soundness:** 3 good
**Presentation:** 3 good
**Contribution:** 3 good
**Rating:** 8
**Confidence:** 3

**Summary:**

Inverse problems $y = A(x) + v$ are commonly formulated using regularized least squares:
$$\min_x \frac{1}{2} \|\| y - A(x) \|\| + \phi(x)$$

where $\phi$ is an appropriate regularizer. Solving this minimization problem often involves the use of proximal gradient methods. This process necessitates the selection of a regularizer, $\phi$, and the knowledge of its proximal map. A natural approach, leading to Maximum A Posteriori (MAP) estimation, is to choose $\phi(x) = -\log p(x)$.

This paper presents a method for learning the proximal map of $\phi(x) = -\log p(x)$ from data, using only samples from $p(x)$. The authors then apply this proposed approach, in conjunction with proximal gradient methods, to various standard inverse problem tasks, demonstrating favorable performance.

**Strengths:**

The use of proximal and descent methods in conjunction with deep neural networks to address inverse problems is a dynamic and exciting field of study. Numerous papers have explored these approaches, attempting to approximate Maximum A Posteriori (MAP) estimation in various ways or training regularizers in a supervised manner (which is not always feasible and not even correct). To the best of my knowledge, this paper stands out as the first to present a principled approach for training proximal maps of the log probability and effectively approximating MAP estimation.

Additionally, the numerical results presented in this paper are well-executed and show promise.

**Weaknesses:**

- A straightforward solution to obtain the proximal map of log p(x) is to initially train an energy-based model, E, and then compute the proximal map of og E. The authors should either compare this approach with their proposed method or provide clarification on why this is not considered viable or advisable. For instance, one might anticipate encountering similar challenges as those faced when training energy models when training Learned Proximal Networks (LPNs).

- In a similar vein, a natural point of comparison for the proposed approach would involve using other state-of-the-art unsupervised methods that rely on generative models as priors. While the paper does make a comparison with the less recent adversarial regularizer, it would be valuable to assess its performance against other more recent unsupervised methods.

---
After rebuttal: The authors have addressed the concerns raised in the reviews and I have increased my score.

**Questions:**

- Equation (3.2) should it be \psi_\thet(x,\alpha) ?

- Page 5, "demillustrates" should be corrected to "illustrates."

- Page 5, the terminology "proximal operator of target distribution" is frequently used but remains undefined. This terminology appears to be uncommon in the field and should be clarified or explained for the readers' benefit.

- It would be insightful to determine whether the log-likelihood computed using LPN matches the log-likelihood computed using other generative models.

- "Hidden" in the appendix is that the LPN is first trained on the l1-loss, then it is trained with the proximal matching loss. This should be stated clearly in the main body of the paper. Why this is needed? How bad is LPN trained just with the proximal matching loss?

---

> ### Author Response · Authors · 2023-11-21
> **Rebuttal**
>
> Thank you for your questions and comments.
>
> 1. > A straightforward solution to obtain the proximal map of log p(x) is to initially train an energy-based model, E, and then compute the proximal map of og E. The authors should either compare this approach with their proposed method or provide clarification on why this is not considered viable or advisable. For instance, one might anticipate encountering similar challenges as those faced when training energy models when training Learned Proximal Networks (LPNs).
>
> Thanks for bringing up this very relevant comment. There are two main reasons why the above suggested by the reviewer is not straightforward: First, training an energy-based generative model E is a statistically complex task, particularly for distributions supported in high-dimensional spaces. While it is true that good approximate models can be obtained, computing the proximal operator will then boil down to the explicit optimization of a non-convex problem, precluding us from any precise guarantees. Our approach circumvents both of these by computing the exact proximal directly (even if the prior is non-convex), i.e., by learning to compute the maximum of a posterior, which is a significantly easier task than computing the complete posterior and then finding its maximum (which is the approach entailed by the energy model).
>
> 2. > In a similar vein, a natural point of comparison for the proposed approach would involve using other state-of-the-art unsupervised methods that rely on generative models as priors. While the paper does make a comparison with the less recent adversarial regularizer, it would be valuable to assess its performance against other more recent unsupervised methods.
>
> Thank you for the suggestion. In our revised manuscript, we have added a new and more recent comparison method (Hurault et al. 2022), based on the Gradient Step (GS) denoiser.
>
> * Hurault, Samuel, Arthur Leclaire, and Nicolas Papadakis. "Proximal denoiser for convergent plug-and-play optimization with nonconvex regularization." ICML 2022.
>
> We chose this method because it is closely related to ours, for the following reasons. First, this method also concerns the topic of solving inverse problems via a variational setup and seeking the Maximum a Posteriori estimate with a data-driven prior. Second, this method also attempts to train a denoiser as a proximal mapping and study convergence guarantees when using the trained denoiser within the PnP framework. Lastly, this method was also suggested by another reviewer as a state-of-the-art method that is comparable to our proposed methodology. Therefore we believe it is a very relevant comparison.
>
> We refer the reviewers to Figure 5 and Table 1 in the revised manuscript for the new results. Our results are comparable and on-par with those from the GS networks, while our method provides the added advantage of providing an exact proximal operator – which GS networks might not (we explain in detail why this is the case in our response to Reviewer 4, comment 11).
>
> **Questions**:
>
> * Equation (3.2) should it be \psi_\thet(x,\alpha) ?
>
> Right. Thank you for pointing this out and we have corrected it.
>
>
> * Page 5, "demillustrates" should be corrected to "illustrates."
>
> Fixed, thanks!
>
> * Page 5, the terminology "proximal operator of target distribution" is frequently used but remains undefined. This terminology appears to be uncommon in the field and should be clarified or explained for the readers' benefit.
>
> Thank you for the suggestion. It means the proximal operator of the log-prior of the data distribution we aim to learn, i.e. prox_{-log p_x}. We have rephrased it in the paper:
>
> Page 5: "Figure 2 illustrates the limitations of these distance metrics for learning the true proximal operator of the log-prior of the underlying data distribution (a Laplacian, in this example)."
>
> Moreover, to avoid any further confusion, we have explicitly defined “log-prior” the first time it appears in the paper: "In this paper, the “log-prior” of a data distribution px means its negative log-likelihood, −log px."
>
> * It would be insightful to determine whether the log-likelihood computed using LPN matches the log-likelihood computed using other generative models.
>
> We agree with this comment. However, we believe it is out of the scope of this work, which focuses on solving inverse problems with data-driven priors. Note that in this work, we did not attempt to use LPN as a generative model (e.g., using LPN to generate human faces from scratch), or compare it with any generative models. However, we do believe LPN has the potential to be adapted into generative models and we are very interested in exploring this connection in future work.

---

> > ### Author Response · Authors · 2023-11-21
> > **Rebuttal continued**
> >
> > * "Hidden" in the appendix is that the LPN is first trained on the l1-loss, then it is trained with the proximal matching loss. This should be stated clearly in the main body of the paper. Why is this needed? How bad is LPN trained just with the proximal matching loss?
> >
> > We pretrained LPN with L1 loss to provide a better initialization for proximal matching training. We have added this detail in the main body in Section 3.1.

---

> > > ### Comment · Reviewer_8FVJ · 2023-11-21
> > >
> > > Dear authors,
> > >
> > > Thank you for the clarifications and for addressing my concerns. I will raise my score to reflect the improvements.

---

### Official Review · Reviewer_fi4Y · 2023-11-06

**Soundness:** 2 fair
**Presentation:** 1 poor
**Contribution:** 2 fair
**Rating:** 5
**Confidence:** 3

**Summary:**

Plug-and-play methods are a framework used for solving inverse problems in image processing, and rely on proximal operators. It's been shown that the proximal operator for the linear inverse problem case is equivalent to a MAP denoiser. In the literature PnP methods almost always rely on MMSE denoiser due to difficulty of learning and designing MAP estimators. In this work, the authors suggest a way to learn a MAP denoiser which they then use in an ADMM algorithm to solve inverse problems. They show the algorithm converges to fixed points of the prior for a particular design of neural networks which guarantees strongly convex mapping. Finally they test the algorithm empirically on toy Laplacian distribution as well as MNIST, CelebA and Mayo-CT datasets.

**Strengths:**

This work addresses a prevalent problem at the core of PnP methods, which is the use of MMSE denoisers instead of MAP denoisers, despite the fact that the convergence results for PnP holds for MAP denoisers. They offer a novel way to learn a MAP denoiser and provide convergence results.

**Weaknesses:**

- The main contribution of the work is to propose a way to learn a MAP denoiser through a proximal loss under equation 3.4. Optimizing for this loss entails that the prior distribution is assumed to be a mixture of Gaussians (or Diracs when $\gamma$ tends to zero) around the training samples. Why is this a good prior? It seems to me that is a too simplistic prior and in the limit of $\gamma \to 0$ a discontinuous prior.
- Although most of the paper was quite clear I found it unclear whether the final implementation enforced convexity on the prior or not. Under section 2 and section 3, a case was made for convexity: a class of learned proximal network was suggested to ensure convexity, and later under (3.1) another regularization was introduced to ensure strong convexity for recovering the prior. I think these sections do not flow well and the thread of logic is easily lost by the reader. The clarity can be improved. Also on a related note, if convexity is assumed, why is it a proper assumption? It most probably would be a too simplistic of an assumption for image priors.
- The details of the algorithm can be more elaborated in the main text. Specially, since the algorithm is very similar to the basic score-based diffusion generative algorithms for solving inverse problems, the differences between the two should be pointed out.
- I find it surprising and contradictory that the PSNR values are higher for this methods compared to other methods that use MMSE denoisers. It is expected that for solving inverse problems using a MAP denoiser result in sharper images with lower PSNR (=higher MSE) while using MMSE denoiser result in higher PSNR (=lower MSE) and more blurry results. This is obvious since, as also stated by the authors, the MMSE denoiser pushed the image towards the mean of the posterior while MAP denoiser pushes the image towards the mode of the posterior. Obviously the mean of the posterior is equivalent to the MSE minimizer, so it should result in lower MSE.
- Additionally, I found it confusing that the qualitative results for CelebA dataset seem pretty much the same across different methods, and not very good in general. The results look pretty blurry for the proposed method which is surprising given that the method relies on a MAP denoiser.
- Finally, in recent years, there has been some score based diffusion models algorithms which use priors in denoisers to solve linear inverse problems, which result in significantly better performance. Considering the disadvantage in performance, what are the advantages of this methods? It seems necessary to include a comparison to those models.

**Questions:**

Please see the questions raised under the weaknesses section.

---

> ### Author Response · Authors · 2023-11-21
> **Rebuttal**
>
> Thank you for your questions and comments.
>
> 1.  > The main contribution of the work is to propose a way to learn a MAP denoiser through a proximal loss under equation 3.4. Optimizing for this loss entails that the prior distribution is assumed to be a mixture of Gaussians (or Diracs when gamma tends to zero) around the training samples. Why is this a good prior? It seems to me that is a too simplistic prior and in the limit of gamma -> 0 a discontinuous prior.
>
> Not quite: the proximal matching loss (Eq. 3.4) does not require the prior distribution to be a mixture of Gaussians or Diracs. There is almost no assumption on the prior to be learned, except the most basic regularity conditions, e.g., for the prior to be continuous in Thm 3.1 (and discrete in Thm B.1). Note that in the paper, immediately after Eq. 3.4, we do mention that the loss \ell_\gamma can be thought of as an \gamma-approximation to a Dirac, but this is just for intuition, and no conditions are imposed on the actual priors.
>
> 2. > Although most of the paper was quite clear I found it unclear whether the final implementation enforced convexity on the prior or not. Under section 2 and section 3, a case was made for convexity: a class of learned proximal network was suggested to ensure convexity, and later under (3.1) another regularization was introduced to ensure strong convexity for recovering the prior. I think these sections do not flow well and the thread of logic is easily lost by the reader. The clarity can be improved. Also on a related note, if convexity is assumed, why is it a proper assumption? It most probably would be a too simplistic of an assumption for image priors.
>
> We believe we see the confusion of the reviewer: We do not enforce convexity on the prior at any point. The review is correct in that convex priors are not well suited for natural images, but note that our method can learn nonconvex priors, as has been demonstrated in the MNIST example (Fig. 3b), and we view this as a main advantage of our method. We will further clarify on this point - thank you.
> In our presentation, while the function $\psi$ is convex, note that it is not the prior - the prior is denoted by $\phi$. Convexity on $\psi$ is enforced to ensure the LPN, which is defined as the gradient of $\psi$, is an exact proximal operator, as required by Proposition 2, which uses the fact that proximal operators of nonconvex priors are equivalent to gradients of convex functions. Please see Figure 1 for a clear illustration of this relation.
>
> 3. >The details of the algorithm can be more elaborated in the main text. Specially, since the algorithm is very similar to the basic score-based diffusion generative algorithms for solving inverse problems, the differences between the two should be pointed out.
>
> We thank the reviewer for this question. While our methodology is used to solve inverse problems, PnP in general, and our LPN in particular, are significantly different from score-based diffusion models. First: conditional diffusion models do not minimize a variational problem like we do in this paper (as in Eq. 2.1), but instead provide samples from the posterior distribution. Second: score-based sampling arises as the result of inverting a diffusion process which requires access to an MMSE denoiser, whereas we are only concerned with networks that compute a MAP estimate for a learned prior. Third, the reviewer should note that our LPN provides a (MAP) solution in a single call (or forward pass) for the denoising task, whereas sampling from the posterior as with diffusion models requires very expensive computation.
> For all these reasons, comparing with diffusion based models is out of the scope of this work. However, based on your suggestion, we have included a detailed discussion of these points in our revised version of the paper (please see Appendix A, “Comparison to Diffusion Models” paragraph). If the reviewer has specific works in mind that we should reference and comment on, we would be happy to do so.
> In terms of algorithm details, we have included the pseudo-code in Algorithms 1 and 4, and details of hyperparameters in Section E. We will also release our code publicly.

---

> > ### Author Response · Authors · 2023-11-21
> > **Rebuttal continued**
> >
> > 4. > I find it surprising and contradictory that the PSNR values are higher for this methods compared to other methods that use MMSE denoisers. It is expected that for solving inverse problems using a MAP denoiser result in sharper images with lower PSNR (=higher MSE) while using MMSE denoiser result in higher PSNR (=lower MSE) and more blurry results. This is obvious since, as also stated by the authors, the MMSE denoiser pushed the image towards the mean of the posterior while MAP denoiser pushes the image towards the mode of the posterior. Obviously the mean of the posterior is equivalent to the MSE minimizer, so it should result in lower MSE.
> >
> > There is no contradiction in our MAP-based restoration algorithm being better than methods based on (approximate) MMSE denoisers: “black-box” denoisers do indeed maximize PSNR for a denoising task, however this does not mean that the obtained PSNR for a general inverse problem as in Eq. 2.1 (with a general forward operator A) will be the highest for those methods. In other words: denoising methods provide an approximate MMSE estimate for a denoising task, but not for the general inverse problem. As the reviewer will have noticed, when these general inverse problems are solved via ADMM (or PGD), an exact MAP is needed. Our method provides such an exact MAP estimate, whereas MMSE denoisers provide iterates that are not perfectly suited for solving the overall optimization problem (which indeed requires the MAP). This is exactly what our numerical results illustrate.
> >
> > 5. > Additionally, I found it confusing that the qualitative results for CelebA dataset seem pretty much the same across different methods, and not very good in general. The results look pretty blurry for the proposed method which is surprising given that the method relies on a MAP denoiser.
> >
> > The reviewer should keep in mind that these images are relatively small (128x128) which is likely responsible for the reviewer’s impressions. Furthermore, this is a typical benchmark for unsupervised restoration methods (Gilton et al. IEEE TCI 2019, Wang et al. ICLR 2023) and we follow the experimental setting in those works. Finally, note that other reviewers commented that our “numerical results are well-executed”.
> >
> > * Gilton, Davis, Greg Ongie, and Rebecca Willett. "Neumann networks for linear inverse problems in imaging." IEEE Transactions on Computational Imaging, 2019.
> > * Wang, Yinhuai, Jiwen Yu, and Jian Zhang. "Zero-shot image restoration using denoising diffusion null-space model." ICLR 2023.

---

> > > ### Author Response · Authors · 2023-11-21
> > > **Rebuttal continued 2**
> > >
> > > 6. > Finally, in recent years, there have been some score based diffusion models algorithms which use priors in denoisers to solve linear inverse problems, which result in significantly better performance. Considering the disadvantage in performance, what are the advantages of these methods? It seems necessary to include a comparison to those models.
> > >
> > > It is true that score-based diffusion models have proven very efficient for unconditional and conditional image generation. However, these models rely on very different modeling assumptions (e.g. by means of a reversed diffusion process), require training denoising networks that can provide estimates for varying degrees of noise, and -just as in the case of regular denoisers- they do not approximate any MAP estimate. In terms of strict advantages, one should again note that our approach solves (provides a MAP estimate) for a denoising problem with a single forward-pass, whereas diffusion models require a large sequence of forward passes of a denoising network.
> > >
> > > Lastly, but also importantly, our method provides an exact proximal operator for a learned prior distribution. Diffusion models have no such guarantee: all these results provide samples from an approximate posterior distribution, which relies on the approximation qualities of the MMSE denoiser - which do not exist for general cases (see Chen, Lee, and Lu, "Improved analysis of score-based generative modeling: User-friendly bounds under minimal smoothness assumptions." ICML, 2023.)
> > > We expand on this in a paragraph explaining the similarities and differences with diffusion models in our revised version of the manuscript (see Appendix A “Contrast to Diffusion Models”).
> > >
> > > We want to further point out that we are approaching the problem of solving inverse problems from a fundamentally different perspective than is adopted in works on score-based generative modeling of images and their application to solving inverse problems. These diffusion works are concerned with obtaining the best possible visual performance in obtaining samples from an underlying data distribution – interpretability or explainability of the predictions in these methods are not a design consideration. In particular, the score approximating network (denoiser) in a score-based diffusion model pipeline is a black-box neural network architecture, with only empirically tried-and-tested design incorporated. Our methodology is fundamentally motivated by incorporating known properties of the inverse problem into the solution for reconstruction, and understanding how learning and data-depending functions can be incorporated into precise and well-understood variational formulations.

---

> > > > ### Comment · Reviewer_fi4Y · 2023-11-21
> > > >
> > > > Thank you for your responses. Since I think most of the comments (except for #3 ) do not resolve the questions, I keep my initial score.

---

> > > > > ### Author Response · Authors · 2023-11-21
> > > > > **Clarifications**
> > > > >
> > > > > Thank you for your comment. Could you please explain why the above do not resolve your questions?
> > > > >
> > > > > In particular, many of these were misconceptions about our work that we have clarified. If there are further objections or disagreements with our responses, we kindly ask that you provide some additional details. We are happy to discuss and clarify further.

---

### Author Response · Authors · 2023-11-21
**Rebuttal, general comments**

We thank all the reviewers for their constructive comments, time, and effort to help improve the manuscript. We summarize our responses to common questions raised by the reviewers and our major revisions below, before proceeding to the responses to each individual comment.

1. Some reviewers have argued a lack of comparisons with state-of-the-art restoration models, like those based on diffusion. We explain below why these models are fundamentally different than the method proposed in this work, and why thus we believe this comparison to be out of the scope of the current manuscript. Our contribution deals exclusively with those data-driven methods that attempt to employ proximal operators, and those that solve a standard variational problem as a reconstruction task (as opposed to, say, providing samples from a conditional distribution, which is the aim of diffusion models). In this way, our contribution focuses on providing a rigorous theoretical tool to develop precise and principled proximal networks, making these algorithms more interpretable. We want to stress the main contribution of this work is to demonstrate that the convex parameterization for prox operators is practically viable (achieving very strong performance with minimal tuning) thanks to our novel prox matching training framework, and therefore our work should have significant opportunity for follow-on impact.

2. Based on some of the reviewers’ suggestions, we have now expanded our numerical comparison and have included new and more recent methods, such as those based on Gradient Step (GS) denoiser networks (Hurault et al. 2022). We refer the reviewers to our response to Reviewer 2, comment 2 for why this method is closely related to ours and highly relevant for comparison. We also refer the reviewers to Figure 5 and Table 1 in the revised manuscript for the new results, which demonstrate that our results are comparable and on-par with those from GS networks. Additionally, our method provides the added advantage of providing an exact proximal operator – which GS networks might not (we explain in detail why this is the case in our response to Reviewer 4, comment 11).

3. We thank Reviewer 4 for pointing out the initial glitch and imperfection in our proof. We have updated the proofs and fixed the raised issues, and we would like to emphasize here that none of the theoretical conclusions is changed because of the proof modification. That is, all the theoretical results in the original manuscript still hold true. Please see the response to individual comments for details.

After addressing every comment raised by all the reviewers, we believe the quality of the manuscript has been substantially improved. We ask the reviewers to kindly evaluate the manuscript in its new shape and we are ready to address any further comments from the reviewers.

* Hurault, Samuel, Arthur Leclaire, and Nicolas Papadakis. "Proximal denoiser for convergent plug-and-play optimization with nonconvex regularization." ICML 2022.

---

### Comment · Area_Chair_aCnj · 2023-11-22

Dear all,

The author-reviewer discussion period is about to end.

@authors: If not done already, please respond to the comments or questions reviewers may further have. Remain short and to the point.

@reviewers: Please read the author's responses and ask any further questions you may have. To facilitate the decision by the end of the process, please also acknowledge that you have read the responses and indicate whether you want to update your evaluation.

You can update your evaluation positively (if you are satisfied with the responses) or negatively (if you are not satisfied with the responses or share other reviewers' concerns). Please note that major changes are a reason for rejection.

You can also keep your evaluation unchanged. In this case, please indicate that you have read the responses, that you do not have any further comments and that you keep your evaluation unchanged.

Best regards,
The AC

---

### Meta-Review · Area_Chair_aCnj · 2023-12-10

**Metareview:**

The reviewers have mixed opinions about the paper (5-8-5-5), tending towards rejection. This paper is concerned with general inverse problems. It introduces Learned Proximal Networks (LNPs) and formally shows they parameterize proximal operators. The paper also contributes a learning strategy, proximal matching, to ensure that LNPs recover the correct proximal of the data distribution. Experiments are thorough and convincing. The authors-reviewers discussion has been constructive and has led to a number of improvements to the paper. The authors have actively engaged with the reviewers and have addressed most of their concerns. Unfortunately, the reviewers have not engaged much with the authors. Specifically, i) it is not clear if Reviewer fi4Y (5) has understood the paper, which could have been clarified by a discussion with the authors, ii) Reviewer hkxu (5) has not replied to the authors' comments, and iii) most of Reviewer XDuj's concerns have been addressed by the authors, leading to borderline rejection (5).

Overall, I believe the paper presents a well-rounded contribution, which is both theoretically sound and empirically convincing. The authors have made a number of improvements to the paper during the author-reviewer discussion period. To me, the paper presents sufficient novelty, interest and quality to be accepted. I recommend acceptance. I also encourage the authors to address the remaining concerns of Reviewer XDuj in the final version of the paper.

**Justification For Why Not Higher Score:**

More rejects than accepts, but overall a solid contribution.

**Justification For Why Not Lower Score:**

The reviewers have not engaged much, which is unfortunate. I believe stronger interactions would have eventually improved the scores obtained by this paper. The paper is good.

---

### Decision · Program_Chairs · 2024-01-16

Accept (poster)